# Continuous Time Series Generation with Irregular Observations

## Abstract

Time series generation (TSG) contributes to diverse fields (e.g., healthcare), but most methods assume regularly sampled data and fixed outputs—mismatched with real-world settings where observations are irregular and sparse. This mismatch is especially problematic in domains such as clinical monitoring, where irregularly recorded data must support downstream tasks with continuous and high-resolution data. Neural Controlled Differential Equations (NCDEs) show significant potential in handling irregular time series, but still face challenges in learning dynamic temporal patterns and continuous TSG. To address this, we propose MN-TSG, a framework that explores MOE (Mixture of Experts)-NCDE and integrates it with existing TSG models for irregular or continuous TSG tasks. The key designs of MOE-NCDE are the dynamic functions with mixture of experts and the decoupled design to better optimize the MOE dynamics. Further, we employ the existing TSG model to learn the joint distribution of the mixture of experts and the time series. In this way, the model can not only generate new samples but also produce suitable experts for them to enable MOE-NCDE for refined continuous TSG tasks. We have validated the effectiveness of our method on ten public and synthetic datasets, outperforming advanced TSG baselines in both irregular-to-regular and irregular-to-continuous generation tasks. The code and datasets are available at the link https://anonymous.4open.science/r/MNTSG-79C2.

## 1 Introduction

Time series generation (TSG) plays a pivotal role across a diverse range of real-world domains, such as healthcare (Deng et al., 2025), finance (Vasiliu et al., 2024), energy systems (Fuest et al., 2025), and climate science (Ye et al., 2024) by producing synthetic sequences that facilitate data augmentation for low-resource contexts (Tang et al., 2025), enable privacy-preserving data sharing (Tian et al., 2024), and support high-fidelity simulations for critical decision-making (Turowski et al., 2024). While numerous TSG methods have been developed, they generally build upon a fundamental assumption that the time series (both training data and generated data) are regular, which is represented as discrete observations with equally spaced timestamps. As depicted in Fig. 1(a), most existing works treat training sequences as uniformly observed and produce outputs sampled at fixed intervals (Mushunje et al., 2023; Li, 2023; Huang et al., 2025; Ge et al., 2025). As shown in Fig. 1(b), few approaches relax the first assumption—accepting irregularly observed training data—but still generate outputs with fixed intervals (Nikitin et al., 2024; Ramponi et al., 2018). This mismatch contrasts sharply with real-world scenarios, where data often arrives irregularly, yet downstream applications demand continuous, high-resolution sequences. For example, in clinical settings like ICU monitoring, electronic health record entries may be recorded at irregular and sparse intervals, sometimes with up to 80% missingness in some features, but tasks like sepsis forecasting or patient trajectory simulation require continuous, temporally dense data (Tian et al., 2024; Li et al., 2025). Thus, continuous time series generation from irregular observations is a pressing and under-addressed challenge within the TSG community.

Continuous time series modeling and the challenge of irregular observations have been actively studied, with several techniques proposed. Among all the techniques, Neural Ordinary Differential Equations (NODE) and its following work, Neural Controlled Differential Equations (NCDE) have been particularly popular, which we will refer to as NCDE-based methods. They combine differential equations in continuous time with learnable neural networks to approximate temporal dynamics.

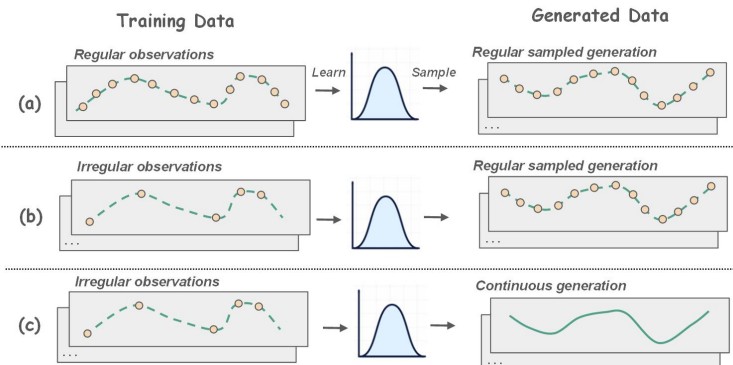

Figure 1: Different TSG tasks categorized by the data continuity and sampling irregularity: (a) regular to regular generation; (b) irregular to regular generation (c) irregular to continuous generation.

This makes NCDE-based methods especially suitable for continuous TS modeling and handling irregular observations. Some recent studies have leveraged NCDE-based methods for TSG (Lim et al., 2023; Naiman et al., 2024), typically following a two-stage paradigm: they first transform irregularly sampled TS into regularly sampled ones using NCDEs or their variants, and then apply standard TSG models such as GANs (Jeon et al., 2022) and VAEs (Naiman et al., 2024) to generate data on a regularly sampled TS.

However, **on the one hand**, they only focus on using NCDE to process irregular inputs and then generate regular time series (Fig. 1(b)), without in-depth research on how to generate and evaluate continuous time series (Fig. 1(c)), which still needs further exploration. **On the other hand**, existing methods simply apply the standard NCDE without making any improvements. However, the performance of the standard NCDE can still be further improved, which is helpful for dealing with irregular data and continuous generation tasks. We have noticed that the standard NCDE faces challenges in learning dynamically changing temporal patterns, especially in the generation of time series with different classes. Moreover, the standard NCDE involves the optimization of multiple components, and end-to-end training may cause the model to fail to focus on the learning of dynamic functions.

In this study, we propose a novel TSG framework, MN-TSG, which explores Mixture of Experts Neural Controlled Differential Equations (MoE-NCDE) and integrates it with existing TSG models for continuous time series generation. To address the aforementioned challenges, we introduce three key architectural innovations. First, we propose a Mixture of Experts (MoE) based dynamic functions for NCDE to effectively capture the diverse dynamics inherent in real-world time series data. Second, we decouple the model architecture to separately learn time series dynamics and the mapping between raw observations and hidden states by pretraining a channel-wise autoencoder. Third, we propose a joint learning approach that simultaneously models discrete observations and underlying MOE dynamics within the TSG framework. Our contributions are as follows:

- We propose the decoupled MOE Neural CDE, which improves the standard NCDE by proposing the mixture-of-experts dynamic functions (MOE dynamics) and a decoupled design to help NCDE better focus on the learning of MOE dynamics and handling dynamically changing patterns.

- We propose the continuous time series generation framework called MNTSG. It learns the joint distribution of dynamic functions and time series samples based on existing generative models. This allows the model to generate new samples while assigning them suitable dynamic function weights, leading to more reasonable continuous time series generation.

- We conduct comprehensive experiments with advanced baselines on four popular TSF datasets, four medical ECG datasets, and two synthetic datasets. In terms of quantitative and qualitative evaluations, results show that our design enhances NCDE's ability to learn dynamically changing patterns from irregular temporal data, achieving better performance in both irregular-to-regular and irregular-to-continuous generation tasks.

## 2 RELATED WORK

The most relevant methods are NCDE-based generators, e.g., KOVAE (Jeon et al., 2022) and GT-GAN (Naiman et al., 2024), which build on standard NCDE and conduct irregular-to-regular generations. Some approaches also attempt to use attention-style mechanisms to capture the relationship between the observed sparse values and the unknown target time-steps, learning the time dependencies and thereby enabling prediction at arbitrary time-steps (Shukla & Marlin; Yalavarthi et al., 2025). Unlike them, we improve the standard NCDE and propose the decoupled MOE NCDE. Moreover, we design a framework for continuous time series generation and explore its evaluations. Complete related work about TSG models for regular and irregular data is provided in the Appendix A.2.

## 3 METHOD

**Overview.** Figure 2 illustrates MNTSG. **Training:** MOE-NCDE is trained on irregular samples using mixture-of-experts dynamic functions to better capture dynamic patterns. Further, a pretrained channel-wise autoencoder replaces $z(0)$ and the reconstruction function with a frozen channel-wise encoder (CE) and frozen channel-wise decoder (CD) to make MOE-NCDE focus on learning MOE dynamics. After training, MOE-NCDE yields imputed regular samples and MOE weights of different dynamic functions, which are then fed into a diffusion model to join training and jointly generate new samples with corresponding expert weights. **Generation:** The generated samples and expert weights are fed back into MOE-NCDE for continuous generation, producing refined, longer time series. Finally, the right half compares traditional NCDE with our decoupled MOE-NCDE.

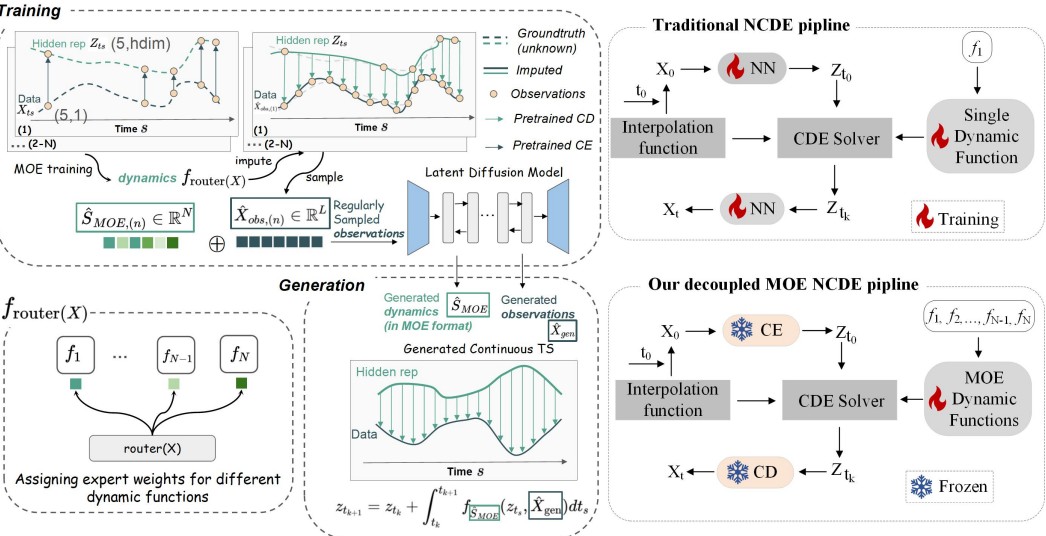

Figure 2: Overall structure of the proposed MNTSG.

**Notation.** Suppose we have $N$ observed multivariate time series. Each time series $\mathcal{X}^{(j)} : [0, T] \to \mathbb{R}^M$, for $j = 1, \ldots, N$, is a continuous multivariate function defined over its time interval, where $M$ is the number of channels/variables. In practice, we observe each multivariate time series at discrete and irregular time points $\{t_0^{(j)}, t_1^{(j)}, \ldots, t_{n_j}^{(j)}\} \subset [0, T]$, where $t_0^{(j)} < t_1^{(j)} < \cdots < t_{n_j}^{(j)}$. The observed dataset is:

$$\mathcal{D} = \left\{ \mathcal{O}^{(j)} = \left\{ (t_i^{(j)}, \mathbf{x}_i^{(j)}, \right\}_{i=0}^{n_j} \mid j = 1, \ldots, N \right\} \tag{1}$$

**Problem Definition.** Given dataset $\mathcal{D}$[1], the goal is to learn a generative model $p(\mathcal{O})$ over discrete and irregular time series for irregular or continuous generation, i.e., our goal is to learn a model

---

[1]We assume the underlying dynamic of the data is continuous and differentiable. Some discussion is made in Section 5.

capable of sampling new sequences $\tilde{\mathcal{X}}_{1:T} \sim p_\theta(\mathcal{X}_{1:T})$. The generated dataset is denoted as $\tilde{\mathcal{D}} = \left\{ \tilde{\mathcal{O}}^{(j)} = \left\{ (\tilde{t}^{(j)}, \tilde{\mathbf{x}}^{(j)}, \right\}^{\tilde{n}_j} \mid \tilde{t} \in [0, T] \right\}$ where $\tilde{t}^{(j)}$ denotes arbitrary time step of $j$-th sample.

## 3.1 Neural Controlled Differential Equations

Neural CDEs (Kidger et al., 2020) generalize Neural ODEs (Chen et al., 2018) by modeling the latent dynamics as being driven by a continuous input path (control path). The CDE is defined as:

$$z(t) = z(0) + \int_0^t f_\theta(z(s)) \, dX(s), \tag{2}$$

where $f_\theta : \mathbb{R}^d \to \mathbb{R}^d$ is a learnable neural dynamics function, and $X : [0, T] \to \mathbb{R}$ is a continuous interpolation (e.g., a natural cubic spline) of the discrete observations $\{(t_i, x_i)\}_{i=0}^n$. $z(0) \in \mathbb{R}^d$ is the initial latent state mapped from $\mathcal{X}_0$ based on a learnable neural network. $z(t)$ represents the hidden state at time step $t$, and a reconstruction function subsequently maps this hidden representation back to the original channel space to produce the predicted time-series values.

This interpolation defines the control path that drives the evolution of the latent state $z(s)$.

`CDESolver` is used to solve controlled differential equations, which can be formalized as:

$$\texttt{CDESolver}(f_\theta, z(0), [0, T], X) \to \{z(t)\}_{t \in [0, T]}, \tag{3}$$

## 3.2 Modeling Diverse Dynamics with Mixture of Experts

Real-world time series data often exhibit diverse dynamics across different samples, making it difficult for a single model to capture all behaviors effectively. By leveraging a Mixture of Experts (MoE) framework, we allow the model to assign specialized dynamics to different samples, enabling more expressive and personalized temporal modeling. This flexibility enhances both the representational capacity and the generalization of the learned dynamics.

The *MoE dynamics* are composed of a router $\mathcal{R} : \mathcal{O} \to \{0, 1, \ldots, N_e - 1\}$ and $N_e$ dynamic functions $f_\theta^{(i)}$ (each is parameterized as a neural network), where $i \in [0, N_e - 1]$. The router $\mathcal{R}$ takes (potentially irregular) observations as input and outputs a routing score for each expert. Specifically, we first use the cubic interpolation function to impute the irregular time series and then feed them into the router $\mathcal{R}$ as observations to obtain all experts' scores for the $j$-th sample. We implement the router by using simple MLPs. Then, a softmax function is used over the scores to compute a dense routing distribution. Specifically, given $\mathbf{r} \in \mathbb{R}^{N_e}$ for all experts' scores of $j$-th sample, the final dynamics function is then computed as a weighted combination of all experts' functions:

$$f_{\text{MoE}} = \sum_{i=0}^{N_e - 1} \mathbf{s}_i \cdot f_\theta^{(i)}, \quad \mathbf{s} = softmax(\mathbf{r}) \tag{4}$$

Finally, by substituting $f_{\text{MoE}}$ into standard NCDE formulas (Eq. 2 and Eq. 3), The Mixture-of-Experts-based MOE-NCDE is obtained.

## 3.3 Decoupled Training of MoE Dynamics

MOE-NCDE involves learning time series dynamic functions with a mixture of experts (*MOE dynamics*) and the mapping between raw observations and hidden states ($z(0)$ and reconstruction function). Existing methods often train all of them end-to-end, which can hinder MOE-dynamics training. Firstly, the quality of $z_0$ and the reconstruction function impact dynamic function learning. Secondly, jointly optimizing them prevents the model from focusing solely on the MOE dynamics.

To address MOE-NCDE training challenges, we propose a decoupled strategy: first pre-train $z(0)$ and the reconstruction function using a channel-wise autoencoder with two lightweight MLPs. The training pipeline is shown in Algorithm 4. Specifically, for each batch, we keep the channel dimension, flatten the rest, and train the encoder to obtain hidden states and the decoder to reconstruct the channel dimension. Then, MOE-NCDE is trained in the condition of $z(0)$ and the reconstruction

---

**Algorithm 1** Training dynamic functions with a mixture of experts (MOE dynamics)

---

**Require:** Observed dataset $\mathcal{D}$, an CDE solver `CDESolver`, $E$ training epochs.
1: Initialize $N$ parameterized dynamics experts $f_{\mathrm{MoE}}$.
2: Initialize an MOE router $\mathcal{R}$.
3: Pretrained channel-wise encoder $f_{CE}$ and decoder $f_{CD}$ with forzen weights.
4: Control path $X$ (Cubic Spline Interpolation Function in this work).
5: **for** $\mathcal{O}^{(j)}$ in $\mathcal{D}$, repeat $E$ epochs **do**
6:    Get all expert weights **s** for $j$-th sample  from $\mathcal{R}(\mathcal{O}^{(j)})$ using Softmax gating.
7:    $\hat{\mathcal{O}}^{(j)} = \mathtt{CDESolver}(f_{\mathbf{s}}, f_{CE}(X(t_0^{(j)})), \{t_0^{(j)}, t_1^{(j)}, \ldots, t_{n_j}^{(j)}\}, X)$
8:    Gradient backward on $\ell_{MSE}(f_{CD}(\hat{\mathcal{O}}^{(j)}), \mathcal{O}^{(j)})$ to update $f_{\mathrm{MoE}}$ (includes router $\mathcal{R}$) only.
9: **end for**

---

function are replaced by the frozen Channel-wise Encoder (CE) and Channel-wise Decoder (CD). This way allows the model to focus solely on the learning of MOE-dynamics. The MoE dynamics $f_{\mathrm{MoE}}$ can be trained through MSE loss based on `CDESolver` and the pretrained channel-wise autoencoder in the algorithm 1.

### 3.4 JOINT DIFFUSION MODELING OF OBSERVATIONS AND EXPERT DYNAMICS

While prior diffusion-based models focus on learning the distribution of discrete time series observations, i.e., $\mathcal{O}$, they often overlook the underlying dynamics. By extending the diffusion model $\mathcal{G}$ to jointly model both the observations $\mathcal{O}$ and the corresponding MoE weights $S$, we enable the diffusion model to generate not only observations but also appropriate expert weights $\hat{S}_{MOE}$ for modeling temporal dynamics.

Notably, the diffusion model prefers data with a fixed length while the observed dataset $\mathcal{D}$ might be irregular with changeable length. We leverage trained $f_{\mathrm{MoE}}$ to impute the irregular data $\mathcal{O}^{(i)}$ to a fixed length $\mathcal{O}_{\mathrm{reg}}^{(i)} \in \mathbb{R}$ firstly. Then, the generation model $\mathcal{G}$ that learn the joint distribution can be trained by the Appendix algorithm 2. In this way, we can obtain new samples and new expert weights $\hat{S}_{MOE}$. Finally, we substitute them into the CDE Solver, and for each generated sample, we perform continuous time series generation based on the respective generated expert weights. We show the process in the Appendix algorithm 3.

## 4 EXPERIMENTS

We conduct extensive experiments using multiple real-world and synthetic datasets. The experiment goal is to investigate on the following research questions: Q1: How effective is our method in handling missing data and generating regular data? (Section 4.2) Q2: Can our method better perform the task of continuous time series generation? (Section 4.3) Q3: Are the relevant designs of our MOE-NCDE truly effective (e.g., MOE dynamic functions and decoupled design)? (Section 4.4)

### 4.1 EXPERIMENTAL SETTINGS

Here we briefly outline the experimental settings, with details shown in the Appendix E.1.

**Datasets.** The datasets we used include: (1) four popular public time series generation datasets, namely Sines, Stocks, Energy, and MuJoCo. (2) Medical ECG-related time series classification datasets with a variety of different temporal patterns from the UCR archive, including ECG200, ECG5K, ECGFD, and TLECG. (3) Synthetic datasets, including integrated datasets of different signal waves and polynomial coefficient fitting datasets.

**Baselines.** We compare our method with several representative advanced time series generation methods, including KO-VAE (Naiman et al., 2024), GTGAN (Jeon et al., 2022), ProFITi Yalavarthi et al. (2025) and HeTVAE (Shukla & Marlin) tailored for irregular time series generation, as well as TimeGAN (Yoon et al., 2019), TimeVAE (Desai et al., 2021), and diffusion model (Huang et al., 2025) used for regular TSG.

Table 1: Irregular time series (30%, 50%, and 70% of observations are dropped) on popular TSG datasets. The performance is averaged over sequence lengths of 12, 24, and 36. The results on each length are shown in Appendix Tables 11, 12, and 13.

| | | 30% | | | | 50% | | | | 70% | | | |
|---|---|---|---|---|---|---|---|---|---|---|---|---|---|
| | | Sines | Stocks | Energy | MuJoCo | Sines | Stocks | Energy | MuJoCo | Sines | Stocks | Energy | MuJoCo |
| DS | Ours | **0.105** | **0.142** | **0.422** | **0.293** | **0.128** | **0.137** | **0.487** | **0.375** | **0.182** | **0.106** | **0.497** | **0.393** |
| | KoVAE | 0.142 | 0.225 | 0.476 | 0.372 | 0.171 | 0.187 | 0.500 | 0.397 | 0.228 | 0.213 | 0.500 | 0.425 |
| | GT-GAN | 0.302 | 0.321 | 0.499 | 0.489 | 0.416 | 0.326 | 0.500 | 0.494 | 0.341 | 0.292 | 0.499 | 0.492 |
| | TimeGAN-NCDE | 0.455 | 0.365 | 0.499 | 0.49 | 0.427 | 0.442 | 0.499 | 0.499 | 0.44 | 0.477 | 0.500 | 0.499 |
| | TimeVAE-NCDE | 0.234 | 0.470 | 0.498 | 0.375 | 0.308 | 0.485 | 0.496 | 0.463 | 0.426 | 0.492 | 0.498 | 0.476 |
| | Diffusion-NCDE | 0.200 | 0.441 | 0.460 | 0.355 | 0.330 | 0.475 | 0.489 | 0.427 | 0.421 | 0.491 | 0.497 | 0.473 |
| | ProFITi | 0.307 | 0.454 | 0.495 | 0.492 | 0.344 | 0.483 | 0.499 | 0.497 | 0.392 | 0.489 | 0.5 | 0.497 |
| | HeTVAE | 0.491 | 0.328 | 0.497 | 0.499 | 0.5 | 0.443 | 0.5 | 0.499 | 0.5 | 0.487 | 0.5 | 0.5 |
| MDD | Ours | **0.953** | **0.25** | 0.270 | 0.347 | **1.093** | **0.281** | 0.252 | **0.318** | **1.308** | **0.299** | 0.279 | **0.297** |
| | KoVAE | 5.134 | 0.709 | 0.347 | 0.374 | 5.584 | 0.662 | 0.368 | 0.419 | 6.393 | 0.624 | 0.377 | 0.619 |
| | GT-GAN | 2.644 | 0.630 | 0.567 | 0.558 | 2.978 | 0.589 | 0.473 | 0.596 | 2.559 | 0.603 | 0.532 | 0.627 |
| | TimeGAN-NCDE | 3.477 | 1.111 | 1.028 | 1.182 | 3.739 | 1.605 | 1.343 | 1.535 | 3.981 | 1.629 | 1.559 | 1.704 |
| | TimeVAE-NCDE | 4.767 | 0.597 | 0.444 | 0.338 | 3.761 | 0.726 | 0.533 | 0.443 | 3.397 | 0.843 | 0.604 | 0.565 |
| | Diffusion-NCDE | 1.462 | 0.341 | 0.292 | 0.371 | 1.498 | 0.476 | 0.351 | 0.37 | 1.997 | 0.743 | 0.513 | 0.487 |
| | ProFITi | 1.163 | 0.297 | **0.255** | **0.295** | 1.236 | 0.408 | 0.35 | 0.353 | 1.443 | 0.445 | 0.442 | 0.448 |
| | HeTVAE | 3.158 | 0.599 | 0.625 | 0.654 | 3.449 | 0.773 | 1.095 | 1.093 | 3.967 | 1.032 | 1.323 | 1.398 |
| KL | Ours | **0.013** | **0.074** | **0.020** | 0.021 | **0.023** | **0.094** | **0.022** | **0.009** | **0.033** | **0.091** | **0.017** | **0.014** |
| | KoVAE | 4.172 | 1.014 | 0.065 | 0.130 | 4.719 | 0.859 | 0.067 | 0.198 | 5.521 | 0.868 | 0.08 | 0.425 |
| | GT-GAN | 0.099 | 0.203 | 0.056 | 0.021 | 0.153 | 0.181 | 0.051 | 0.028 | 0.163 | 0.223 | 0.057 | 0.048 |
| | TimeGAN-NCDE | 4.963 | 1.742 | 0.259 | 0.560 | 2.954 | 6.776 | 0.585 | 1.153 | 12.478 | 11.63 | 0.995 | 1.810 |
| | TimeVAE-NCDE | 1.999 | 0.224 | 0.088 | 0.075 | 1.487 | 0.259 | 0.101 | 0.118 | 1.196 | 0.345 | 0.136 | 0.172 |
| | Diffusion-NCDE | 0.056 | 0.085 | 0.065 | 0.045 | 0.070 | 0.094 | 0.058 | 0.058 | 0.095 | 0.105 | 0.087 | 0.078 |
| | ProFITi | 0.079 | 0.074 | 0.022 | **0.015** | 0.112 | 0.121 | 0.033 | 0.015 | 0.138 | 0.132 | 0.045 | 0.031 |
| | HeTVAE | 1.178 | 1.085 | 0.248 | 0.258 | 4.236 | 1.4 | 0.917 | 0.996 | 2.488 | 1.612 | 1.578 | 2.202 |

**Implementation details.** We conduct experiments using the publicly available baseline code and train it to convergence with the recommended settings. In our method, the number of experts is fixed at four, and the standard diffusion model is adopted as the generation backbone. For the four public datasets and two synthetic datasets, we construct samples for training and evaluation using three different lengths: 12, 24, and 36, and finally report the average metrics. For the medical ECG classification datasets, each sample has already been well processed, and we use the default parameters. We randomly discard 30%, 50%, and 70% of the temporal observations along the time dimension to construct irregularly sampled datasets (Naiman et al., 2024).

**Evaluation metrics.** Following previous works (Naiman et al., 2024; Huang et al., 2025), we apply the following metrics for evaluating generation performance: (1) Discriminative score (DS), (2) Marginal distribution difference (MDD), (3) Kullback-Leibler divergence (KL).

## 4.2 Performance of Irregular Time Series Generation (TSG)

**Quantitative evaluation.** In this experiment, we assess our model's performance on irregular time series generation. We integrate regular time series generation methods with standard NCDE, using it to impute missing values before generation. Results in Tables 1 and 2 show that our method consistently outperforms baselines in both popular TSG and medical datasets. While baselines handle irregular data via NCDE, their performance is limited due to optimization challenges and difficulties in capturing dynamic patterns. In contrast, our MOE-NCDE uses a mixture of experts and a decoupled training strategy with a pre-trained channel-wise autoencoder to focus on learning MOE-dynamics. This makes our method better capture dynamic temporal patterns and more effective for irregular time series data.

**Qualitative evaluation.** We also qualitatively assess the similarity between real and generated sequences using two visualizations: (i) t-SNE projection into two dimensions, and (ii) kernel density estimation of their probability density functions (PDFs). As shown in Figure 3, in the t-SNE plots (top), our approach shows high overlap between real and synthetic samples, while our PDFs (bottom) display more similar trends than baselines KO-VAE and GT-GAN.

**Privacy protection analysis.** We adopt the Membership Inference Risk (MIR) metric (Tian et al., 2024) to assess whether synthetic data generated by various models reveals information about the real data, as shown in Appendix Table 10 due to limited space. While our method achieves superior fidelity in terms of DS, KL, and MDD, the MIR results also show that our model exhibits the lowest

Table 2: Irregular time series (30%, 50%, and 70% of observations are dropped) on medical datasets.

| | | 30% | | | | 50% | | | | 70% | | | |
|---|---|---|---|---|---|---|---|---|---|---|---|---|---|
| | | ECG200 | ECG5k | ECGFD | TLECG | ECG200 | ECG5k | ECGFD | TLECG | ECG200 | ECG5k | ECGFD | TLECG |
| **DS** | Ours | **0.188** | **0.182** | **0.120** | **0.240** | **0.250** | **0.326** | **0.070** | **0.160** | **0.390** | **0.424** | **0.140** | **0.340** |
| | KoVAE | 0.500 | 0.500 | 0.420 | 0.400 | 0.485 | 0.498 | 0.410 | 0.440 | 0.485 | 0.500 | 0.380 | 0.400 |
| | GT-GAN | 0.467 | 0.493 | 0.450 | 0.490 | 0.457 | 0.495 | 0.380 | 0.470 | 0.475 | 0.497 | 0.430 | 0.490 |
| | TimeGAN-NCDE | 0.495 | 0.500 | 0.410 | 0.500 | 0.482 | 0.478 | 0.420 | 0.500 | 0.490 | 0.500 | 0.420 | 0.490 |
| | TimeVAE-NCDE | 0.440 | 0.371 | 0.350 | 0.430 | 0.427 | 0.370 | 0.380 | 0.440 | 0.415 | 0.400 | 0.400 | 0.450 |
| | Diffusion-NCDE | 0.458 | 0.334 | 0.390 | 0.430 | 0.460 | 0.396 | 0.430 | 0.430 | 0.458 | 0.429 | 0.380 | 0.400 |
| | ProFITi | 0.48 | 0.496 | 0.3 | 0.45 | 0.462 | 0.497 | 0.38 | 0.45 | 0.48 | 0.498 | 0.36 | 0.46 |
| | HeTVAE | 0.5 | 0.496 | 0.42 | 0.5 | 0.5 | 0.498 | 0.43 | 0.5 | 0.5 | 0.5 | 0.43 | 0.5 |
| **MDD** | Ours | **0.25** | **0.09** | **0.85** | **1.016** | **0.247** | **0.109** | **0.794** | **0.972** | **0.253** | **0.108** | **0.833** | **1.064** |
| | KoVAE | 0.551 | 0.364 | 1.588 | 1.467 | 0.565 | 0.359 | 1.581 | 1.439 | 0.523 | 0.357 | 1.604 | 1.475 |
| | GT-GAN | 0.495 | 0.409 | 0.97 | 1.158 | 0.506 | 0.437 | 0.968 | 1.091 | 0.556 | 0.296 | 0.99 | 1.177 |
| | TimeGAN-NCDE | 0.807 | 0.592 | 1.609 | 1.656 | 0.804 | 0.573 | 1.683 | 1.679 | 0.806 | 0.558 | 1.608 | 1.681 |
| | TimeVAE-NCDE | 0.417 | 0.173 | 1.026 | 1.136 | 0.37 | 0.175 | 0.970 | 1.265 | 0.363 | 0.154 | 0.939 | 1.131 |
| | Diffusion-NCDE | 0.389 | 0.187 | 0.945 | 1.125 | 0.369 | 0.190 | 0.973 | 1.082 | 0.328 | 0.198 | 0.958 | 1.130 |
| | ProFITi | 0.31 | 0.185 | 0.989 | 1.074 | 0.298 | 0.196 | 1.006 | 1.115 | 0.325 | 0.247 | 1.003 | 1.138 |
| | HeTVAE | 0.807 | 0.492 | 1.51 | 1.608 | 0.806 | 0.461 | 1.538 | 1.576 | 0.81 | 0.569 | 1.532 | 1.594 |
| **KL** | Ours | **0.053** | **0.014** | **0.052** | **0.129** | **0.09** | **0.021** | **0.099** | **0.073** | **0.113** | **0.018** | **0.04** | **0.153** |
| | KoVAE | 8.645 | 7.584 | 10.158 | 9.482 | 8.693 | 7.537 | 10.112 | 9.440 | 8.692 | 7.537 | 10.144 | 9.477 |
| | GT-GAN | 5.129 | 3.305 | 2.004 | 5.314 | 5.013 | 3.592 | 2.543 | 7.912 | 7.126 | 2.672 | 3.754 | 12.539 |
| | TimeGAN-NCDE | 17.57 | 16.615 | 13.165 | 17.182 | 17.56 | 14.408 | 13.154 | 17.191 | 17.57 | 9.431 | 13.165 | 17.191 |
| | TimeVAE-NCDE | 0.917 | 0.113 | 2.236 | 2.397 | 0.641 | 0.096 | 2.106 | 3.005 | 0.731 | 0.073 | 2.103 | 1.386 |
| | Diffusion-NCDE | 0.639 | 0.095 | 1.867 | 0.788 | 0.87 | 0.125 | 1.156 | 0.632 | 0.365 | 0.136 | 1.117 | 1.336 |
| | ProFITi | 0.085 | 0.053 | 0.204 | 0.245 | 0.073 | 0.044 | 0.262 | 0.346 | 0.149 | 0.066 | 0.395 | 1.188 |
| | HeTVAE | 17.216 | 8.574 | 8.218 | 16.49 | 17.199 | 6.251 | 13.445 | 16.41 | 17.193 | 10.316 | 13.445 | 16.458 |

privacy leakage risk among all methods. This indicates that our jointly generated dynamics and data not only improve quality but also show lower privacy leakage risk.

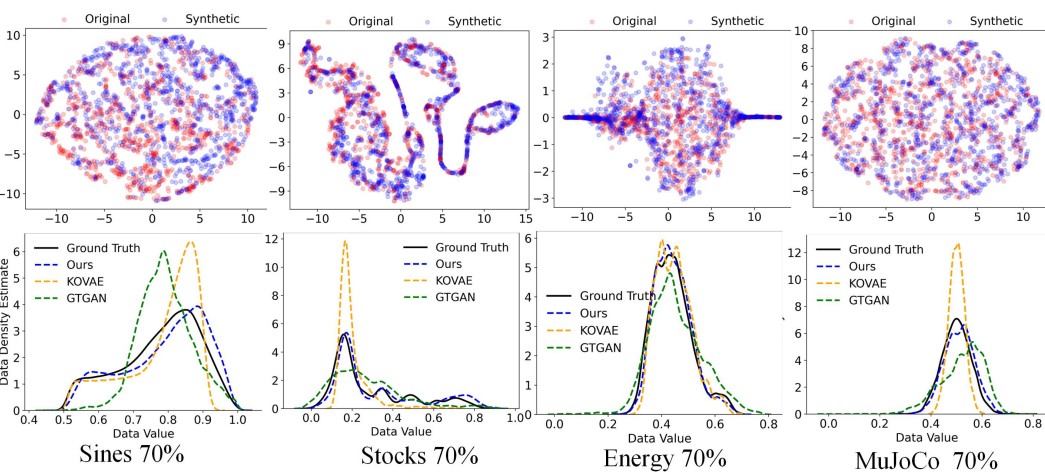

Sines 70%    Stocks 70%    Energy 70%    MuJoCo 70%

Figure 3: We qualitatively evaluate our method with 2D t-SNE plots (top) and probability density functions of real, KoVAE, and GT-GAN data (bottom). Additional cases are in Appendix Figure 27.

## 4.3 PERFORMANCE OF CONTINUOUS TIME SERIES GENERATION (TSG)

**Refined continuous generation on the downstream task.** In this experiment, we evaluate the benefits of continuous TSG with irregular observations. Specifically, we begin by defining a multivariate prediction task. Baselines use regular generation with length $t-1$ data to predict the last time step. Then, we apply NCDE for continuous generation, in which we generate new data points in the middle of every two time steps and insert them into the original window. As a result, the length of the historical window is approximately doubled, called *refined sample*. Finally, we further use the refined sample to predict the same last time step to compare the performance with baselines. We set $t = 12, 24, 36$ and report average MSE and MAE. Since KOVAE and GTGAN are not designed for continuous generation, we use their trained NCDE to interpolate generated samples for refinement.

Table 3 shows that the performance of KOVAE-NCDE and GTGAN-NCDE is even worse than that of their non-continuous generation counterparts KOVAE and GTGAN, indicating that the method of

Table 3: Continuous generation of MOE-NCDE on irregular time series (30%, 50%, 70% missing), averaged over lengths 12, 24, and 36. "Ours-$\hat{S}_{MOE}$" further applies generated experts and series for refined continuous TSG with MOE-NCDE, doubling sequence length. Detailed results are in Appendix Tables 14–16, with visualizations in Figures 17–23.

| | | 30% | | | | 50% | | | | 70% | | | |
|---|---|---|---|---|---|---|---|---|---|---|---|---|---|
| | | Sines | Stocks | Energy | MuJoCo | Sines | Stocks | Energy | MuJoCo | Sines | Stocks | Energy | MuJoCo |
| MSE | Ours-$\hat{S}_{MOE}$ | **0.026** | **0.002** | **0.013** | **0.029** | **0.026** | **0.002** | **0.013** | **0.029** | **0.028** | **0.002** | **0.013** | **0.029** |
| | Ours | 0.049 | 0.005 | 0.014 | 0.031 | 0.05 | 0.005 | 0.014 | 0.030 | 0.054 | 0.005 | 0.014 | 0.030 |
| | KOVAE-NCDE | 0.076 | 0.041 | 0.023 | 0.030 | 0.076 | 0.036 | 0.021 | 0.032 | 0.079 | 0.0300 | 0.020 | 0.033 |
| | KOVAE | 0.044 | 0.025 | 0.014 | 0.030 | 0.044 | 0.019 | 0.014 | 0.031 | 0.045 | 0.014 | 0.014 | 0.032 |
| | GTGAN-NCDE | 0.089 | 0.021 | 0.027 | 0.053 | 0.076 | 0.018 | 0.026 | 0.053 | 0.070 | 0.021 | 0.028 | 0.058 |
| | GTGAN | 0.044 | 0.014 | 0.019 | 0.042 | 0.051 | 0.008 | 0.017 | 0.047 | 0.046 | 0.011 | 0.019 | 0.047 |
| MAE | Ours-$\hat{S}_{MOE}$ | **0.125** | **0.025** | **0.075** | **0.129** | **0.126** | **0.026** | **0.073** | **0.131** | **0.128** | **0.025** | **0.075** | **0.131** |
| | Ours | 0.174 | 0.036 | 0.075 | 0.137 | 0.176 | 0.036 | 0.074 | 0.136 | 0.182 | 0.036 | 0.076 | 0.136 |
| | KOVAE-NCDE | 0.199 | 0.132 | 0.114 | 0.139 | 0.200 | 0.123 | 0.110 | 0.142 | 0.204 | 0.116 | 0.105 | 0.146 |
| | KOVAE | 0.170 | 0.094 | 0.080 | 0.139 | 0.172 | 0.076 | 0.080 | 0.140 | 0.175 | 0.061 | 0.080 | 0.144 |
| | GTGAN-NCDE | 0.216 | 0.106 | 0.116 | 0.183 | 0.209 | 0.100 | 0.122 | 0.186 | 0.199 | 0.108 | 0.123 | 0.193 |
| | GTGAN | 0.176 | 0.069 | 0.093 | 0.160 | 0.186 | 0.056 | 0.089 | 0.174 | 0.174 | 0.057 | 0.099 | 0.173 |

using standard NCDE for continuous generation is not effective. In contrast, Ours-$\hat{S}_{MOE}$ is better than its non-continuous version, improving the forecasting performance. This is attributed to our carefully designed MOE-NCDE and the joint training process, which not only generates new samples but also produces MOE-weights tailored for each sample, thereby enabling better continuous generation tasks using the MOE-NCDE. The visualization analysis of why continuous time series is effective can be found in Appendix F.2.

**Refined continuous generation on solving for the analytical solutions of polynomial functions.** We further assess continuous generation via analytical solutions using a cubic function $y = ax^3 + bx^2 + cx + d$. For each coefficient $a, b, c, d$, 2500 samples are drawn from the normal distributions with different means and standard deviations. With $x$ linearly spaced over 24 values in $[-1, 1]$, we compute $y$ to form time series (visualized in Appendix figure 28). Then, the model is trained on $y$, aiming for coefficients $\hat{a}, \hat{b}, \hat{c}, \hat{d}$ recalculated from generated $\hat{y}$ to match the original distributions.

Figure 4 shows that analytical solutions from our generated series closely match the true distribution. Both irregular to regular generation and irregular to continuous generation preserve distributional similarity, with refined continuous generation (Fig. 4(d)) closer to ground truth than non-continuous. This confirms the effectiveness of our approach, whereas baselines relying on standard NCDE deviate significantly (Appendix Figures 29, 30 and 31).

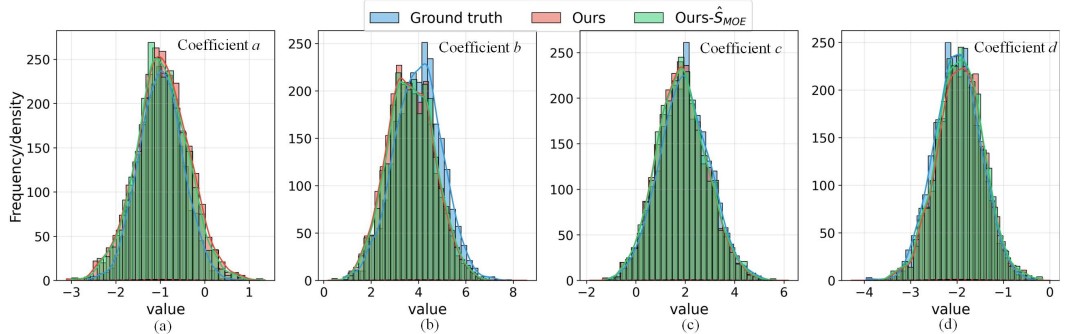

Figure 4: Analysis of cubic polynomial solutions: "Ours" denotes regular sample generation from irregular observations, while "Ours-$\hat{S}_{MOE}$" uses generated MOE weights for continuous TSG (doubling length) before computing solutions. Additional cases are in Appendix Figures 28–31.

## 4.4 ABLATION STUDY

Based on the experiments of imputing irregular time series data, we demonstrate the effectiveness of each component of MOE-NCDE from both quantitative and qualitative perspectives.

**Quantitative evaluation.** Table 4 shows that removing MOE dynamics and the decoupled design, the model's accuracy decreased to varying degrees. When both of them are removed, the perfor-

Table 4: Ablation study of MOE-Neural CDE on Irregular time series (30%, 50%, and 70% of observations are dropped) on medical ECG datasets.

| | | 30% | | | 50% | | | 70% | | |
| | | ECG200 | ECG5K | ECGFD | ECG200 | ECG5K | ECGFD | ECG200 | ECG5K | ECGFD |
|---|---|---|---|---|---|---|---|---|---|---|
| MDD | Ours | **0.211** | **0.062** | **0.424** | **0.237** | **0.079** | **0.507** | 0.246 | **0.095** | **0.624** |
| | w/o MOE | 0.226 | 0.152 | 0.740 | 0.255 | 0.179 | 0.778 | **0.244** | 0.196 | 0.784 |
| | w/o Decoupled | 0.370 | 0.191 | 0.878 | 0.397 | 0.174 | 0.850 | 0.361 | 0.154 | 0.865 |
| | w/o MOE-Decoup. | 0.412 | 0.173 | 0.965 | 0.363 | 0.175 | 0.968 | 0.361 | 0.157 | 0.984 |
| KL | Ours | **0.052** | **0.009** | **0.055** | **0.107** | **0.014** | **0.037** | 0.098 | **0.02** | **0.037** |
| | w/o MOE | 0.097 | 0.091 | 0.068 | 0.111 | 0.13 | 0.037 | **0.082** | 0.155 | 0.076 |
| | w/o Decoupled | 0.466 | 0.094 | 0.035 | 0.502 | 0.081 | 0.028 | 0.836 | 0.083 | 0.034 |
| | w/o MOE-Decoup. | 0.877 | 0.109 | 1.203 | 0.583 | 0.096 | 1.211 | 0.667 | 0.074 | 1.323 |

Table 5: Comparison between four experts (each with 2 linear layers) and a single expert with stacked 8 linear layers on medical datasets. Former and latter have the same number of parameters.

| | | 30% | | | 50% | | | 70% | | |
| | | ECG200 | ECG5k | ECGFD | ECG200 | ECG5k | ECGFD | ECG200 | ECG5k | ECGFD |
|---|---|---|---|---|---|---|---|---|---|---|
| DS | Four-experts-eight-layers | **0.2** | **0.205** | **0.09** | **0.405** | **0.315** | **0.12** | **0.34** | **0.288** | **0.13** |
| | One-expert-eight-layers | 0.328 | 0.364 | 0.13 | 0.44 | 0.389 | 0.23 | 0.392 | 0.412 | 0.21 |
| MDD | Four-experts-eight-layers | **0.211** | **0.062** | **0.424** | **0.237** | **0.079** | **0.529** | **0.246** | **0.095** | **0.613** |
| | One-expert-eight-layers | 0.288 | 0.12 | 0.694 | 0.321 | 0.152 | 0.831 | 0.314 | 0.165 | 0.837 |
| KL | Four-experts-eight-layers | **0.052** | **0.009** | **0.055** | **0.107** | **0.014** | **0.05** | **0.098** | **0.02** | **0.049** |
| | One-expert-eight-layers | 0.136 | 0.056 | 0.065 | 0.236 | 0.098 | 0.189 | 0.224 | 0.123 | 0.165 |

mance further deteriorates, indicating that the proposed MOE dynamics and decoupled design are crucial components. The former helps NCDE learn dynamic patterns, while the latter decouples the training of MOE-NCDE by using a pre-trained channel-wise autoencoder to replace the $z(0)$ and the reconstruction function in NCDE, thereby better focusing on the learning of the MOE-dynamics.

**Qualitative evaluation.** We construct synthetic signals (e.g., sine, piecewise, sawtooth) and divide them into training and test sets. We train the MOE-NCDE and observe its generation effect on the test set under component ablations (Fig. 5). Only using both MOE dynamics and decoupled design does the model produce data resembling the real signals, while removing either causes high errors, confirming their effectiveness in our decoupled MOE Neural CDE.

**Comparisons between Mixture-of-experts and single-expert parameter stacking.** Additional experiments confirm the benefit of cooperation among multiple experts. Table 5, compares NCDE performance between (i) four experts (each with one MLP and two linear layers, eight layers in total) and (ii) a single expert with four MLPs stacked (same eight layers). The mixture-of-experts approach significantly outperforms the single expert with stacked layers, demonstrating that having mixture-of-experts enables better learning of diverse temporal dynamics.

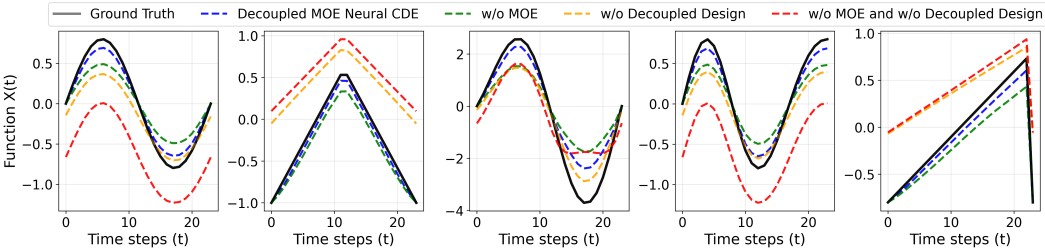

Figure 5: Qualitative analysis of MOE-NCDE ablation study on different synthetic signals.

**Visualizing specific temporal dynamics learned by MoE-NCDE.** Figure 6 demonstrates the strategy of decomposing complex temporal dynamics for separate learning and then integrating them, which substantially enhances NCDE's capability to model diverse temporal behaviors.

### 4.5 HYPERPARAMETER ANALYSIS AND TIME COMPLEXITY ANALYSIS

**Hyperparameter Analysis.** Figure 7 shows that MOE-NCDE performance is not highly sensitive to the number of experts: using more than one can both improve performance, with four experts

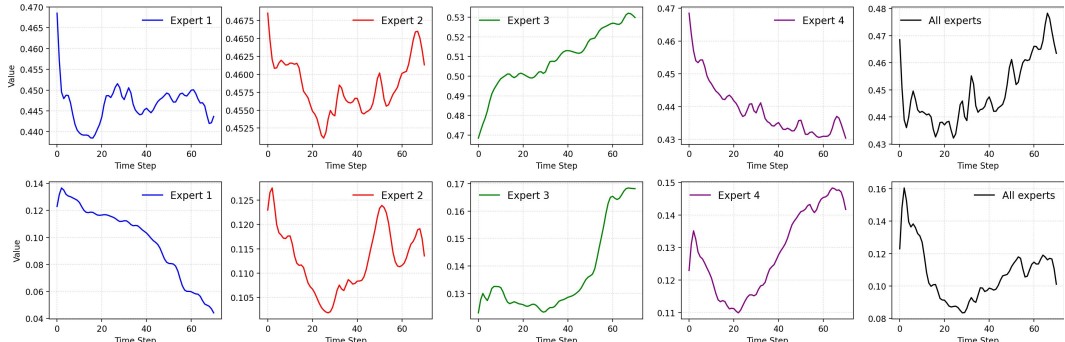

Figure 6: Visualizing the specific temporal dynamics learned by MoE-NCDE on the **MuJoCo** dataset in the continuous generation task. We observe that the temporal patterns learned by different experts exhibit local or global diversity, while the combined patterns produced by all experts retain characteristics of those learned individually.The cases on the energy dataset are shown in Appendix Figure 10.

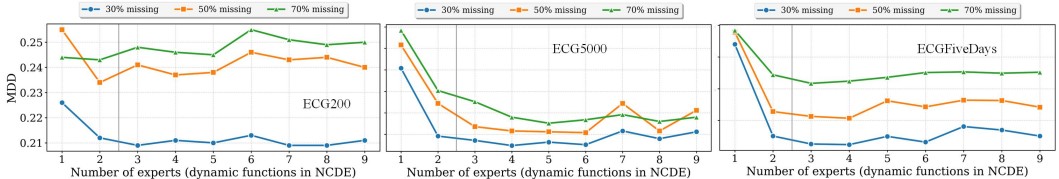

Figure 7: The impact of the number of experts (1 to 9 in the x-axis) on the MDD metric. The result on the DS metric is shown in Appendix Figure 26.

yielding satisfactory performance. Appendix Figure 26 shows the analysis on the DS metric, which also draws a similar conclusion. However, the default four experts may not be the optimal parameter, indicating that there is still room for further improvement in our metrics.

**Time Complexity Analysis.** Naturally, MOE-NCDE has higher time complexity than standard NCDE (e.g., 1.2s vs. 0.81s for double-length refined generation on 3,674 stock samples of length 12) due to mixture-of-experts dynamic functions. However, since we learn the joint distribution of the experts' weights and time series samples, the model not only generates samples but also assigns appropriate weights to them simultaneously. This avoids the need to re-extract features and perform forward propagation to compute weights, thereby reducing computational overhead.

## 5 CONCLUSION AND LIMITATIONS

This work investigates the task of continuous time series generation. On the one hand, to enhance NCDE's ability to impute data at arbitrary time intervals, we propose a mixture-of-experts dynamic function and a decoupled design, called Decoupled MOE Neural CDE. This helps NCDE better focus on learning the dynamic functions, thereby enabling it to subsequently better handle irregular data and continuous generation tasks. On the other hand, we offer new insights into the continuous generation task by proposing to learn the joint distribution of dynamic functions and time series samples. This allows the model to generate new samples while assigning them appropriate dynamic function weights for better continuous time series generation. Extensive experiments on ten datasets show that our method outperforms advanced baselines in both quantitative and qualitative evaluations for different generation tasks.

**Limitations.** We have an assumption that the underlying dynamics are continuous and differentiable, which lies in the requirements of Neural CDE. This smoothness assumption enables the use of differential equation solvers for trajectory modeling but restricts their applicability in certain real-world scenarios where dynamics often involve abrupt changes, discontinuities, or non-differentiable behaviors. In future works, we may examine potentials to resolve or relax this assumption.

# 6 ETHICS STATEMENT

This work adheres to the ICLR Code of Ethics. In this study, no human subjects or animal experimentation were involved. All datasets used in this paper were sourced in compliance with relevant usage guidelines, ensuring no violation of privacy. We have taken care to avoid any biases or discriminatory outcomes in our research process. No personally identifiable information was used, and no experiments were conducted that could raise privacy or security concerns. We are committed to maintaining transparency and integrity throughout the research process.

# 7 REPRODUCIBILITY STATEMENT

To ensure the reproducibility of our algorithms and experimental results, we provide a code link at ABSTRACT for downloading and viewing the algorithm and experimental details. The hyperparameters and datasets used in the experiments are also included in the code link to facilitate easier reproduction of the paper's experimental results.

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

# A   APPENDIX

## A.1   THE USE OF LARGE LANGUAGE MODELS (LLMS)

Large Language Models (LLMs) were used to aid in the writing and polishing of the manuscript. Specifically, we used an LLM to assist in refining the language, improving readability, and ensuring clarity in various sections of the paper. The model helped with tasks such as sentence rephrasing and enhancing the overall flow of the text.

It is important to note that the LLM was not involved in the ideation, research methodology, or experimental design. All research concepts, ideas, and analyses were developed and conducted by the authors. The contributions of the LLM were solely focused on improving the linguistic quality of the paper, with no involvement in the scientific content or data analysis.

The authors take full responsibility for the content of the manuscript, including any text generated or polished by the LLM. We have ensured that the LLM-generated text adheres to ethical guidelines and does not contribute to plagiarism or scientific misconduct.

---

**Algorithm 2** Joint Diffusion Training for $(\mathcal{O}_{\mathrm{reg}}, s)$

---

**Require:** Dataset $\mathcal{D}_{\mathrm{reg}}$, hyperparameter diffusion time step $T_d$ and the learned MOE weights $s$ for each sample.

1: **while** not converged **do**
2:     Sample $x_0 = (\mathcal{O}_{\mathrm{reg}}, s)$ from dataset $\mathcal{D}_{\mathrm{reg}}$
3:     Randomly sample time step $n \sim \mathcal{U}(1, T_d)$
4:     Randomly sample noise $\epsilon \sim \mathcal{N}(0, I)$
5:     Corrupt data $x_t = \sqrt{\bar{\alpha}_t} \cdot x_0 + \sqrt{1 - \bar{\alpha}_t} \cdot \epsilon$
6:     Predict noise: $\hat{\epsilon} = \epsilon_\theta(x_t, t)$
7:     Compute loss: $\mathcal{L} \leftarrow \|\epsilon - \hat{\epsilon}\|^2$
8:     Update $\theta$
9: **end while**

---

**Algorithm 3** Continuous Time Series Generation.

---

**Require:** Observed dataset $\mathcal{D}$, an CDE solver `CDESolver`

1: Train a $f_{\mathrm{MoE}}$ with Algorithm 1.
2: Impute the observed dataset $\mathcal{D}$ to $\mathcal{D}_{\mathrm{reg}}$ with $f_{\mathrm{MoE}}$.
3: Train a diffusion model $\mathcal{G}$ with Algorithm 2.
4: Generate $\hat{s}, \hat{\mathcal{O}}$ using diffusion model $\mathcal{G}$.
5: $\hat{\mathcal{O}}^{(i)} = \texttt{CDESolver}(f_{\hat{s}_i}, f_{CE}(\hat{X}(t_0^{(i)})), \{t_0^{(i)}, t_1^{(i)}, \ldots, t_{n_i}^{(i)}\}, \hat{X})$
6: **return** $f_{CD}(\mathcal{O}^{(i)})$

---

## A.2   RELATED WORK

**Time Series Generation for regular Data.**   Time series generation has gained increasing attention as a way to produce synthetic data for training, privacy preservation, and simulation. Early work explored adversarial learning approaches such as TimeGAN (Yoon et al., 2019), which leverage adversarial objectives to jointly capture temporal dynamics and sample realism. Variational methods have also been developed, including TimeVAE (Desai et al., 2021) and Vector Quantized TimeVAE (Lee et al., 2023). More recently, diffusion-based models (Lin et al., 2024) have emerged, synthesizing high-fidelity sequences by iteratively denoising random noise. These approaches illustrate the rapid methodological progress in generative modeling for time series.

**Time Series Modeling with Irregular Data.**   Irregularly sampled data becomes a critical challenge in time series modeling. An important line of work introduces continuous-time dynamics to naturally handle uneven sampling, such as Neural ODEs (Chen et al., 2018) and their extensions, which enrich the dynamics with additional inputs like the slope of imputed observations (Kidger et al., 2020), also called Neural CDE. Other approaches directly encode timestamp information into

sequence models, e.g., using specialized irregularity-aware architectures (Zhang et al., 2024). However, it cannot model the temporal patterns at arbitrary time points and cannot be directly applied to continuous generation tasks. Collectively, these methods underscore the importance of explicitly modeling temporal irregularity as a prerequisite for generating reliable and useful sequences.

Based on Neural CDE, several methods for handling irregular-to-regular data generation have been proposed, such as those based on VAE (Naiman et al., 2024) and GAN (Jeon et al., 2022). However, these methods haven't explored how to improve NCDE, and haven't investigated how to perform and evaluate continuous time series generation tasks. In contrast, this paper conducts an in-depth study and discussion on these aspects.

## B    CHANNEL-WISE AUTOENCODER AND ITS CONNECTION TO NCDE

**Explanation of pre-training part**: We pre-train an MLP-based encoder ($f_{CE}$)–decoder ($f_{CD}$) pair on the original irregular data by performing channel-wise encoding and reconstruction. All non-channel dimensions are flattened, and the model is trained via MSE loss to minimize the reconstruction error. Since the input and output dimensions of $f_{CE}$ and $f_{CD}$ match the number of time-series variables (channels), they can be directly used to map hidden states within NCDE. The full channel-wise autoencoder procedure is now given as Algorithm 4 in the revised manuscript.

**Explanation of z(0)**: The latent initial state $z(0)$ represents the encoded hidden representation of the time series at $t_0$. The latent trajectory follows the controlled differential equation (Eq. 2). To obtain $z_0 \in \mathbb{R}^{BatchSize \times HiddenDim}$, we take the observed values $x_0 \in \mathbb{R}^{BatchSize \times Channels}$ at $t = 0$ from $X$ and map them into the latent space using the pre-trained encoder $f_{CE}$.

**Explanation of reconstruction function**: After solving the NCDE, we obtain latent trajectories, $z_T \in \mathbb{R}^{BatchSize \times SeqLen \times HiddenDim}$. To recover the projected variables, we apply the decoder $f_{CD}$, yielding $f_{CD}(z_T)$, which serves as the reconstruction function.

---

**Algorithm 4** Training channel-wise autoencoder

---

**Require:** Observed dataset $\mathcal{D}$, Control path $X$ (Cubic Spline Interpolation Function), *isnan* function and $E$ training epochs.
1: Initialize parameterized channel-wise MLP based encoder $f_{CE}$ and decoder $f_{CD}$.
2: **for** $\mathcal{O}^{(i)}$ in $\mathcal{D}$, repeat $E$ epochs **do**
3:     Get $m_i$ from $\mathcal{R}(\mathcal{O}^{(i)})$ using *isnan* function.
4:     Get $\mathcal{O}^{(i)}_{\text{filled}}$ using $X$ to fill missing values in $\mathcal{O}^{(i)}$.
5:     Get $\widetilde{\mathcal{O}}^{(i)}_{\text{filled}}$ by flattening $\mathcal{O}_{\text{filled}}$ from $[B, T, D]$ to $[B \times T, D]$ for channel-wise reconstruction.
6:     Get $\widetilde{m}_i$ by flattening $\mathcal{O}_{\text{filled}}$ from $[B, T, D]$ to $[B \times T, D]$
7:     **Encode and decode at channel-wise:** $h = f_{CE}(\mathcal{O}^i_{\text{filled}})$, $\mathcal{O}^i_{\text{recon}} = f_{CD}(h)$
8:     **Compute Loss** using non-missing values.
        $\mathcal{L} = \ell_{\text{MSE}}(\mathcal{O}^i_{\text{recon}}[\neg\text{mask}], \mathcal{O}^i[\neg\text{mask}])$.
9: **end for**

---

## C    PRELIMINARY

**Neural ordinary differential equations**    Neural ODEs (Chen et al., 2018) provide a continuous-time modeling framework particularly suitable for irregularly-sampled time series. The latent dynamics are governed by the following differential equation:

$$z(t) = z(0) + \int_0^t f_\theta(z(s)) \, ds, \tag{5}$$

where $z(s) \in \mathbb{R}^d$ is the latent state at time $s$, and $f_\theta : \mathbb{R}^d \to \mathbb{R}^d$ is a neural network parameterized by $\theta$ represent the dynamics of the differential equations. The latent state can be initialized as $z(0)$. $(\cdot) : \mathbb{R} \to \mathbb{R}^d$ is an encoder that maps the observed time series $\mathcal{O}$ to the latent space. A decoder $d(\cdot) : \mathbb{R}^d \to \mathbb{R}$ can be used to map the latent trajectory $z(t)$ back to the data space to reconstruct or generate the continuous signal.

## D    MORE DISCUSSIONS ABOUT CONTRIBUTIONS

The core novelty of our method lies not in "simple combination" but in synergistic innovations tailored to the unaddressed challenge of irregular-to-continuous time series generation, which requires rethinking how these components interact to solve domain-specific pain points.

First, our adaptation of MoE to NCDE is novel in both purpose and mechanism. Unlike conventional MoE applications in NLP or computer vision, we explicitly design the MoE dynamics to capture heterogeneous temporal behaviors (e.g., diverse ECG rhythms across patients with different diseases) that standard NCDEs struggle to model. This is enabled by a task-specific router that maps irregular time-series characteristics—such as missing-value patterns and local volatility—into expert weights, enabling dynamic specialization across diverse temporal dynamics. Existing MoE or NCDE research has not explored this direction for time series generation. Furthermore, MoE allows us to decouple the modeling of temporal dynamics into expert-weight vectors that can be learned independently and generated through diffusion, ensuring that every newly generated sample receives a tailored dynamic function.

Second, we introduce a decoupled training strategy that addresses a known limitation of NCDE optimization. Prior NCDE-based models optimize both the initial latent mapping $z(0)$ and the dynamic function jointly, often leading to optimization conflicts (e.g., reconstruction dominating dynamics learning). Our pre-trained channel-wise autoencoder provides a clean and stable initialization for $z(0)$ and reconstruction, allowing NCDE to focus specifically on learning the underlying temporal dynamics. This design is essential to making MoE-NCDE not only feasible but also significantly more effective.

Finally, we extend diffusion models beyond standard TSG generation. Existing diffusion-based TSG works only model the distribution of observed values. We propose a joint-generation mechanism that simultaneously generates observations and MoE weights, ensuring that each synthesized time series is coupled with its own specialized dynamic function for downstream continuous refinement. This closes the methodological gap between "regular sequence generation" and "continuous generation," a capability not previously addressed in the literature.

In summary, we propose the irregular-to-regular-to-continuous generation framework, MNTSG, in which we improve NCDE so it can be seamlessly coupled with any state-of-the-art regular time-series generator to learn their joint distribution. Afterwards, MNTSG injects the newly sampled NCDE-MOE weights into the pre-trained NCDE, enabling fine-grained and continuous generation for each newly generated TS data.

## E    ADDITIONAL EXPERIMENTAL SETTINGS

### E.1    DATASETS

We evaluate on four popular TSG datasets with diverse characteristics, following established benchmarks for generative time series (Yoon et al., 2019; Jeon et al., 2022). Sines is a simulated multivariate dataset where each channel follows a sinusoidal function with random frequency and phase, capturing continuous and periodic patterns. Stocks contains daily Google stock data (2004–2019) across six financial indicators, exhibiting aperiodic, random-walk behaviors. Energy is the UCI appliance energy prediction dataset (Candanedo, 2017), comprising 28 correlated channels with noisy periodicity and continuous-valued measurements. Finally, MuJoCo (**Mu**lti-**Jo**int dynamics with **Co**ntact) (Todorov et al., 2012) provides simulated physical dynamics with 14 channels.

The used medical ECG datasets are ECG200, ECG5000 (ECG5K), ECGFiveDays (ECGFD), and TwoLeadECG (TLECG), which can be downloaded from the link[2]. Their sequence lengths are 96, 140, 136, and 82. The number of classes is 2, 5, 2, and 2, respectively. The data sizes are 200, 5000, 884, and 1162, respectively.

---

[2]`www.timeseriesclassification.com`

## E.2 IMPLEMENTATION DETAILS

For the TSG backbone, we follow the standard diffusion model architecture. The denoising network adopts a U-Net with four down- and up-sampling stages. Each stage contains two residual blocks and one cross-attention block, where residual blocks use two 1D convolution layers, and cross-attention employs 1D convolutions for input/output projections with eight attention heads. A middle block, placed between the down- and up-sampling paths, consists of two residual blocks and one attention block. In addition, we include input and output layers, each implemented with a single 1D convolution. SiLU is used as the activation function throughout. We train the model for 600 epochs with a batch size of 256, a learning rate of $1 \times 10^{-4}$. More details can be found in our code link[3].

## F ADDITIONAL EXPERIMENTAL RESULTS

### F.1 FURTHER DISCUSSIONS ON MOE WEIGHTS

**Regarding positivity of the generated weights.** During training, all expert weights are positive. In practice, the diffusion model learns this distribution well, and we have not observed negative weights in the dataset. As shown in Table 6, the minimum expert weight across all samples and datasets remains strictly above zero.

**Regarding the sum-to-one property of the generated weights.** We do not enforce a hard constraint to force the generated weights to sum to 1. Instead, we allow the diffusion model to learn this property implicitly—a form of adaptive soft constraint. As reported in Table 7, the sum of the generated weights is extremely close to 1 across all datasets, indicating that the model naturally preserves this property without explicit enforcement. Moreover, Table 9 compares forecasting performance using (i) raw generated weights and (ii) post-normalized weights that sum to 1. The results show almost no difference between the two settings. This suggests that the learned joint distribution already produces weight vectors that are appropriate for each sample. Finally, Figure 8 and Figure 9 visualize the effect of normalizing the weights. The qualitative results are fully consistent with the above quantitative findings: whether or not the weights are normalized to sum to 1 has no noticeable impact on the generated trajectories.

**Average weight of single expert.** Table 8 reports the average weight assigned to each expert across all samples in different datasets. The weights are not sparse, and different datasets both tend to prefer different experts, demonstrating that the expert structure is indeed utilized and optimized in a meaningful manner rather than collapsing to a single expert.

Table 6: Global minimum of the individual expert weights generated by the diffusion model across all samples and datasets.

|  | Sines | Stocks | Energy | MuJoCo |
|---|---|---|---|---|
| Global minimum | 0.1566 | 0.1553 | 0.1345 | 0.1393 |

Table 7: Summary statistics of the sum of expert weights produced by the diffusion model across all samples from different datasets.

|  | Sines | Stocks | Energy | MuJoCo |
|---|---|---|---|---|
| Mean | 1.0114 | 1.0046 | 1.0089 | 1.0085 |
| Maximum | 1.0255 | 1.0305 | 1.0235 | 1.0163 |
| Minimum | 0.9894 | 0.9766 | 0.9887 | 0.9726 |
| Standard deviation | 0.0027 | 0.0044 | 0.0045 | 0.0068 |

### F.2 VISUALIZATION ANALYSIS OF CONTINUOUS GENERATION

As shown in Figures 17, 18 and 19, from the visualization of continuous generation, it can be observed that the time series generated by our method generally conforms to the distribution of the original temporal patterns. Based on the dynamic functions learned during training, our method generates contextually relevant local fluctuation patterns at finer time intervals. This provides richer

---

[3]https://anonymous.4open.science/r/MNTSG-79C2

Table 8: Average expert weights learned by NCDE across all samples from different datasets.

|  | Expert 1 | Expert 2 | Expert 3 | Expert 4 |
|---|---|---|---|---|
| Sines | **0.31384414** | 0.2231202 | 0.26881662 | 0.19421977 |
| Stocks | 0.19200034 | 0.21767974 | **0.32304975** | 0.2672697 |
| Energy | 0.2413441 | **0.31066164** | 0.2065838 | 0.24141027 |
| MuJoCo | 0.18345672 | 0.241069 | 0.2046078 | **0.370866** |
| ECG200 | 0.21229412 | **0.44273788** | 0.20304906 | 0.14191894 |
| ECG5K | 0.21999635 | 0.182932 | **0.3610456** | 0.2360259 |
| ECGFD | 0.17836495 | **0.31938562** | 0.2739047 | 0.2283447 |
| TLECG | 0.19264506 | 0.2577268 | **0.3210036** | 0.2286246 |

Table 9: Impact of enforcing the weights to sum to 1 on forecasting performance under continuous generation. Results are averaged over sequence lengths of 12, 24, and 36. "Normalize" indicates that each weight vector is divided by its sum so that it sums to 1, while "Original" refers to directly using the raw weights generated by the diffusion model.

|  |  | 30% | | | | 50% | | | | 70% | | | |
|---|---|---|---|---|---|---|---|---|---|---|---|---|---|
|  |  | Sines | Stocks | Energy | MuJoCo | Sines | Stocks | Energy | MuJoCo | Sines | Stocks | Energy | MuJoCo |
| MSE | Original | 0.026 | 0.002 | 0.013 | 0.029 | 0.026 | 0.002 | 0.013 | 0.029 | 0.028 | 0.002 | 0.013 | 0.029 |
| | Normalize | 0.026 | 0.002 | 0.013 | 0.029 | 0.026 | 0.002 | 0.013 | 0.029 | 0.028 | 0.002 | 0.014 | 0.029 |
| MAE | Original | 0.125 | 0.025 | 0.075 | 0.129 | 0.126 | 0.026 | 0.073 | 0.131 | 0.128 | 0.025 | 0.075 | 0.131 |
| | Normalize | 0.125 | 0.025 | 0.075 | 0.129 | 0.126 | 0.026 | 0.073 | 0.131 | 0.128 | 0.025 | 0.075 | 0.131 |

Table 10: Privacy-leakage risk evaluation of different methods on medical datasets, based on the Membership Inference Risk (MIR) metric. Lower is better, and a higher MIR score indicates that synthetic data more easily reveals information about real samples, implying a higher privacy risk.

|  |  | 30% | | | | 50% | | | | 70% | | | | |
|---|---|---|---|---|---|---|---|---|---|---|---|---|---|---|
|  |  | ECG200 | ECG5k | ECGFD | TLECG | ECG200 | ECG5k | ECGFD | TLECG | ECG200 | ECG5k | ECGFD | TLECG | Avg. |
| MIR | Ours | **0.6826** | 0.6944 | **0.6734** | **0.6671** | **0.6671** | 0.6676 | **0.6866** | 0.7302 | 0.697 | **0.6866** | **0.6765** | **0.6866** | **0.6846** |
| | KoVAE | 0.6873 | **0.6849** | 0.7194 | 0.6882 | 0.6761 | 0.6873 | 0.8846 | 0.8679 | 0.7931 | 0.7077 | 0.697 | 0.697 | 0.7325 |
| | GT-GAN | 0.6993 | 0.8772 | 0.9709 | 0.9794 | 0.7698 | 0.7628 | 0.9583 | 0.9787 | **0.6765** | 0.7541 | 0.807 | 0.7667 | 0.8334 |
| | TimeGAN-NCDE | 0.7463 | 0.7463 | 0.7463 | 0.6817 | 0.7163 | 0.8482 | 0.807 | 0.807 | 0.807 | 0.7667 | 0.7419 | 0.807 | 0.7685 |
| | TimeVAE-NCDE | 0.7194 | 0.7067 | 0.6897 | 0.6676 | 0.6676 | 0.6698 | 0.7541 | 0.7541 | 0.7797 | 0.7419 | 0.7419 | 0.7419 | 0.7195 |
| | Diffusion-NCDE | 0.7067 | 0.7067 | 0.6873 | 0.6693 | 0.6698 | 0.6698 | 0.7541 | 0.7797 | 0.8364 | 0.7419 | 0.7188 | 0.7302 | 0.7226 |
| | ProFITi | 0.7067 | 0.7042 | 0.7194 | 0.6988 | 0.6798 | 0.6969 | 0.7797 | 0.902 | 0.807 | 0.7797 | 0.7188 | 0.8846 | 0.7565 |
| | HetVAE | 0.7692 | 0.6993 | 0.7117 | 0.7981 | 0.8163 | 0.8299 | 0.7931 | 0.8214 | 0.807 | 0.7188 | 0.7302 | 0.8519 | 0.7789 |

fine-grained information for downstream task decision-making, thereby contributing to improved accuracy in downstream prediction tasks.

This advantage is attributed to two main aspects. First, our proposed decoupled MOE NCDE enables NCDE to focus on learning multiple dynamic functions, adapting to complex and changing temporal patterns. This enhances its ability to handle irregular time series data and serve continuous generation tasks effectively. Second, by allowing the model to learn the joint distribution of mixture-of-experts dynamic functions and time series samples, it assigns new and appropriate expert weights to the generated time series samples, which are further used to weight the dynamic functions for continuous generation.

The visualization results demonstrate the effectiveness of the continuous time series generation method we proposed. In contrast, the continuous generation performances of GTGAN (Figures 20, 21 and 22) and KOVAE (Figures 23, 24 and 25) are relatively poor. The data they generate significantly deviates from the original temporal patterns, introducing substantial noise. This, in turn, leads to worse performance in downstream prediction tasks.

### F.2.1 HYPERPARAMETER SENSITIVITY ANALYSIS ON DISCRIMINATIVE SCORE METRIC

### F.2.2 IRREGULAR TIME SERIES GENERATION

**Results on each sequence length** In Table 1 of the main text, we reported the average results for time lengths of 12, 24, and 36. Here, we present the results of different methods for each length, as shown in Tables 11, 12, and 13. The conclusions are consistent with those in the main text. In the task of generating regular time series from irregular time series data, our method consistently outperforms the baseline models. This demonstrates that the proposed MOE-NCDE is effective.

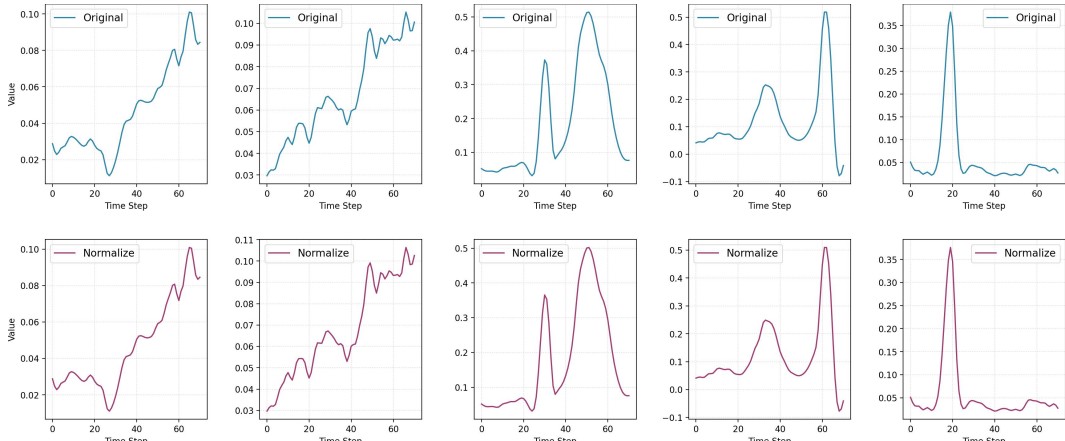

Figure 8: Visualizing the generated patterns (energy dataset) by MOE-NCDE in the condition of normalizing or not normalizing the generated MOE weights.

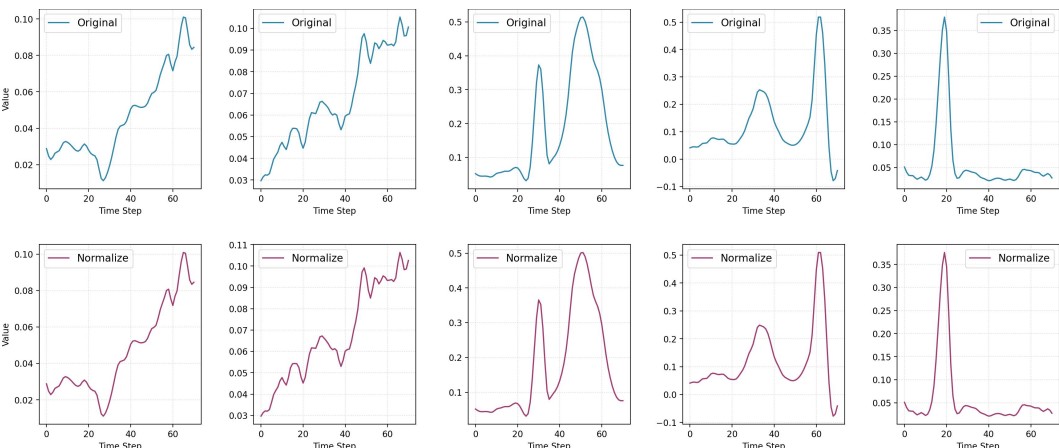

Figure 9: Visualizing the generated patterns (MuJoCo dataset) by MOE-NCDE in the condition of normalizing or not normalizing the generated MOE weights.

Through the mixture-of-experts dynamic functions and the decoupled training strategy, it enables NCDE to better capture dynamic temporal patterns, thereby better handling missing values and achieving the best results.

**More qualitative evaluations** In the main text, we presented a qualitative study with 70% missing values in Figure 3. Here, we further demonstrate the performance with 30% and 50% missing values in Figure 27. The results show that the data distribution learned by our method is closer to the original data distribution than that of the current baseline models. This further illustrates the effectiveness of the proposed MOE-NCDE method.

### F.2.3 CONTINUOUS TIME SERIES GENERATION

**Results of each sequence length on the downstream task** In Table 3 of the main text, we reported the average forecasting performance for time lengths of 12, 24, and 36. Here, we present the results of different methods for each length, as shown in Tables 14, 15, and 16. The conclusions drawn are consistent with those in the main text. Through refined continuous time series generation, our method has enabled downstream task models to achieve better forecasting performance. This is because the refined continuous generation provides more useful and richer information, which helps the models obtain better training effects.

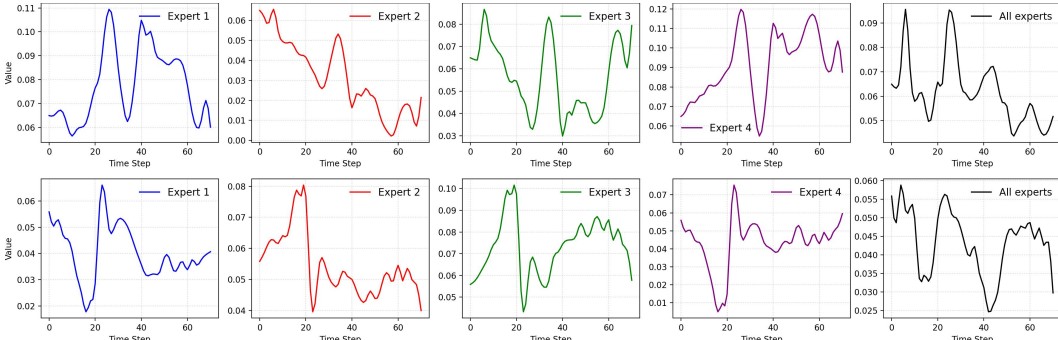

Figure 10: Visualizing the specific temporal dynamics learned by MoE-NCDE on the **energy** dataset in the continuous generation task. We randomly selected two samples for visualization. From left to right, the plots show the results obtained by using only Expert 1, only Expert 2, ..., up to using all experts with their learned temporal dynamics for data generation. To illustrate this effect, we load the trained NCDE model and manually adjust the expert weights to observe how the generated patterns change. When a single expert is used, the weights of all other experts are set to zero. We observe that different experts capture distinct temporal behaviors. Because real-world time series often exhibit complex and heterogeneous dynamics, relying on a single expert is insufficient. Decomposing these dynamics across multiple experts and subsequently recombining them significantly enhances NCDE's capability to model diverse temporal patterns.

Table 11: Irregular time series (30%, 50% and 70% of observations are dropped) on popular datasets with sequence length 12.

| | | 30% | | | | 50% | | | | 70% | | |
|---|---|---|---|---|---|---|---|---|---|---|---|---|
| | | Sines | Stocks | Energy | MuJoCo | Sines | Stocks | Energy | MuJoCo | Sines | Stocks | Energy | MuJoCo |
| **DS** | Ours | **0.077** | **0.084** | **0.386** | **0.268** | **0.13** | **0.042** | **0.472** | **0.417** | **0.222** | **0.054** | **0.494** | **0.45** |
| | KoVAE | 0.218 | 0.103 | 0.500 | 0.358 | 0.202 | 0.106 | 0.500 | 0.390 | 0.211 | 0.228 | 0.500 | 0.451 |
| | GT-GAN | 0.300 | 0.385 | 0.499 | 0.498 | 0.423 | 0.357 | 0.499 | 0.496 | 0.434 | 0.370 | 0.498 | 0.497 |
| | TimeGAN-NCDE | 0.468 | 0.369 | 0.498 | 0.487 | 0.422 | 0.472 | 0.499 | 0.499 | 0.429 | 0.479 | 0.500 | 0.498 |
| | TimeVAE-NCDE | 0.354 | 0.493 | 0.496 | 0.433 | 0.466 | 0.497 | 0.491 | 0.477 | 0.500 | 0.497 | 0.498 | 0.480 |
| | Diffusion-NCDE | 0.264 | 0.428 | 0.426 | 0.323 | 0.353 | 0.472 | 0.482 | 0.437 | 0.480 | 0.491 | 0.497 | 0.475 |
| | ProFITi | 0.29 | 0.44 | 0.492 | 0.49 | 0.339 | 0.478 | 0.498 | 0.496 | 0.41 | 0.486 | 0.5 | 0.497 |
| | HeTVAE | 0.5 | 0.368 | 0.493 | 0.499 | 0.5 | 0.478 | 0.499 | 0.5 | 0.5 | 0.493 | 0.5 | 0.5 |
| **MDD** | Ours | **1.015** | **0.16** | **0.303** | **0.303** | **1.337** | **0.2** | **0.251** | **0.319** | **1.783** | **0.219** | **0.319** | **0.34** |
| | KoVAE | 10.137 | 0.289 | 0.343 | 0.326 | 10.12 | 0.288 | 0.377 | 0.372 | 10.13 | 0.312 | 0.399 | 0.635 |
| | GT-GAN | 3.574 | 0.746 | 0.598 | 0.540 | 3.898 | 0.600 | 0.414 | 0.565 | 3.38 | 0.733 | 0.562 | 0.585 |
| | TimeGAN-NCDE | 4.599 | 1.109 | 1.020 | 1.261 | 4.799 | 1.625 | 1.395 | 1.485 | 5.35 | 1.717 | 1.499 | 1.761 |
| | TimeVAE-NCDE | 8.661 | 0.712 | 0.451 | 0.463 | 6.045 | 0.864 | 0.587 | 0.586 | 5.184 | 1.054 | 0.652 | 0.707 |
| | Diffusion-NCDE | 2.150 | 0.328 | 0.306 | 0.276 | 2.128 | 0.538 | 0.365 | 0.428 | 2.902 | 0.854 | 0.506 | 0.508 |
| | ProFITi | 1.277 | 0.32 | 0.254 | 0.298 | 1.342 | 0.416 | 0.342 | 0.361 | 1.584 | 0.443 | 0.442 | 0.446 |
| | HeTVAE | 3.594 | 0.597 | 0.635 | 0.507 | 4.25 | 0.748 | 1.11 | 1.092 | 5.134 | 0.986 | 1.273 | 1.105 |
| **KL** | Ours | **0.007** | **0.05** | **0.019** | **0.017** | **0.014** | **0.075** | **0.025** | **0.018** | **0.033** | **0.066** | **0.013** | **0.024** |
| | KoVAE | 9.941 | 0.139 | 0.076 | 0.168 | 9.511 | 0.149 | 0.084 | 0.282 | 9.788 | 0.151 | 0.111 | 0.579 |
| | GT-GAN | 0.144 | 0.283 | 0.063 | 0.020 | 0.187 | 0.212 | 0.050 | 0.029 | 0.164 | 0.314 | 0.066 | 0.031 |
| | TimeGAN-NCDE | 5.230 | 1.742 | 0.210 | 0.575 | 1.125 | 7.401 | 0.716 | 0.997 | 15.176 | 12.148 | 0.854 | 2.397 |
| | TimeVAE-NCDE | 3.551 | 0.298 | 0.089 | 0.158 | 2.776 | 0.345 | 0.128 | 0.245 | 2.113 | 0.534 | 0.148 | 0.338 |
| | Diffusion-NCDE | 0.004 | 0.012 | 0.004 | 0.007 | 0.028 | 0.040 | 0.019 | 0.036 | 0.074 | 0.095 | 0.046 | 0.042 |
| | ProFITi | 0.078 | 0.084 | 0.022 | 0.016 | 0.082 | 0.124 | 0.032 | 0.016 | 0.11 | 0.137 | 0.044 | 0.032 |
| | HeTVAE | 0.036 | 0.302 | 0.376 | 0.102 | 2.655 | 0.46 | 1.124 | 0.934 | 0.288 | 0.78 | 1.275 | 1.079 |

**More results for the analytical solutions of polynomial functions** As shown in Figure 28, we visualize the patterns of different curves in cubic polynomials. Since the polynomial coefficients are randomly sampled from normal distributions, respectively, there are differences in the patterns of different curves. The existing standard NCDE, which only uses a single dynamic function, has

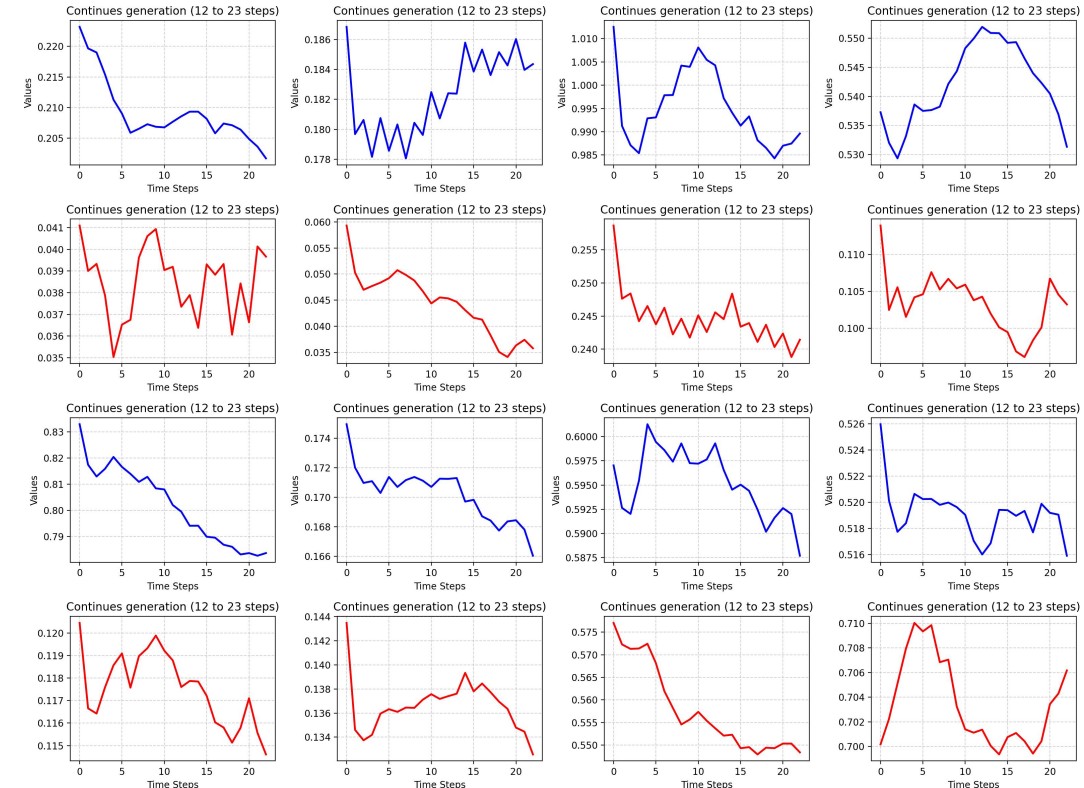

Figure 11: Visualizing continuous generation on the stock dataset of **our method MNTSG**. We refine 12 time steps to 23 time steps. The model is trained on the data with 30% missing values. **The interpolation pattern of NCDE is jointly governed by the learned dynamics and the MOE weights produced by the diffusion model. During continuous generation, NCDE is observed to produce diverse patterns, and the saw-tooth pattern is only one of them.**

difficulty dealing with such dynamically changing temporal patterns. The dataset generation script can be found in the code link.

In the main text, we demonstrated the performance of our method in terms of analysis solution evaluation with 30% missing values. Here, we further show its performance with 50% and 70% missing values. As shown in Figure 29, after refined continuous generation, our method can still obtain good analysis solutions that conform to the true distribution. This further illustrates that our continuous generation method is effective and the generated finer-grained time values comply with the original data distribution. Additionally, we observe that continuous generation helps obtain better analysis solutions on coefficient $d$, as shown in the fourth subplot of Figure 29 from left to right. By continuously generating refined curves (with a length twice that of the original), and then computing the analysis solutions on them, we can achieve more accurate solutions compared to normally generated samples. The enhanced accuracy is attributed to the richer information provided by the continuous generation process, which brings the distribution of the analysis solutions closer to the true distribution. This further illustrates the effectiveness of our continuous generation method.

We also present the comparison results of our method with KOVAE and GTGAN. As shown in Figures 30 and 31, since both KOVAE and GTGAN only use the standard NCDE, they have difficulty handling dynamically changing temporal patterns. The analysis solution distributions learned from the samples generated by them deviate from the true distribution. Our method, benefiting from the mixture-of-experts dynamic functions and decoupled design, can better learn different dynamic functions and thus better handle dynamically changing temporal patterns for more accurate interpolation.

**Impact of control-path smoothness in NCDEs.** Although the control path $X(s)$ is smooth due to spline interpolation, the NCDE dynamics do not impose smoothness on the final generated trajectory.

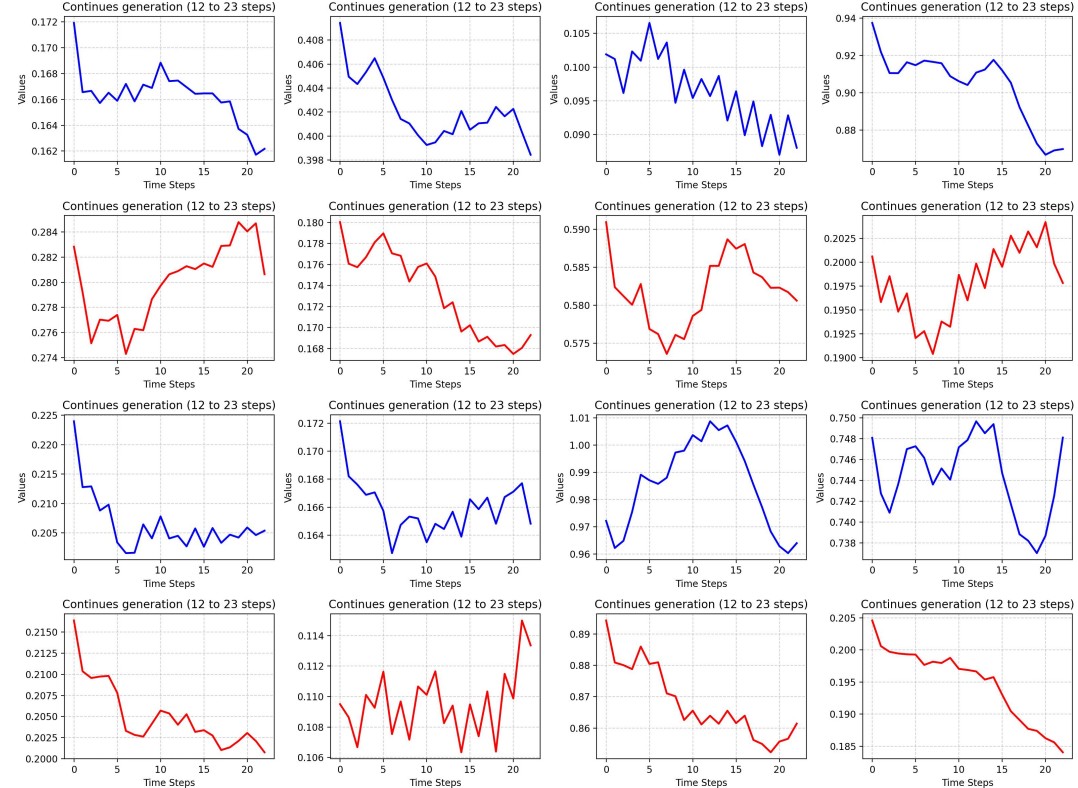

Figure 12: Visualizing continuous generation on the stock dataset of **our method MNTSG**. We refine 12 time steps to 23 time steps. The model is trained on the data with 30% missing values. **The generation pattern of NCDE is jointly governed by the learned dynamics and the MOE weights produced by the diffusion model. During continuous generation, NCDE is observed to produce diverse patterns, and the saw-tooth pattern is only one of them.**

As shown in Eq. (1), the control path $X(s)$ is smooth, but the evolution of $z(t)$ is governed by the nonlinear neural network $f_\theta(z(s))$. Thus, even with a smooth control path, the latent trajectory $z(t)$ can evolve in a highly non-smooth and expressive manner. Moreover, the mixture-of-experts component further facilitates the modeling of non-smooth and highly diverse temporal patterns. To validate this, we added visualizations of sequences generated solely by NCDE (Appendix Figures 14–16 in the revised manuscript). These results show:

- For volatile datasets such as energy (Figure 14) and stock (Figure 15), NCDE generates naturally fluctuating trajectories, without excessive smoothing.

- For intrinsically smooth data (e.g., sine in Figure 16), NCDE produces smooth trajectories.

In summary, while the control path is smooth, the nonlinear MOE–NCDE dynamics allow the generated sequences to exhibit both smooth and abrupt temporal behaviors, depending entirely on the underlying temporal dynamics.

**Discussions about generated patterns of MOE-NCDE.** We provide additional visualizations in Appendix Figures 11 to 13. They show that the continuous trajectories generated by NCDE vary according to the learned dynamics and the MOE weights produced by the diffusion model. During continuous generation, NCDE exhibits diverse behaviors—including stable, fluctuating, and non-smooth patterns. The saw-tooth pattern is not representative of all cases but rather one instance of the broader range of dynamics the model can generate.

**Discussion of how to select the expert number and its generalization.** We recommend determining the number of experts based on two complementary principles:

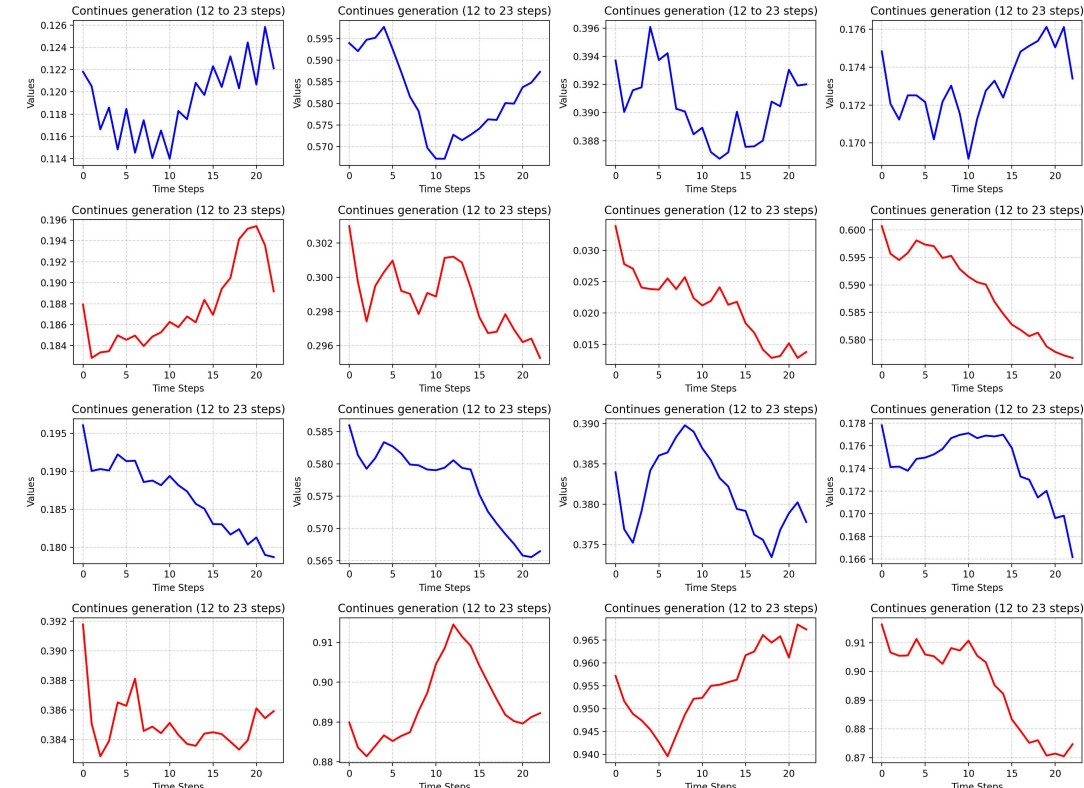

Figure 13: Visualizing continuous generation on the stock dataset of **our method MNTSG**. We refine 12 time steps to 23 time steps. The model is trained on the data with 30% missing values. **The generation pattern of NCDE is jointly governed by the learned dynamics and the MOE weights produced by the diffusion model. During continuous generation, NCDE is observed to produce diverse patterns, and the saw-tooth pattern is only one of them.**

- Data-driven temporal-pattern analysis. We can perform clustering on dynamic features such as autocorrelation and volatility to estimate how many distinct temporal modes exist in a dataset. For our datasets (e.g., ECG with 2–5 disease classes, multivariate sine/stock datasets with periodic and aperiodic sub-patterns), four experts are sufficient to cover the dominant dynamic types without unnecessary model complexity.

- Empirical validation. As shown in Figure 7 of the revised manuscript, we evaluate expert numbers ranging from 1 to 9. Performance consistently improves up to 4 experts, and then the benefit becomes negligible—adding more experts increases computation while offering no obvious benefit. Therefore, we select 4 experts as a balanced and generalizable default for TSG tasks.

Besides, Figure 7 further indicates that generalization performance is not highly sensitive to the expert number. While four experts provide the best overall results, using more experts does not significantly degrade performance, though it increases computational cost. Thus, the model's generalization ability is stable with respect to this hyperparameter.

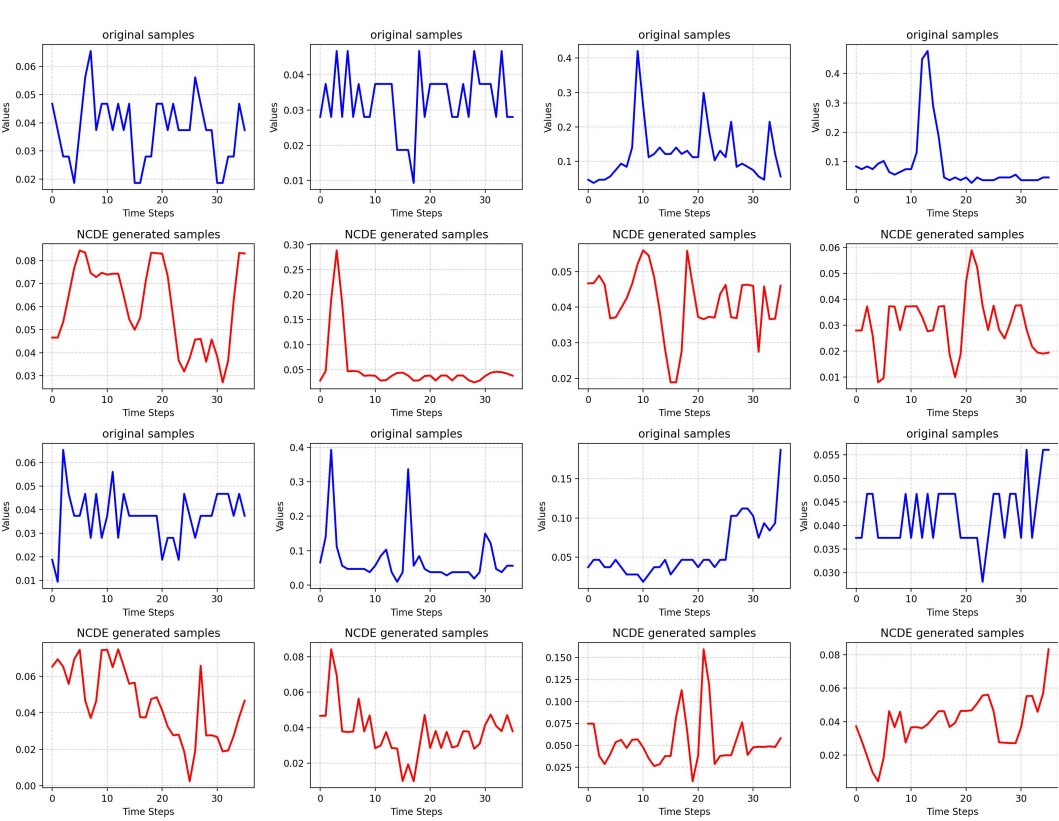

Figure 14: Visualizations demonstrate that NCDE does not produce overly smooth curves on the energy dataset: the generated pattern is dictated by the data themselves. For example, the sine dataset is intrinsically smooth (Figure 16 ), so NCDE also yields smooth samples, whereas energy and stock (Figure 15) datasets are naturally volatile, and **NCDE accordingly generates fluctuating rather than excessively smooth sequences. The samples are randomly generated, and there is no one-to-one correspondence between the images.**

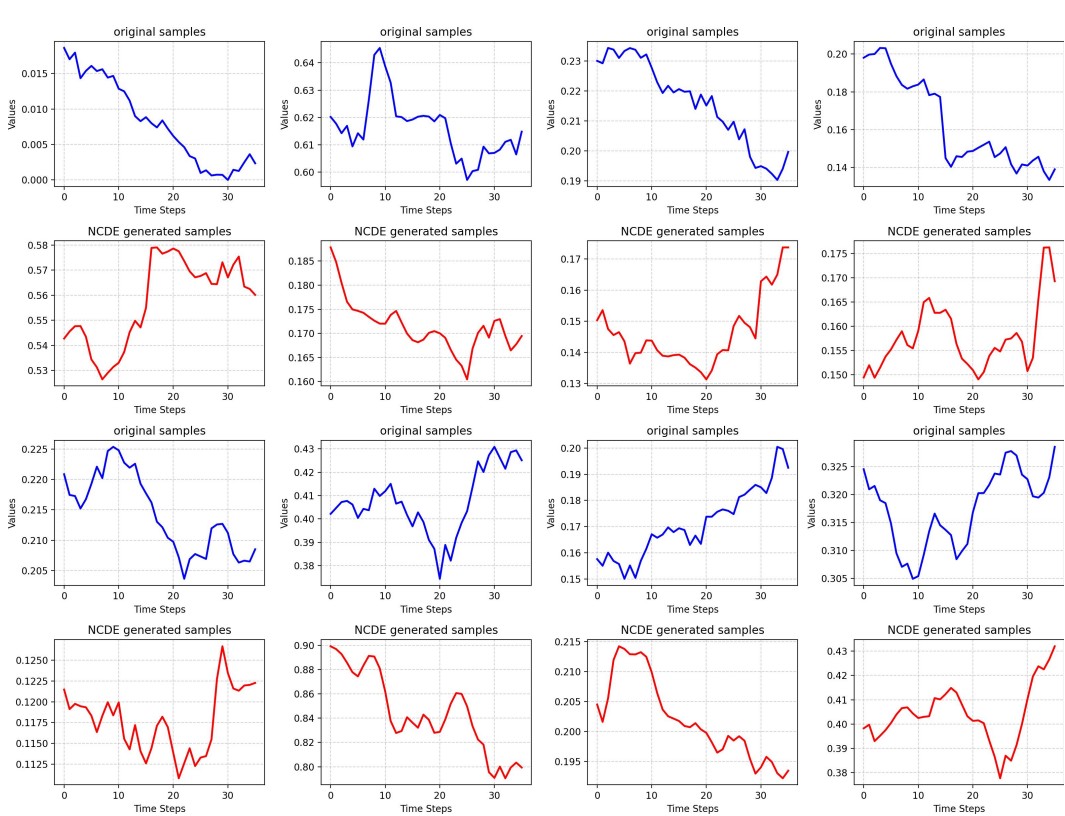

Figure 15: Visualizations demonstrate that NCDE does not produce overly smooth curves on the stock dataset: the generated pattern is dictated by the data themselves. For example, the sine dataset is intrinsically smooth, so NCDE also yields smooth samples, whereas energy and stock datasets are naturally volatile, and **NCDE accordingly generates fluctuating rather than excessively smooth sequences. The samples are randomly generated, and there is no one-to-one correspondence between the images.**

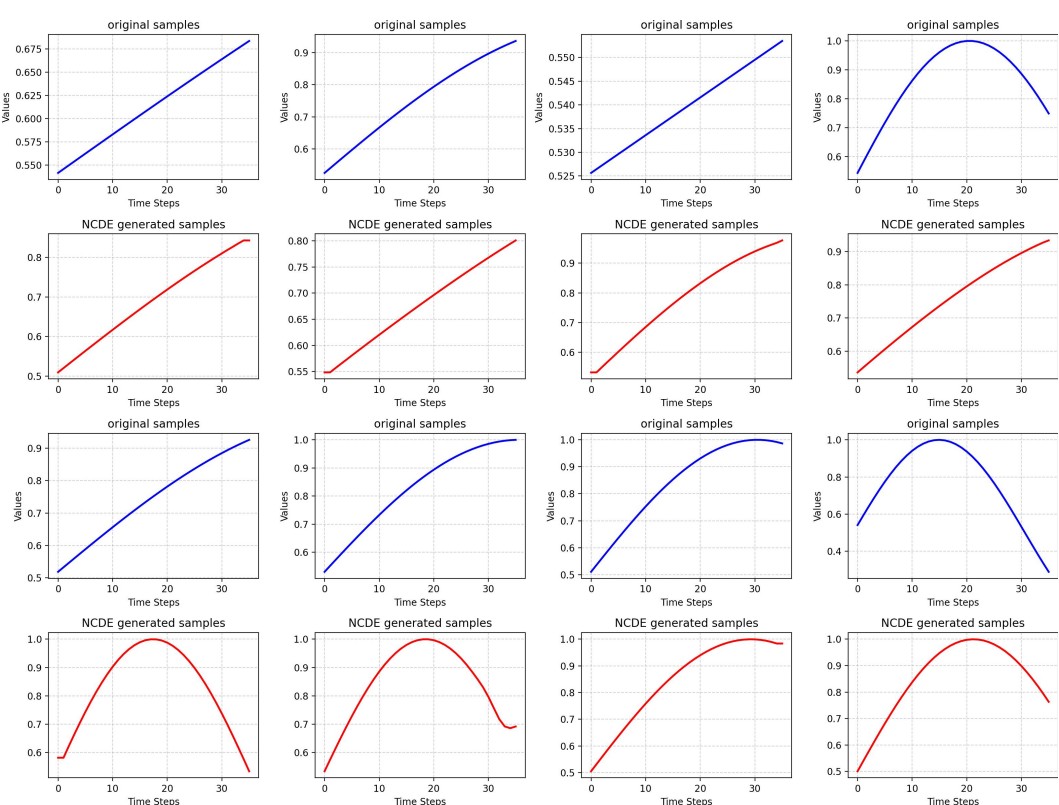

Figure 16: Visualizations demonstrate that NCDE does not produce overly smooth curves on the energy dataset: the generated pattern is dictated by the data themselves. For example, the sine dataset is intrinsically smooth, so NCDE also yields smooth samples, whereas energy and stock datasets are naturally volatile, and **NCDE accordingly generates fluctuating rather than excessively smooth sequences. The samples are randomly generated, and there is no one-to-one correspondence between the images.**

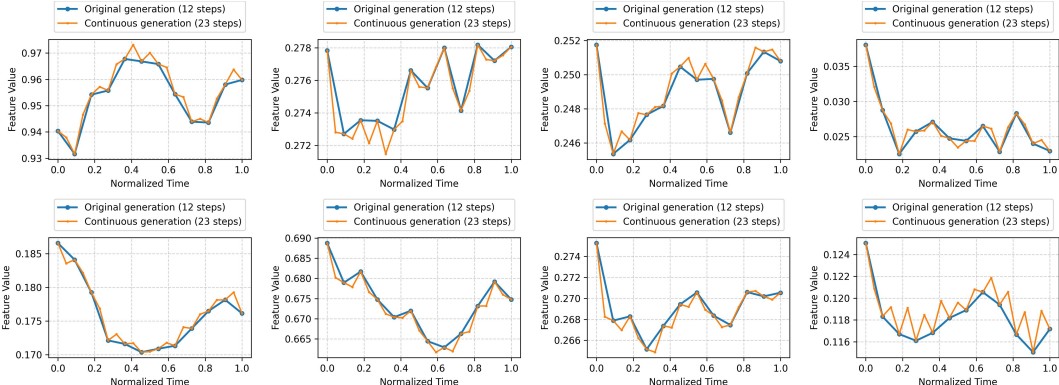

Figure 17: Visualizing continuous generation on the stock dataset of **our method MNTSG**. The model is trained on the data with 30% missing values. Here we add a comparison with the original sequence with 12 time steps.

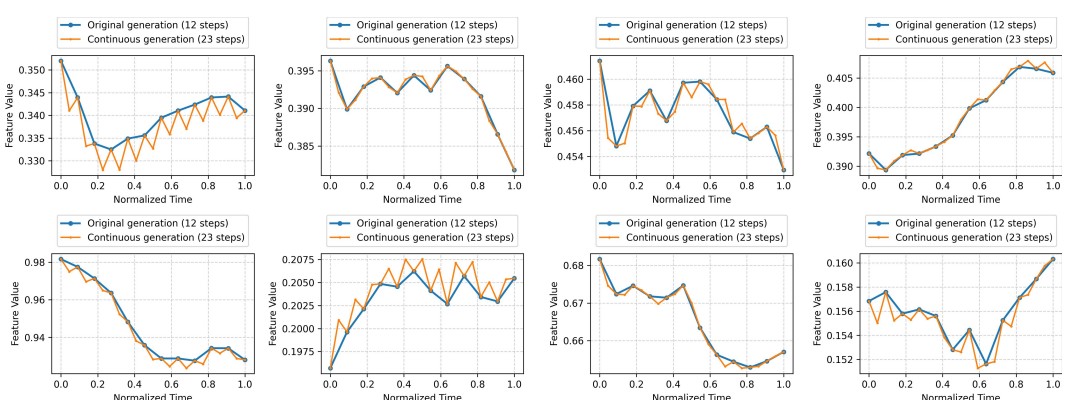

Figure 18: Visualizing continuous generation on the stock dataset of **our method MNTSG**. The model is trained on the data with 50% missing values. Here we add a comparison with the original sequence with 12 time steps.

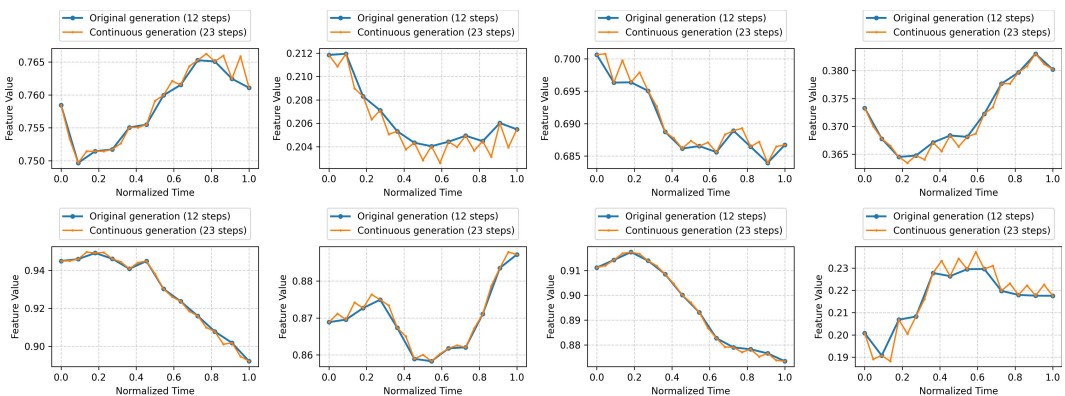

Figure 19: Visualizing continuous generation on the stock dataset of **our method MNTSG**. The model is trained on the data with 70% missing values. Here we add a comparison with the original sequence with 12 time steps.

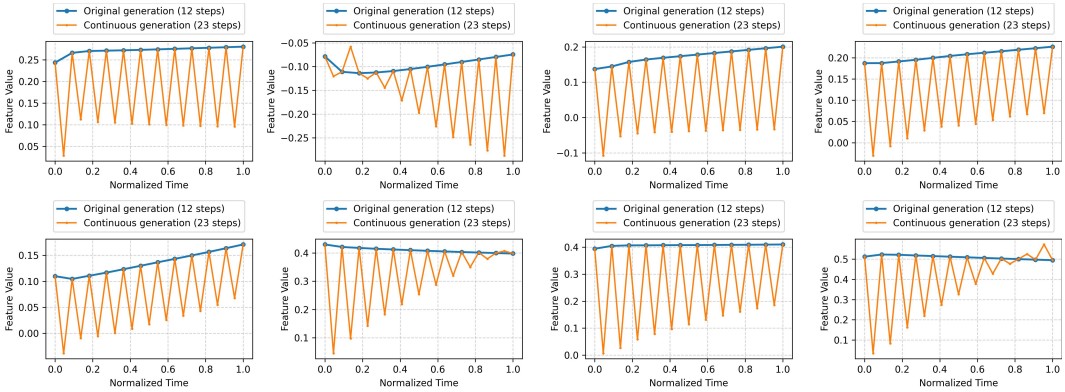

Figure 20: Visualizing continuous generation on the stock dataset of **baseline GTGAN**. The model is trained on the data with 30% missing values.

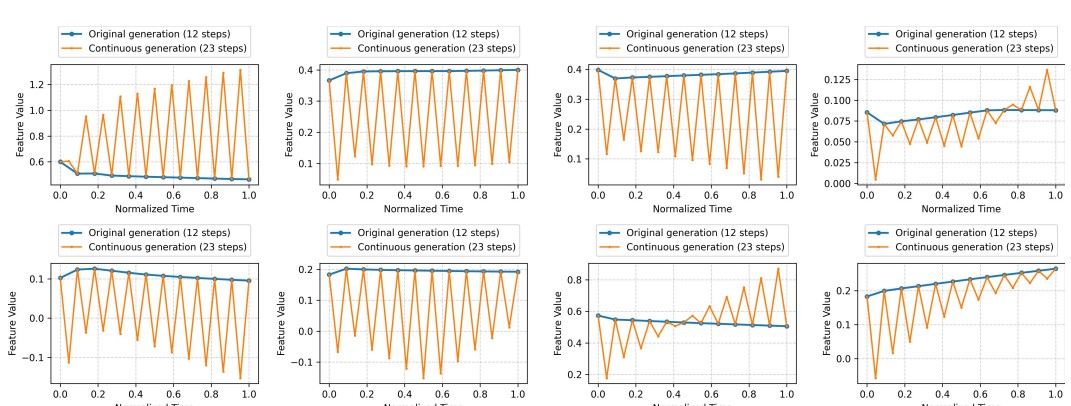

Figure 21: Visualizing continuous generation on the stock dataset of **baseline GTGAN**. The model is trained on the data with 50% missing values.

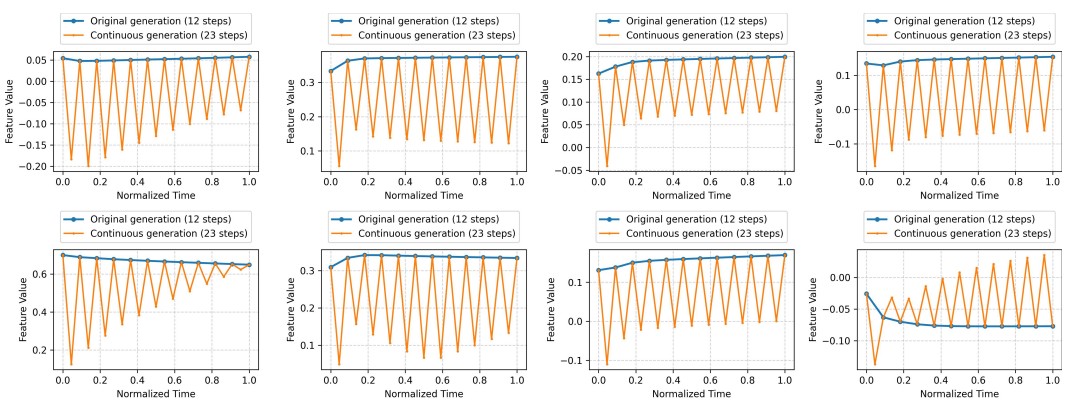

Figure 22: Visualizing continuous generation on the stock dataset of **baseline GTGAN**. The model is trained on the data with 70% missing values.

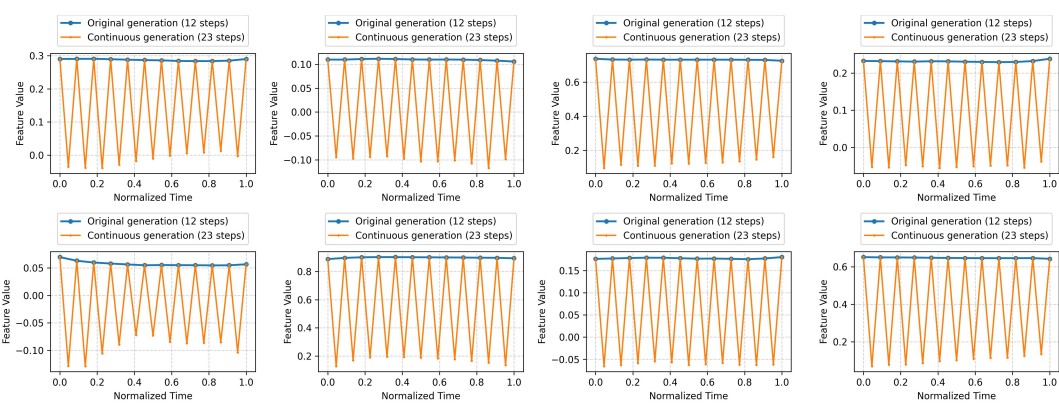

Figure 23: Visualizing continuous generation on the stock dataset of **baseline KOVAE**. The model is trained on the data with 30% missing values.

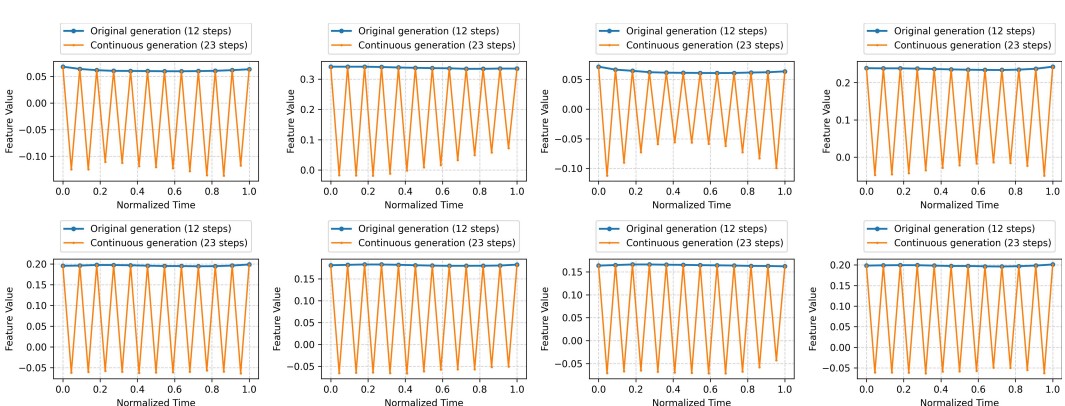

Figure 24: Visualizing continuous generation on the stock dataset of **baseline KOVAE**. The model is trained on the data with 50% missing values.

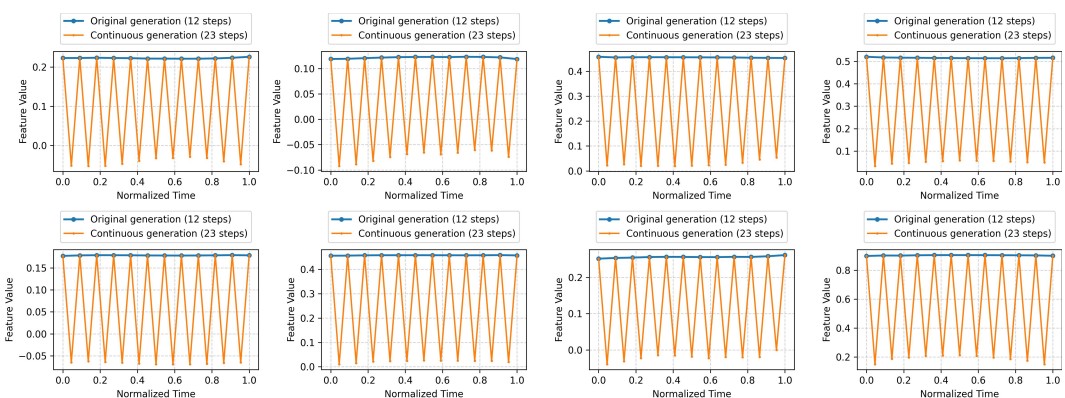

Figure 25: Visualizing continuous generation on the stock dataset of **baseline KOVAE**. The model is trained on the data with 70% missing values.

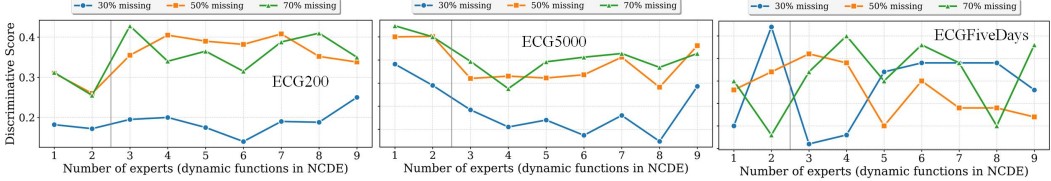

Figure 26: The impact of the number of experts (1 to 9 in the x-axis) on the DS metric.

Table 12: Irregular time series (30%, 50% and 70% of observations are dropped) on popular datasets with sequence length 24.

| | | 30% | | | | 50% | | | | 70% | | |
|---|---|---|---|---|---|---|---|---|---|---|---|---|
| | | Sines | Stocks | Energy | MuJoCo | Sines | Stocks | Energy | MuJoCo | Sines | Stocks | Energy | MuJoCo |
| **DS** | Ours | **0.069** | **0.171** | **0.414** | **0.249** | **0.072** | **0.151** | **0.491** | **0.306** | **0.127** | **0.179** | **0.499** | **0.347** |
| | KoVAE | 0.055 | 0.285 | 0.500 | 0.358 | 0.090 | 0.208 | 0.500 | 0.363 | 0.212 | 0.236 | 0.500 | 0.403 |
| | GT-GAN | 0.276 | 0.305 | 0.500 | 0.480 | 0.338 | 0.325 | 0.500 | 0.491 | 0.286 | 0.273 | 0.500 | 0.489 |
| | TimeGAN-NCDE | 0.457 | 0.351 | 0.500 | 0.486 | 0.433 | 0.454 | 0.500 | 0.497 | 0.438 | 0.472 | 0.499 | 0.499 |
| | TimeVAE-NCDE | 0.191 | 0.481 | 0.499 | 0.324 | 0.175 | 0.493 | 0.499 | 0.462 | 0.431 | 0.490 | 0.499 | 0.464 |
| | Diffusion-NCDE | 0.148 | 0.460 | 0.455 | 0.373 | 0.286 | 0.490 | 0.489 | 0.409 | 0.392 | 0.489 | 0.496 | 0.470 |
| | ProFITi | 0.298 | 0.459 | 0.498 | 0.496 | 0.365 | 0.487 | 0.499 | 0.498 | 0.409 | 0.492 | 0.5 | 0.498 |
| | HeTVAE | 0.48 | 0.306 | 0.498 | 0.499 | 0.5 | 0.401 | 0.5 | 0.499 | 0.5 | 0.476 | 0.5 | 0.5 |
| **MDD** | Ours | **0.752** | **0.269** | **0.218** | **0.336** | **0.775** | **0.251** | **0.216** | **0.341** | **0.971** | **0.299** | **0.265** | **0.285** |
| | KoVAE | 2.756 | 0.905 | 0.347 | 0.341 | 3.314 | 0.833 | 0.361 | 0.380 | 5.221 | 0.763 | 0.368 | 0.633 |
| | GT-GAN | 2.257 | 0.551 | 0.429 | 0.576 | 2.467 | 0.563 | 0.396 | 0.532 | 2.277 | 0.532 | 0.413 | 0.555 |
| | TimeGAN-NCDE | 3.162 | 1.134 | 1.046 | 1.173 | 3.555 | 1.597 | 1.226 | 1.553 | 3.609 | 1.592 | 1.605 | 1.651 |
| | TimeVAE-NCDE | 3.408 | 0.580 | 0.418 | 0.307 | 3.042 | 0.709 | 0.521 | 0.411 | 2.862 | 0.804 | 0.597 | 0.527 |
| | Diffusion-NCDE | 1.303 | 0.401 | 0.250 | 0.336 | 1.402 | 0.403 | 0.356 | 0.338 | 1.667 | 0.692 | 0.557 | 0.486 |
| | ProFITi | 0.961 | 0.311 | 0.261 | 0.284 | 1.446 | 0.472 | 0.358 | 0.333 | 1.645 | 0.436 | 0.444 | 0.407 |
| | HeTVAE | 2.775 | 0.582 | 0.607 | 0.722 | 3.431 | 0.787 | 0.993 | 1.09 | 3.677 | 1.071 | 1.229 | 1.445 |
| **KL** | Ours | **0.011** | **0.084** | **0.009** | **0.007** | **0.022** | **0.092** | **0.014** | **0.006** | **0.035** | **0.105** | **0.023** | **0.013** |
| | KoVAE | 1.696 | 1.347 | 0.058 | 0.150 | 1.861 | 1.205 | 0.061 | 0.198 | 2.840 | 1.380 | 0.071 | 0.508 |
| | GT-GAN | 0.065 | 0.163 | 0.042 | 0.032 | 0.139 | 0.161 | 0.053 | 0.019 | 0.190 | 0.184 | 0.051 | 0.034 |
| | TimeGAN-NCDE | 4.573 | 1.380 | 0.227 | 0.666 | 4.339 | 5.887 | 0.489 | 1.248 | 4.854 | 10.508 | 1.119 | 1.409 |
| | TimeVAE-NCDE | 2.18 | 0.209 | 0.067 | 0.046 | 1.291 | 0.245 | 0.102 | 0.070 | 1.038 | 0.284 | 0.132 | 0.112 |
| | Diffusion-NCDE | 0.060 | 0.438 | 0.073 | 0.033 | 0.053 | 0.440 | 0.099 | 0.041 | 0.077 | 0.498 | 0.114 | 0.047 |
| | ProFITi | 0.128 | 0.091 | 0.028 | 0.01 | 0.137 | 0.161 | 0.036 | 0.012 | 0.189 | 0.123 | 0.043 | 0.026 |
| | HeTVAE | 0.323 | 0.792 | 0.175 | 0.318 | 8.712 | 1.203 | 0.495 | 0.915 | 0.337 | 1.398 | 1.429 | 2.343 |

Table 13: Irregular time series (30%, 50% and 70% of observations are dropped) on popular datasets with sequence length 36.

| | | 30% | | | | 50% | | | | 70% | | |
|---|---|---|---|---|---|---|---|---|---|---|---|---|
| | | Sines | Stocks | Energy | MuJoCo | Sines | Stocks | Energy | MuJoCo | Sines | Stocks | Energy | MuJoCo |
| **DS** | Ours | **0.169** | **0.170** | **0.466** | **0.363** | **0.183** | **0.218** | **0.499** | **0.403** | **0.196** | **0.085** | **0.499** | **0.381** |
| | KoVAE | 0.152 | 0.286 | 0.429 | 0.399 | 0.222 | 0.247 | 0.500 | 0.439 | 0.261 | 0.176 | 0.500 | 0.422 |
| | GT-GAN | 0.331 | 0.272 | 0.499 | 0.489 | 0.487 | 0.297 | 0.500 | 0.494 | 0.303 | 0.234 | 0.500 | 0.489 |
| | TimeGAN-NCDE | 0.440 | 0.376 | 0.499 | 0.496 | 0.425 | 0.399 | 0.498 | 0.500 | 0.453 | 0.48 | 0.500 | 0.499 |
| | TimeVAE-NCDE | 0.157 | 0.437 | 0.500 | 0.368 | 0.282 | 0.464 | 0.499 | 0.451 | 0.347 | 0.490 | 0.498 | 0.485 |
| | Diffusion-NCDE | 0.189 | 0.435 | 0.498 | 0.368 | 0.352 | 0.462 | 0.496 | 0.436 | 0.391 | 0.493 | 0.497 | 0.474 |
| | ProFITi | 0.342 | 0.481 | 0.499 | 0.498 | 0.354 | 0.493 | 0.5 | 0.498 | 0.357 | 0.494 | 0.5 | 0.498 |
| | HeTVAE | 0.494 | 0.309 | 0.499 | 0.498 | 0.5 | 0.449 | 0.5 | 0.499 | 0.5 | 0.493 | 0.5 | 0.5 |
| **MDD** | Ours | **1.091** | **0.322** | **0.288** | **0.402** | **1.167** | **0.392** | **0.288** | **0.293** | **1.17** | **0.38** | **0.253** | **0.267** |
| | KoVAE | 2.509 | 0.933 | 0.350 | 0.456 | 3.319 | 0.865 | 0.366 | 0.506 | 3.828 | 0.798 | 0.363 | 0.588 |
| | GT-GAN | 2.101 | 0.593 | 0.673 | 0.558 | 2.568 | 0.604 | 0.610 | 0.692 | 2.021 | 0.543 | 0.620 | 0.74 |
| | TimeGAN-NCDE | 2.670 | 1.091 | 1.019 | 1.111 | 2.863 | 1.592 | 1.408 | 1.568 | 2.984 | 1.578 | 1.574 | 1.7 |
| | TimeVAE-NCDE | 2.232 | 0.499 | 0.463 | 0.243 | 2.195 | 0.604 | 0.491 | 0.331 | 2.144 | 0.670 | 0.564 | 0.462 |
| | Diffusion-NCDE | 0.934 | 0.295 | 0.319 | 0.308 | 0.963 | 0.487 | 0.331 | 0.345 | 1.421 | 0.682 | 0.475 | 0.468 |
| | ProFITi | 0.935 | 0.252 | 0.259 | 0.289 | 1.024 | 0.394 | 0.366 | 0.337 | 1.161 | 0.45 | 0.444 | 0.452 |
| | HeTVAE | 3.104 | 0.619 | 0.634 | 0.734 | 2.665 | 0.784 | 1.182 | 1.098 | 3.091 | 1.038 | 1.466 | 1.644 |
| **KL** | Ours | **0.020** | **0.089** | **0.031** | **0.039** | **0.033** | **0.115** | **0.026** | **0.003** | **0.031** | **0.101** | **0.014** | **0.004** |
| | KoVAE | 0.878 | 1.555 | 0.061 | 0.072 | 2.785 | 1.224 | 0.057 | 0.115 | 3.935 | 1.072 | 0.057 | 0.188 |
| | GT-GAN | 0.087 | 0.163 | 0.064 | 0.012 | 0.134 | 0.171 | 0.051 | 0.035 | 0.134 | 0.171 | 0.054 | 0.078 |
| | TimeGAN-NCDE | 5.086 | 2.104 | 0.341 | 0.440 | 3.398 | 7.039 | 0.551 | 1.214 | 17.403 | 12.235 | 1.012 | 1.625 |
| | TimeVAE-NCDE | 0.265 | 0.166 | 0.107 | 0.021 | 0.395 | 0.186 | 0.074 | 0.038 | 0.437 | 0.216 | 0.128 | 0.067 |
| | Diffusion-NCDE | 0.087 | 0.235 | 0.151 | 0.042 | 0.083 | 0.334 | 0.044 | 0.071 | 0.123 | 0.398 | 0.163 | 0.039 |
| | ProFITi | 0.08 | 0.054 | 0.022 | 0.012 | 0.172 | 0.115 | 0.036 | 0.011 | 0.192 | 0.121 | 0.047 | 0.029 |
| | HeTVAE | 3.174 | 2.16 | 0.194 | 0.353 | 1.34 | 2.537 | 1.131 | 1.14 | 6.838 | 2.659 | 2.03 | 3.184 |

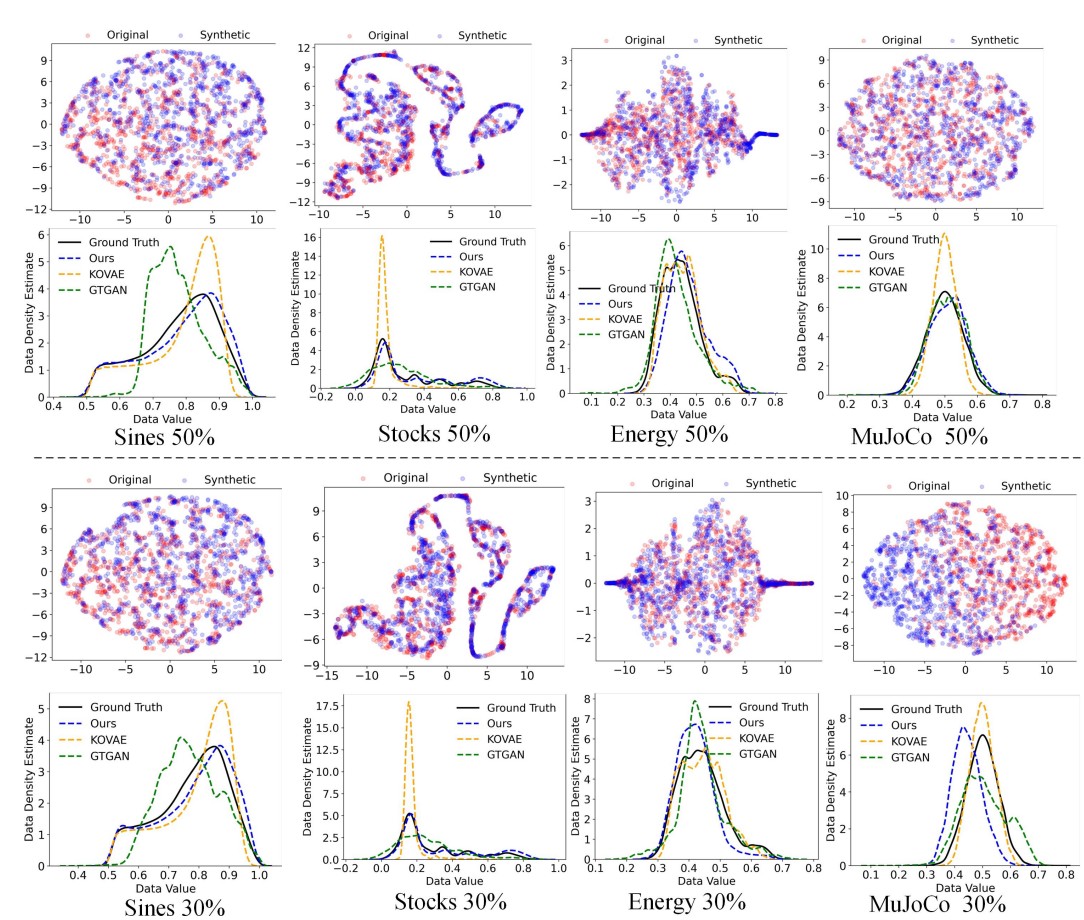

Figure 27: Qualitative evaluations on 30% and 50% missing values using 2D t-SNE plots of generated and real data (top) and probability density functions of real, KoVAE, and GT-GAN distributions (bottom).

Table 14: Continues generation of MOE-Neural CDE on Irregular time series (30%, 50%, and 70% of observations are dropped) on popular datasets with sequence length 12.

|  |  | 30% | | | | 50% | | | | 70% | | | |
|---|---|---|---|---|---|---|---|---|---|---|---|---|---|
|  |  | Sines | Stocks | Energy | MuJoCo | Sines | Stocks | Energy | MuJoCo | Sines | Stocks | Energy | MuJoCo |
| MSE | Ours-$\hat{S}_{MOE}$ | **0.016** | **0.002** | **0.014** | **0.027** | **0.017** | **0.002** | **0.013** | **0.029** | **0.017** | **0.002** | **0.014** | **0.027** |
|  | Ours | 0.030 | 0.004 | 0.014 | 0.025 | 0.031 | 0.005 | 0.014 | 0.027 | 0.034 | 0.004 | 0.014 | 0.026 |
|  | KOVAE-NCDE | 0.070 | 0.013 | 0.03 | 0.028 | 0.070 | 0.012 | 0.028 | 0.029 | 0.07 | 0.011 | 0.027 | 0.031 |
|  | KOVAE | 0.036 | 0.006 | 0.014 | 0.027 | 0.036 | 0.006 | 0.014 | 0.027 | 0.036 | 0.006 | 0.014 | 0.028 |
|  | GTGAN-NCDE | 0.064 | 0.039 | 0.029 | 0.041 | 0.052 | 0.028 | 0.024 | 0.040 | 0.060 | 0.028 | 0.030 | 0.040 |
|  | GTGAN | 0.034 | 0.029 | 0.018 | 0.032 | 0.043 | 0.011 | 0.016 | 0.031 | 0.033 | 0.026 | 0.018 | 0.030 |
| MAE | Ours-$\hat{S}_{MOE}$ | **0.107** | **0.025** | **0.077** | **0.124** | **0.109** | **0.025** | **0.074** | **0.127** | **0.111** | **0.024** | **0.076** | **0.126** |
|  | Ours | 0.140 | 0.034 | 0.077 | 0.124 | 0.143 | 0.035 | 0.075 | 0.128 | 0.149 | 0.033 | 0.077 | 0.127 |
|  | KOVAE-NCDE | 0.190 | 0.085 | 0.137 | 0.132 | 0.190 | 0.083 | 0.134 | 0.134 | 0.190 | 0.079 | 0.128 | 0.141 |
|  | KOVAE | 0.157 | 0.039 | 0.080 | 0.131 | 0.157 | 0.040 | 0.080 | 0.130 | 0.156 | 0.041 | 0.080 | 0.135 |
|  | GTGAN-NCDE | 0.193 | 0.147 | 0.111 | 0.162 | 0.181 | 0.122 | 0.115 | 0.162 | 0.183 | 0.122 | 0.121 | 0.16 |
|  | GTGAN | 0.153 | 0.110 | 0.090 | 0.141 | 0.174 | 0.077 | 0.087 | 0.142 | 0.15 | 0.103 | 0.093 | 0.139 |

Table 15: Continues generation of MOE-Neural CDE on Irregular time series (30%, 50%, and 70% of observations are dropped) on popular datasets with sequence length 24.

| | | 30% | | | | 50% | | | | 70% | | | |
| | | Sines | Stocks | Energy | MuJoCo | Sines | Stocks | Energy | MuJoCo | Sines | Stocks | Energy | MuJoCo |
| --- | --- | --- | --- | --- | --- | --- | --- | --- | --- | --- | --- | --- | --- |
| MSE | Ours-$\hat{S}_{MOE}$ | **0.02** | **0.003** | **0.013** | **0.027** | **0.020** | **0.002** | **0.013** | **0.027** | **0.022** | **0.002** | **0.013** | **0.027** |
| | Ours | 0.055 | 0.007 | 0.013 | 0.030 | 0.055 | 0.006 | 0.013 | 0.029 | 0.059 | 0.006 | 0.014 | 0.029 |
| | KOVAE-NCDE | 0.080 | 0.053 | 0.021 | 0.030 | 0.080 | 0.049 | 0.018 | 0.032 | 0.084 | 0.039 | 0.018 | 0.033 |
| | KOVAE | 0.044 | 0.033 | 0.014 | 0.029 | 0.044 | 0.022 | 0.014 | 0.030 | 0.047 | 0.018 | 0.014 | 0.031 |
| | GTGAN-NCDE | 0.077 | 0.010 | 0.02 | 0.056 | 0.079 | 0.01 | 0.023 | 0.053 | 0.073 | 0.013 | 0.022 | 0.058 |
| | GTGAN | 0.047 | 0.007 | 0.017 | 0.038 | 0.047 | 0.005 | 0.016 | 0.048 | 0.048 | 0.003 | 0.016 | 0.047 |
| MAE | Ours-$\hat{S}_{MOE}$ | **0.106** | **0.025** | **0.074** | **0.131** | **0.107** | **0.025** | **0.072** | **0.129** | **0.109** | **0.025** | **0.074** | **0.131** |
| | Ours | 0.179 | 0.040 | 0.074 | 0.138 | 0.18 | 0.037 | 0.073 | 0.137 | 0.187 | 0.038 | 0.075 | 0.135 |
| | KOVAE-NCDE | 0.191 | 0.152 | 0.109 | 0.140 | 0.192 | 0.144 | 0.096 | 0.143 | 0.199 | 0.135 | 0.096 | 0.146 |
| | KOVAE | 0.166 | 0.119 | 0.08 | 0.137 | 0.167 | 0.079 | 0.08 | 0.139 | 0.176 | 0.071 | 0.080 | 0.143 |
| | GTGAN-NCDE | 0.19 | 0.079 | 0.104 | 0.188 | 0.199 | 0.078 | 0.115 | 0.188 | 0.197 | 0.086 | 0.108 | 0.194 |
| | GTGAN | 0.179 | 0.052 | 0.087 | 0.157 | 0.173 | 0.045 | 0.084 | 0.177 | 0.173 | 0.034 | 0.086 | 0.173 |

Table 16: Continues generation of MOE-Neural CDE on Irregular time series (30%, 50%, and 70% of observations are dropped) on popular datasets with sequence length 36.

| | | 30% | | | | 50% | | | | 70% | | | |
| | | Sines | Stocks | Energy | MuJoCo | Sines | Stocks | Energy | MuJoCo | Sines | Stocks | Energy | MuJoCo |
| --- | --- | --- | --- | --- | --- | --- | --- | --- | --- | --- | --- | --- | --- |
| MSE | Ours-$\hat{S}_{MOE}$ | **0.041** | **0.002** | **0.013** | **0.031** | **0.042** | **0.002** | **0.013** | **0.031** | **0.044** | **0.003** | **0.013** | **0.032** |
| | Ours-NCDE | 0.041 | 0.002 | 0.013 | 0.031 | 0.042 | 0.002 | 0.013 | 0.031 | 0.044 | 0.003 | 0.013 | 0.033 |
| | Ours | 0.063 | 0.005 | 0.013 | 0.038 | 0.064 | 0.005 | 0.014 | 0.035 | 0.068 | 0.005 | 0.014 | 0.036 |
| | KOVAE-NCDE | 0.077 | 0.057 | 0.018 | 0.033 | 0.078 | 0.046 | 0.018 | 0.035 | 0.082 | 0.039 | 0.017 | 0.036 |
| | KOVAE | 0.051 | 0.035 | 0.014 | 0.035 | 0.052 | 0.029 | 0.014 | 0.036 | 0.051 | 0.018 | 0.014 | 0.038 |
| | GTGAN-NCDE | 0.126 | 0.013 | 0.032 | 0.061 | 0.098 | 0.015 | 0.031 | 0.065 | 0.076 | 0.021 | 0.034 | 0.076 |
| | GTGAN | 0.052 | 0.006 | 0.022 | 0.054 | 0.064 | 0.007 | 0.020 | 0.062 | 0.058 | 0.004 | 0.023 | 0.065 |
| MAE | Ours-$\hat{S}_{MOE}$ | **0.162** | **0.026** | **0.074** | **0.133** | **0.161** | **0.027** | **0.074** | **0.135** | **0.163** | **0.026** | **0.075** | **0.137** |
| | Ours | 0.203 | 0.035 | 0.074 | 0.149 | 0.204 | 0.036 | 0.074 | 0.144 | 0.209 | 0.036 | 0.076 | 0.145 |
| | KOVAE-NCDE | 0.214 | 0.159 | 0.096 | 0.146 | 0.218 | 0.141 | 0.099 | 0.149 | 0.225 | 0.133 | 0.091 | 0.151 |
| | KOVAE | 0.186 | 0.123 | 0.080 | 0.150 | 0.191 | 0.111 | 0.081 | 0.151 | 0.193 | 0.072 | 0.081 | 0.155 |
| | GTGAN-NCDE | 0.266 | 0.093 | 0.132 | 0.198 | 0.247 | 0.099 | 0.135 | 0.208 | 0.218 | 0.115 | 0.141 | 0.226 |
| | GTGAN | 0.197 | 0.046 | 0.103 | 0.182 | 0.209 | 0.048 | 0.098 | 0.202 | 0.201 | 0.034 | 0.117 | 0.208 |

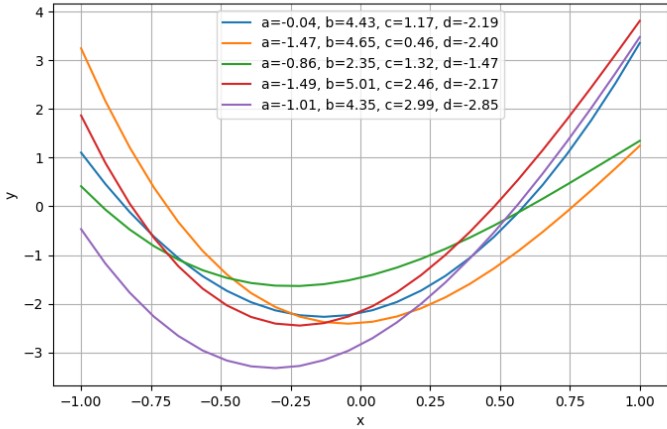

Figure 28: Visualization of cubic polynomial curve data. Due to different coefficients, the polynomial curves display different patterns.

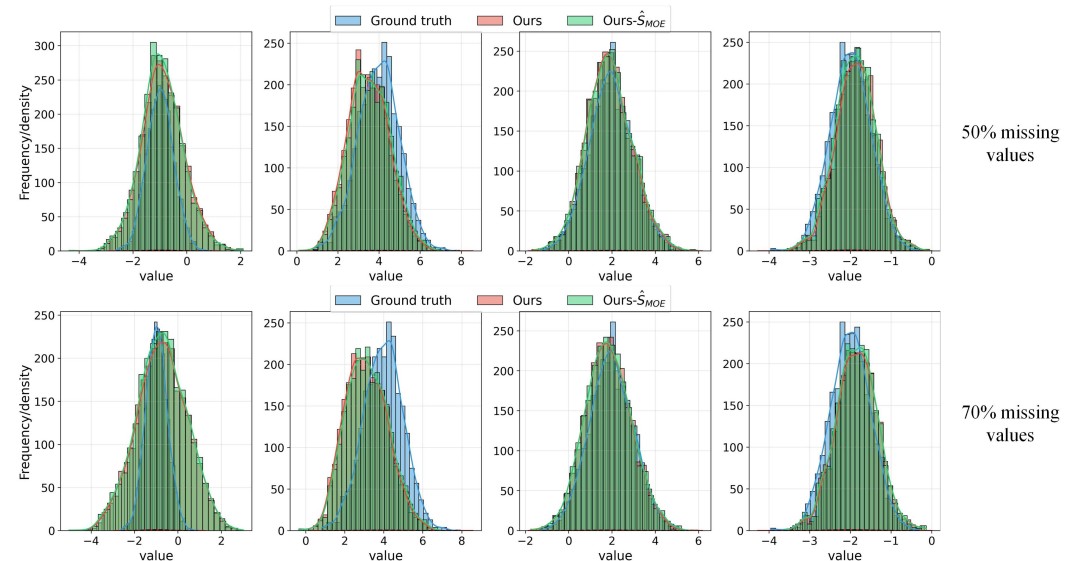

Figure 29: Our method's performance of computing the analysis solution of the cubic polynomial function by using samples with continuous generation (double sequence length than the original one). The four subplots from left to right sequentially represent the coefficients $a$, $b$, $c$, and $d$ of the cubic polynomial.

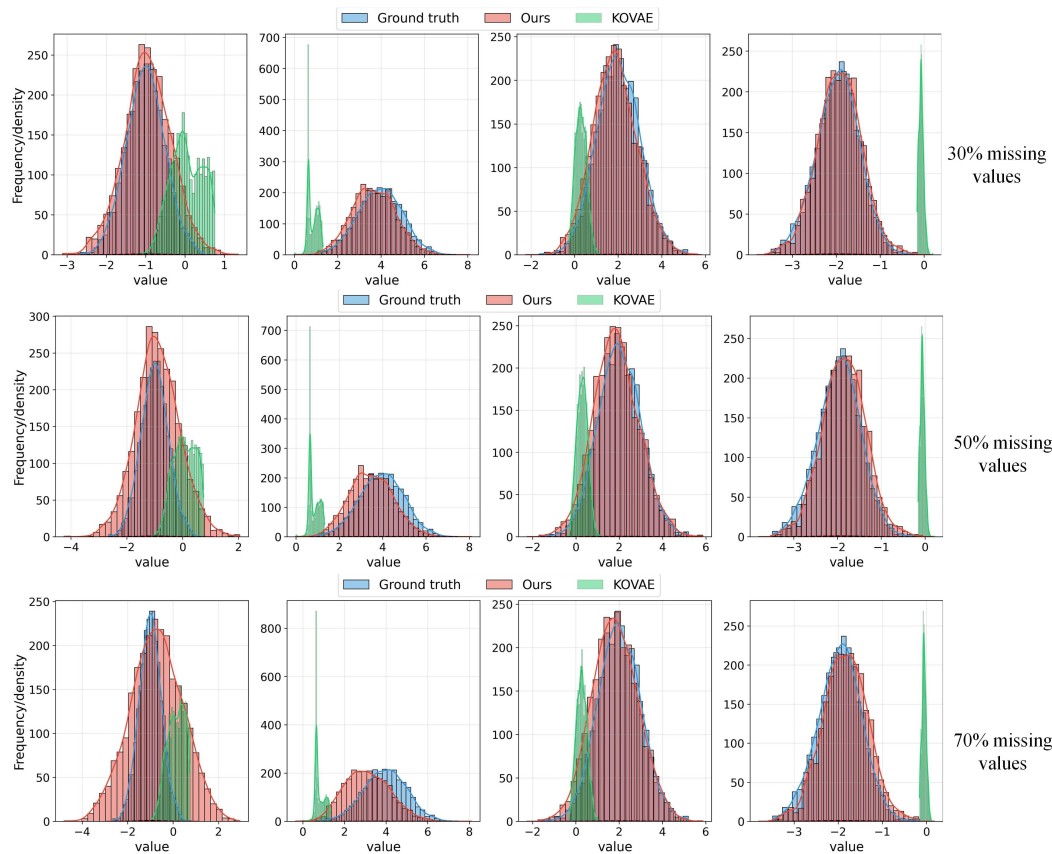

Figure 30: Performance comparison between our method and KOVAE for computing the analysis solution of the cubic polynomial function by using generated samples. The four subplots from left to right sequentially represent the coefficients $a$, $b$, $c$, and $d$ of the cubic polynomial.

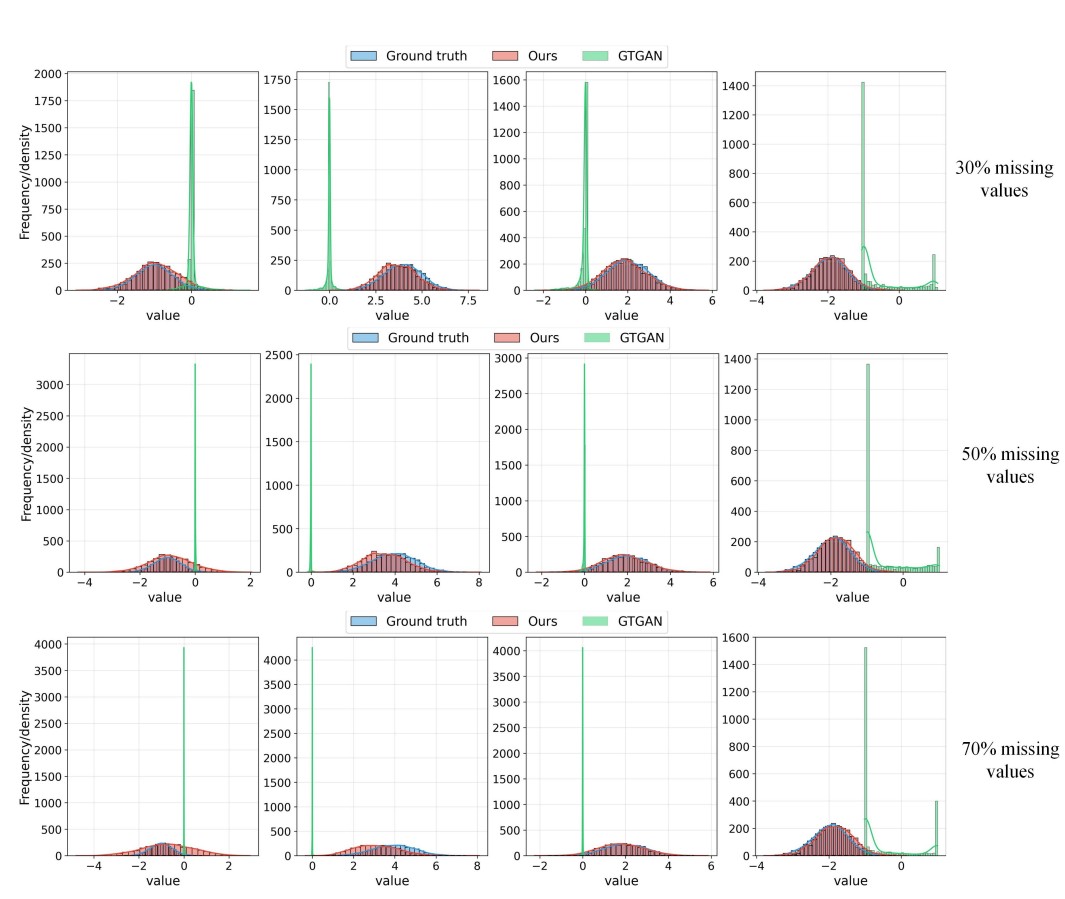

Figure 31: Performance comparison between our method and KOVAE for computing the analysis solution of the cubic polynomial function by using generated samples.

