# OpenReview forum: "Continuous Time Series Generation with Irregular Observations"
_ICLR.cc/2026/Conference — Submitted to ICLR 2026_

### Official Review · Reviewer_gENa · 2025-10-29

**Soundness:** 3
**Presentation:** 2
**Contribution:** 3
**Rating:** 6
**Confidence:** 4

**Summary:**

This paper introduces MN-TSG, a framework for continuous time-series generation with irregularly sampled observations, which extends Neural Controlled Differential Equations (NCDEs) with a mixture-of-experts and diffusion generative models. It proposes a Mixture-of-Experts NCDE (MoE-NCDE) method that can capture varied temporal dynamics via a learnable router for weightedcombination.
It employs a decoupled training strategy by pretraining a channel-wise autoencoder before learning dynamics.
Finally, the model integrates a joint diffusion generator that simultaneously models observations and expert weights.

**Strengths:**

1. The MoE-NCDE extends NCDE approach by incoroporating a MoE method. Standard NCDEs jointly train the encoder, dynamics, and decoder, making optimization unstable. Here, the authors pre-train an autoencoder to help NCDE focuses solely on learning the MoE-dynamics.

2. The model trains a diffusion generator G not only on time-series samples O but also on their expert-weight vectors S.

3. Experiment designs are comprehensive. The results show improved dynamic realism and smoother interpolation compared to NCDE, latent ODE, and imputation-based baselines.

4. The visualization of tsne and density plot helps interpret the generated data, enhancing its interpretability.

**Weaknesses:**

1. The paper use existing NCDE, MoE, decoupled training, and diffusion models. The main novelty lies in combining them with NCDEs rather than introducing new theoretical mechanisms.

2. It is not clear how to select the expert number and how does this hyperparameter impact to the generalization.

3. This paper may incorporate the privacy analysis about whether we can recover the patient records.

**Questions:**

NA

---

> ### Author Response · Authors · 2025-11-21
> **Response to Weakness 1 to 2**
>
> **Weakness 1:** The paper uses existing NCDE, MoE, decoupled training, and diffusion models. The main novelty lies in combining them with NCDEs rather than introducing new theoretical mechanisms.
>
> **Response to weakness 1:** We sincerely appreciate the reviewer’s concern about our method builds upon existing components—including NCDEs, MoEs, decoupled training strategies, and diffusion models—we would like to emphasize that the contribution of our work does not lie in simply combining these modules, but rather in introducing task-specific innovations that address challenges unique to irregular-to-continuous time series generation. These innovations reshape how the components interact and collectively solve a previously unexplored problem.
>
> **First**, our adaptation of MoE to NCDE is novel in both purpose and mechanism. Unlike conventional MoE applications in NLP or computer vision, we explicitly design the MoE dynamics to capture heterogeneous temporal behaviors (e.g., diverse ECG rhythms across patients with different diseases) that standard NCDEs struggle to model. This is enabled by a task-specific router that maps irregular time-series characteristics—such as missing-value patterns and local volatility—into expert weights, enabling dynamic specialization across diverse temporal dynamics. Existing MoE or NCDE research has not explored this direction for time series generation. Furthermore, MoE allows us to decouple the modeling of temporal dynamics into expert-weight vectors that can be learned independently and generated through diffusion, ensuring that every newly generated sample receives a tailored dynamic function.
>
> **Second**, we introduce a decoupled training strategy that addresses a known limitation of NCDE optimization. Prior NCDE-based models optimize both the initial latent mapping $z(0)$ and the dynamic function jointly, often leading to optimization conflicts (e.g., reconstruction dominating dynamics learning). Our pre-trained channel-wise autoencoder provides a clean and stable initialization for $z(0)$ and reconstruction, allowing NCDE to focus specifically on learning the underlying temporal dynamics. This design is essential to making MoE-NCDE not only feasible but also significantly more effective.
>
> **Finally**, we extend diffusion models beyond standard TSG generation. Existing diffusion-based TSG works only model the distribution of observed values. We propose a joint-generation mechanism that simultaneously generates observations and MoE weights, ensuring that each synthesized time series is coupled with its own specialized dynamic function for downstream continuous refinement. This closes the methodological gap between “regular sequence generation” and “continuous generation,” a capability not previously addressed in the literature.
>
> We will more clearly present the core innovations of our method. We propose the MNTSG framework for the novel task of irregular-to-regular-to-continuous generation. In MNTSG, we improve NCDE so it can be seamlessly coupled with any state-of-the-art regular time-series generator to learn their joint distribution. Afterwards, MNTSG injects the newly sampled NCDE-MOE weights into the pre-trained NCDE, enabling fine-grained and continuous generation for each newly generated TS data.
>
> The above analysis has been added to **line 756** section D MORE DISCUSSIONS ABOUT CONTRIBUTIONS.
>
>
>
> **Weakness 2:** It is not clear how to select the expert number and how does this hyperparameter impact to the generalization.
>
> **Response to Weakness  2:** Regarding how to select the expert number, we recommend determining the number of experts based on two complementary principles:
>
> (1) Data-driven temporal-pattern analysis. We can perform clustering on dynamic features such as autocorrelation and volatility to estimate how many distinct temporal modes exist in a dataset. For our datasets (e.g., ECG with 2–5 disease classes, multivariate sine/stock datasets with periodic and aperiodic sub-patterns), four experts are sufficient to cover the dominant dynamic types without unnecessary model complexity.
>
> (2) Empirical validation. As shown in Figure 7 of the revised manuscript, we evaluate expert numbers ranging from 1 to 9. Performance consistently improves up to 4 experts, and then the benefit becomes negligible—adding more experts increases computation while offering no obvious benefit. Therefore, we select 4 experts as a balanced and generalizable default for TSG tasks.
>
> **Impact on generalization.** Figure 7 further indicates that generalization performance is not highly sensitive to the expert number. While four experts provide the best overall results, using more experts does not significantly degrade performance, though it increases computational cost. Thus, the model’s generalization ability is stable with respect to this hyperparameter.
>
> We have added the above analysis in **lines 1133 to 1179** in the revised paper.

---

> ### Author Response · Authors · 2025-11-21
> **Response to Weakness 3**
>
> **Weakness 3:** This paper may incorporate the privacy analysis about whether we can recover the patient records.
>
> **Response to Weakness  3:** We thank the reviewer for highlighting this important consideration. Following your valuable suggestion, we have added a privacy evaluation to the revised manuscript. Specifically, we adopt the Membership Inference Risk (MIR) metric [1] to assess whether synthetic data generated by various models reveal information about the real data. The results are shown in **Table 1** here. While our method achieves superior fidelity in terms of DS, KL and MDD metrics, the MIR results also show that our model exhibits the lowest privacy leakage risk among all baselines. This indicates that our jointly generated dynamics and data not only improve quality but also show lower privacy leakage risk.
>
> We sincerely appreciate the reviewer’s constructive feedback, which has helped improve the completeness and practical relevance of our work. We have added the analysis in **lines 320 to 323** and **lines 345 to 347** in the revised paper.
>
>
> **Table 1: Privacy-leakage risk of different methods on medical datasets, evaluated using the Membership Inference Risk (MIR) metric. A higher MIR score indicates that synthetic data more easily reveals information about real samples, implying higher privacy risk. Thus, lower MIR values are preferred.**
>
> |    | ECG200-30%|ECG5K-30%| ECGFD-30%|TLECG-30%|ECG200-50%|ECG5K-50%| ECGFD-50%|TLECG-50%|ECG200-70%|ECG5K-70%| ECGFD-70%|TLECG-70%|Average|
> |:-:|:-:|:-:|:-:|:----:|:-:|:-:|:-:|:----:|:-:|:-:|:-:|:----:|:----:|
> |Ours|**0.6826** |0.6944 |**0.6734** |**0.6671** |**0.6671** |**0.6676** |**0.6866** |**0.7302** |0.697 |**0.6866** |**0.6765** |**0.6866** |**0.6846**|
> |KoVAE|0.6873 |**0.6849** |0.7194 |0.6882 |0.6761 |0.6873 |0.8846 |0.8679 |0.7931 |0.7077 |0.697 |0.697 |0.7325|
> |GT-GAN|0.6993 |0.8772 |0.9709 |0.9794 |0.7698 |0.7628 |0.9583 |0.9787 |**0.6765** |0.7541 |0.807 |0.7667 |0.8334|
> |TimeGAN|0.7463 |0.7463 |0.7463 |0.6817 |0.7163 |0.8482 |0.807 |0.807 |0.807 |0.7667 |0.7419 |0.807 |0.7685|
> |Diffusion|0.7067 |0.7067 |0.6873 |0.6693 |0.6698 |0.6698 |0.7541 |0.7797 |0.8364 |0.7419 |0.7188 |0.7302 |0.7226|
> |TimeVAE|0.7194 |0.7067 |0.6897 |0.6676 |0.6676 |0.6698 |0.7541 |0.7541 |0.7797 |0.7419 |0.7419 |0.7419 |0.7195|
>
> [1] Reliable Generation of Privacy-preserving Synthetic Electronic Health Record Time Series via Diffusion Models. Journal of the American Medical Informatics Association. 2024.

---

> > ### Comment · Reviewer_gENa · 2025-11-23
> > **Feedback**
> >
> > Thanks for your response.  I would like to keep my score of 6.

---

> ### Author Response · Authors · 2025-11-23
>
> Thank you for increasing your confidence score from 4 to 5 and for maintaining a positive overall assessment of our work. We sincerely appreciate your time and thoughtful feedback.

---

### Official Review · Reviewer_YdVt · 2025-10-29

**Soundness:** 2
**Presentation:** 3
**Contribution:** 2
**Rating:** 4
**Confidence:** 4

**Summary:**

This paper proposes MoE-NCDE, a framework combining a mixture-of-experts neural controlled differential equation (CDE) with a diffusion model to generate continuous-time trajectories from irregularly sampled observations. The CDE component is used to construct the continuous dynamics from irregular observations, while the diffusion model learns the joint distribution of the regular observations and expert weights.

**Strengths:**

1. The proposed method integrates a continuous-time dynamic layer (MoE-NCDE) on top of discrete diffusion generation, helps address diffusion’s discretization constraint.
2. The paper includes multiple datasets, metrics, and ablation studies, with reproducible implementation details given.

**Weaknesses:**

1. Lack of justification for the CDE choice:

1.1 The CDE component requires spline-interpolated control paths to perform integration, meaning it cannot generate trajectories by itself and must rely on the diffusion model. However, continuous-time formuations such as ODEs or SDEs can already generate continuous trajectories directly from initial conditions, fully addressing the irregular-to-continuous generation task stated in the title. No comparison with these alternatives Is provided.

1.2 The spline-interpolation imposes a strong smoothness prior on the control path, which conflicts with the target domains (e.g. healthcare and finance) where real processes often involve abrupt events or shocks. As a result, the generated trajectories may appear smooth but fail to reflect the irregular and event-driven nature of such data.

2. The proposed MoE-NCDE introduces multiple experts, but without explicit constraints or diversity objectives, there’s no guarantee that different experts learn distinct dynamic behaviors. Each expert is optimized jointly under the same loss, receiving similar gradients through soft gating. Consequently, the MoE may effectively function as a single, large network, and the reported performance gains could primarily reflect increased model capacity rather than meaningful expert specialization.
3. In Appendix B.2, the generated trajectories appear jagged, while there’s no ground-truth data for the interpolated intervals, so the additional dynamics cannot be verified and may simply be artifacts of the model’s interpolation.

**Questions:**

Please see the Weaknesses part.

---

> ### Author Response · Authors · 2025-11-21
> **Response to Weakness 1**
>
> **Weaknesses (W) 1: Lack of justification for the CDE choice**
>
> **W 1_1:** The CDE component requires spline-interpolated control paths to perform integration, meaning it cannot generate trajectories by itself and must rely on the diffusion model. However, continuous-time formuations such as ODEs or SDEs can already generate continuous trajectories directly from initial conditions, fully addressing the irregular-to-continuous generation task stated in the title. No comparison with these alternatives Is provided.
>
> **Response to W 1_1:** We sincerely thank the reviewer for raising this insightful concern. Our choice of CDE is motivated by enabling MNTSG to serve as a decoupled framework that flexibly benefits from the generative capabilities of any TSG model in the continuous generation stage. Before generating new trajectories, what the framework fundamentally needs is an interpolation module capable of accurately imputing irregular observations based on their underlying temporal dynamics. After generating new data, MNTSG also requires an interpolation step to perform continuous generation, using both the generated temporal dynamics and the corresponding interpolation parameters. From the perspective of interpolation, ODE-based models are intrinsically less expressive than CDE.
>
> We clarify the advantages of CDE over ODE/SDE from both theoretical and empirical perspectives.
>
> **Theoretical perspective.** The NCDE formula is:
>
> \begin{equation}
>     z(t) = z(0) + \int_0^t f_\theta(z(s)) dX(s), (1)
> \end{equation}
>
> The ODE formula is :
> \begin{equation}
>     z(t) = z(0) + \int_0^t f_\theta(z(s)) ds, (2)
> \end{equation}
>
> In NCDE, the control path $X(s)$ is constructed directly from the irregular observations. This observation-driven control path, together with the learned dynamics, guides the evolution of $z(t)$ throughout the entire time horizon. Conditioning on irregular inputs greatly facilitates learning meaningful temporal representations and leads to more accurate interpolations.
>
> In contrast, ODE/SDE models evolve the latent state solely from $z(0)$ based on learned dynamics, without direct conditioning on the observed values. Consequently, the latent trajectory may drift away from the true data distribution when interpolation is required.
>
> This observation-driven advantage of NCDE has also been validated by prior works such as KoVAE and GT-GAN. Motivated by this, we adopt NCDE and further enhance it with a mixture-of-experts design to better capture diverse temporal dynamics.
>
> Meanwhile, SDE introduces only a stochastic term but still lacks a control-path mechanism. Thus, for interpolation of irregular observations, control-path–guided CDE is a more appropriate choice.
>
> **Experimental perspective.** We have in fact compared with ODE-based methods. GT-GAN incorporates ODE-RNN and CTFP[1], where the latter borrows concepts from SDEs and employs a Wiener process to introduce stochasticity. Ablation studies of GT-GAN consistently show that GT-GAN outperforms the two ODE-based methods. Additionally, KoVAE is reported to perform even better than GT-GAN, and our experiments include comparisons against both GT-GAN and KoVAE.
>
> *[1] Modeling Continuous Stochastic Processes with Dynamic Normalizing Flows. NeurIPS 2020.*
>
> **W 1_2:** The spline-interpolation imposes a strong smoothness prior on the control path, which conflicts with the target domains (e.g. healthcare and finance) where real processes often involve abrupt events or shocks. As a result, the generated trajectories may appear smooth but fail to reflect the irregular and event-driven nature of such data.
>
> **Response to W 1_2:** We appreciate the reviewer for highlighting this important concern. Although the control path $X(s)$ is smooth due to spline interpolation, the NCDE dynamics do not impose smoothness on the final generated trajectory. As shown in Eq. (1), **the control path $X(s)$ is smooth, but the evolution of $z(t)$ is governed by the nonlinear neural network $f_\theta(z(s))$**.
> Thus, even with a smooth control path, the latent trajectory $z(t)$ can evolve in a highly non-smooth and expressive manner.  **Moreover, the mixture-of-experts component further facilitates the modeling of non-smooth and highly diverse temporal patterns.**
>
> To validate this, we added visualizations of sequences generated solely by NCDE (**Appendix Figures 14–16** in the revised manuscript). These results show:
>
> **(1)** For volatile datasets such as energy (**Figure 14**) and stock (**Figure 15**), NCDE generates naturally fluctuating trajectories, without excessive smoothing.
>
> **(2)** For intrinsically smooth data (e.g., sine in **Figure 16**), NCDE produces smooth trajectories.
>
> Overall, while the control path is smooth, the nonlinear MOE–NCDE dynamics allow the generated sequences to exhibit fluctuating temporal behaviors, depending entirely on the underlying temporal dynamics. The above analysis has been added to **lines 1078 to 1079 and lines 1112 to 1123**.

---

> ### Author Response · Authors · 2025-11-21
> **Response to Weakness 2 to 3**
>
> **W 2:** The proposed MoE-NCDE introduces multiple experts, but without explicit constraints or diversity objectives, there’s no guarantee that different experts learn distinct dynamic behaviors. Each expert is optimized jointly under the same loss, receiving similar gradients through soft gating. Consequently, the MoE may effectively function as a single, large network, and the reported performance gains could primarily reflect increased model capacity rather than meaningful expert specialization.
>
> **Response to W 2:** We sincerely appreciate the reviewer’s insightful concern. To address it, we conducted three analyses to examine whether the experts in MoE-NCDE indeed learn different behaviors rather than merely increasing model capacity.
>
> **For quantitative evaluations**, first, **Table 1** here compares NCDE performance between **(i)** four experts (**one MLP and two linear layers per expert, eight layers in total**) and **(ii)** a single expert with four MLPs stacked (**also eight layers in total**). **We ensure that 1-expert-NCDE has the same number of parameters as 4-experts-NCDE.** The mixture of experts achieves substantially better performance, **indicating that the benefits arise from the expert decomposition rather than from simply enlarging the model**. We have added the analysis in **lines 463 to 468** in the revised paper.
>
> Second, **Table 2** here reports the average weight assigned to each expert across all samples in different datasets. The weights are not sparse, and different datasets both tend to prefer different experts, demonstrating that the expert structure is indeed utilized and optimized in a meaningful manner rather than collapsing to a single expert. We have added the analysis in **lines 839 to 843** in the revised paper.
>
> Finally, for **qualitative evaluation**, we visualize specific temporal dynamics learned by MoE-NCDE.
> In Figure 6 of the revised manuscript, we observe that the temporal patterns learned by different experts exhibit local or global diversity, while the combined patterns produced by all experts retain characteristics of those learned individually. The cases on the energy dataset in Appendix Figure 10 also shows similar conclusion. This indicates that the experts can learn different patterns, achieving the decomposed learning of complex dynamics and demonstrating the effectiveness of the multi-expert design. We have added the analysis in **lines 479 to 481** in the revised paper.
>
> **W 3:** In Appendix B.2, the generated trajectories appear jagged, while there’s no ground-truth data for the interpolated intervals, so the additional dynamics cannot be verified and may simply be artifacts of the model’s interpolation.
>
> **Response to W 3:** We thank the reviewer for raising this important point. The saw-tooth–like interpolation observed in Appendix B.2 represents only one of the possible generation patterns. In the revised manuscript, we provide additional visualizations in **Appendix Figures 11–13**. **These figures show that the continuous trajectories generated by NCDE vary according to the learned dynamics and the MOE weights produced by the diffusion model**. During continuous generation, **NCDE exhibits diverse behaviors—including stable, fluctuating, and non-smooth patterns**. The saw-tooth pattern is not representative of all cases but rather one instance of the broader range of dynamics the model can generate. We have added the analysis in **lines 1126 to 1132** in the revised paper.
>
>
> **Table 1:  Comparison between four experts (two linear layers per expert) and a single expert with stacked parameters (eight linear layers).**
> |    | ECG200-30%|ECG5K-30%| ECGFD-30%|ECG200-50%|ECG5K-50%| ECGFD-50%|ECG200-70%|ECG5K-70%| ECGFD-70%
> |:-:|:-:|:-:|:-:|:----:|:-:|:-:|:-:|:----:|:-:|
> |4-experts-DS|**0.2**| **0.205**| **0.09**| **0.405**| **0.315**| **0.12**| **0.34**| **0.288**| **0.13**|
> |1-expert-DS|0.328 |0.364 |0.13 |0.44 |0.389 |0.23 |0.392 |0.412 |0.21|
> |4-experts-MDD|**0.211**| **0.062**| **0.424**| **0.237**| **0.079**| **0.529**| **0.246**| **0.095**| **0.613**|
> |1-expert-MDD|0.288 |0.12 |0.694 |0.321 |0.152 |0.831 |0.314 |0.165 |0.837|
> |4-experts-KL|**0.052**| **0.009**| **0.055**| **0.107**| **0.014**| **0.05**| **0.098**| **0.02**| **0.049**|
> |1-expert-KL|0.136 |0.056 |0.065 |0.236 |0.098 |0.189 |0.224 |0.123 |0.165|
>
>
> **Table 2:  Average expert weights learned by NCDE across all samples from different datasets.**
> |    | Expert 1|Expert 2| Expert 3|Expert 4|
> |:-:|:-:|:-:|:-:|:----:|
> |Sines|**0.31384414** |0.2231202 | 0.26881662 |0.19421977|
> |Stocks|0.19200034 |0.21767974 |**0.32304975** |0.2672697|
> |Energy|0.2413441  |**0.31066164** |0.2065838  |0.24141027|
> |MuJoCo|0.18345672 |0.241069   |0.2046078  |**0.370866**|
> |ECG200|0.21229412 |**0.44273788** |0.20304906 |0.14191894|
> |ECG5K|0.21999635 |0.182932   |**0.3610456**  |0.2360259|
> |ECGFD|0.17836495 |**0.31938562** |0.2739047  |0.2283447|
> |TLECG|0.19264506 |0.2577268  |**0.3210036**  |0.2286246|

---

### Official Review · Reviewer_5Rc9 · 2025-11-01

**Soundness:** 3
**Presentation:** 4
**Contribution:** 3
**Rating:** 8
**Confidence:** 4

**Summary:**

The paper tackles the problem of continuous time series generation from irregularly observed data, a scenario highly relevant to domains such as healthcare and finance where data points often arrive at uneven intervals. The authors introduce MNTSG, a framework that innovates on Neural Controlled Differential Equations (NCDEs) by introducing a mixture-of-experts (MOE) dynamic function architecture and a decoupled training strategy. This structure is further integrated with diffusion-based generative models to jointly model time series samples and their dynamic function assignments. Extensive experiments across public, medical, and synthetic datasets show the proposed approach outperforming established baselines, with both quantitative and qualitative evaluation.

**Strengths:**

1. The motivation and research gap is clearly articulated.
2. The proposed method design is clearly motivated.
3. The paper presented strong and comprehensive evaluation including various qualitative and quantitive aspects.

**Weaknesses:**

1. The motivation choose MoE to learn the complex various temporal dynamics is not well described. Is there any hidden correlations? Or do you observe any specific temporal dynamics can be learnt from MoE?
2. About the generated data evaluation, although current format is quite strong and follow previous standard evaluation setting, why not apply to real irregular downstream tasks to evaluate the quality of the generated data?
3. The paper uses channel-wise autoencoder for reconstruction task. What's the trade-off between channel-wise versus channel-correlated stucture?
4. The overall framework still looks like assembly of existing modules, the key innovation of proposed methods is somehow hidden.

**Questions:**

see above

---

> ### Author Response · Authors · 2025-11-21
> **Response to Weakness 1**
>
> Thank you very much for dedicating your valuable time to carefully review our manuscript and provide constructive comments. Your suggestions have helped us clarify the research logic and highlight the core contributions of our work. We address each of your questions in turn below.
>
> **Weakness 1:** The motivation choose MoE to learn the complex various temporal dynamics is not well described. Is there any hidden correlations? Or do you observe any specific temporal dynamics can be learnt from MoE?
>
> **Response to Weakness 1**: Real-world temporal data (e.g., medical ECG signals, financial stock data) often exhibits substantial heterogeneity in dynamic patterns. For example, ECG signals from different patients may include stable rhythms, sudden fluctuations, or periodic anomalies. A single dynamic function, as used in standard NCDE, struggles to simultaneously capture global regularity and local specificity.
>
> The Mixture-of-Experts (MoE) framework addresses this limitation through expert specialization: each expert focuses on learning a specific type of sub-dynamic (e.g., stable trends, impulsive fluctuations, periodic changes), while the router dynamically assigns weights based on local input features. This design allows MoE-NCDE to more accurately capture complex, heterogeneous temporal patterns than standard NCDE.
>
> **Regarding the specific temporal dynamics learned by MoE:** As shown in **Figures 6 and 10** of the revised manuscript, we have added visualization experiments about temporal patterns learned by individual experts and by all experts jointly.
> We observe that different experts capture distinct local or global patterns, while the combined patterns produced by all experts retain characteristics of those learned individually. This illustrates a strategy of decomposing complex temporal dynamics for separate learning and then integrating them, which substantially enhances NCDE’s ability to model diverse temporal behaviors. The above analysis has been added to **lines 479 to 481** of the revised manuscript.
>
> **Quantitative analysis further demonstrates the effectiveness of MoE from three perspectives:**
>
> **(1)** ECG datasets in **Table 4** of the manuscript are collected from patients with different diseases, exhibiting pronounced temporal heterogeneity. Removing MoE results in a clear performance drop, confirming the importance of multiple experts.
>
> **(2)** **Figure 5** of the manuscript presents experiments on five manually constructed synthetic signals with distinct temporal patterns. When NCDE is trained without MoE, the generated data exhibits noticeable distributional shifts, highlighting MoE’s role in capturing diverse dynamics.
>
> **(3)** Additional experiments confirm the benefit of cooperation among multiple experts. **Table 1** here compares NCDE performance between (i) four experts (each with one MLP and two linear layers, eight layers in total) and (ii) a single expert with four MLPs stacked (same eight layers). **In other words, we ensure that 4-experts-NCDE has the same number of parameters as 1-expert-NCDE.** The mixture-of-experts approach significantly outperforms the single expert with stacked layers, demonstrating that having mixture-of-experts enables better learning of diverse temporal dynamics. The above analysis has been added to **lines 463 to 468** of the revised manuscript.
>
> **Table 1:  Comparison between four experts (each with 2 linear layers) and a single expert with stacked parameters (8 linear layers).**
>
> |    | ECG200-30%|ECG5K-30%| ECGFD-30%|ECG200-50%|ECG5K-50%| ECGFD-50%|ECG200-70%|ECG5K-70%| ECGFD-70%
> |:-:|:-:|:-:|:-:|:----:|:-:|:-:|:-:|:----:|:-:|
> |4-experts-DS|**0.2**| **0.205**| **0.09**| **0.405**| **0.315**| **0.12**| **0.34**| **0.288**| **0.13**|
> |1-expert-DS|0.328 |0.364 |0.13 |0.44 |0.389 |0.23 |0.392 |0.412 |0.21|
> |4-experts-MDD|**0.211**| **0.062**| **0.424**| **0.237**| **0.079**| **0.529**| **0.246**| **0.095**| **0.613**|
> |1-expert-MDD|0.288 |0.12 |0.694 |0.321 |0.152 |0.831 |0.314 |0.165 |0.837|
> |4-experts-KL|**0.052**| **0.009**| **0.055**| **0.107**| **0.014**| **0.05**| **0.098**| **0.02**| **0.049**|
> |1-expert-KL|0.136 |0.056 |0.065 |0.236 |0.098 |0.189 |0.224 |0.123 |0.165|

---

> > ### Author Response · Authors · 2025-11-21
> > **Response to Weakness 2 to 4**
> >
> > **Weakness 2:** About the generated data evaluation, although current format is quite strong and follow previous standard evaluation setting, why not apply to real irregular downstream tasks to evaluate the quality of the generated data?
> >
> > **Response to Weakness 2**:  We thank the reviewer for this valuable suggestion. The main challenge in real-world settings is that the missing values have no ground truth, making it difficult to objectively evaluate generation quality. Therefore, following prior work [1,2], randomly discarding values from regularly sampled data has become a standard and effective evaluation strategy.
> > We appreciate the reviewer’s insight, and we note that designing metrics for real-world missing data evaluation—where ground truth is unavailable—is an important direction for future exploration.
> >
> > *[1] Generative modeling of regular and irregular time series data via koopman VAE.*
> >
> > *[2] GT-GAN: general purpose time series synthesis with generative adversarial networks.*
> >
> > **Weakness 3:** The paper uses channel-wise autoencoder for reconstruction task. What's the trade-off between channel-wise versus channel-correlated stucture?
> >
> > **Response to Weakness 3**: We thank the reviewer for this insightful comment. In the current work, we focus on channel-wise reconstruction and have not explicitly modeled cross-channel correlations. However, we believe these two directions are complementary—cross-channel interaction can be incorporated to enhance channel-wise reconstruction, enabling richer representation learning and potentially improving overall performance.
> > Our approach can be seamlessly integrated with representation-learning methods that operate along the channel dimension. For example, self-attention could first be applied across channels before performing channel-wise reconstruction.
> >
> > **Weakness 4:** The overall framework still looks like an assembly of existing modules, the key innovation of proposed methods is somehow hidden.
> >
> > **Response to Weakness 4**: Thank you for your valuable advice. we would like to clarify that the core novelty of our method lies not in "simple combination" but in synergistic innovations tailored to the unaddressed challenge of irregular-to-continuous time series generation, which requires rethinking how these components interact to solve domain-specific pain points.
> >
> > **First**, our adaptation of MoE to NCDE is novel in both purpose and mechanism. Unlike conventional MoE applications in NLP or computer vision, we explicitly design the MoE dynamics to capture heterogeneous temporal behaviors (e.g., diverse ECG rhythms across patients with different diseases) that standard NCDEs struggle to model. This is enabled by a task-specific router that maps irregular time-series characteristics—such as missing-value patterns and local volatility—into expert weights, enabling dynamic specialization across diverse temporal dynamics. Existing MoE or NCDE research has not explored this direction for time series generation. Furthermore, MoE allows us to decouple the modeling of temporal dynamics into expert-weight vectors that can be learned independently and generated through diffusion, ensuring that every newly generated sample receives a tailored dynamic function.
> >
> > **Second**, we introduce a decoupled training strategy that addresses a known limitation of NCDE optimization. Prior NCDE-based models optimize both the initial latent mapping $z(0)$ and the dynamic function jointly, often leading to optimization conflicts (e.g., reconstruction dominating dynamics learning). Our pre-trained channel-wise autoencoder provides a clean and stable initialization for $z(0)$ and reconstruction, allowing NCDE to focus specifically on learning the underlying temporal dynamics. This design is essential to making MoE-NCDE not only feasible but also significantly more effective.
> >
> > **Finally**, we extend diffusion models beyond standard TSG generation. Existing diffusion-based TSG works only model the distribution of observed values. We propose a joint-generation mechanism that simultaneously generates observations and MoE weights, ensuring that each synthesized time series is coupled with its own specialized dynamic function for downstream continuous refinement. This closes the methodological gap between “regular sequence generation” and “continuous generation,” a capability not previously addressed in the literature.
> >
> > We will more clearly present the core innovations of our method. We propose the MNTSG framework for the novel task of irregular-to-regular-to-continuous generation. In MNTSG, we improve NCDE so it can be seamlessly coupled with any state-of-the-art regular time-series generator to learn their joint distribution. Afterwards, MNTSG injects the newly sampled NCDE-MOE weights into the pre-trained NCDE, enabling fine-grained and continuous generation for each newly generated TS data.
> >
> > The above analysis has been added to **line 756** section D MORE DISCUSSIONS ABOUT CONTRIBUTIONS.

---

### Official Review · Reviewer_UQbt · 2025-11-06

**Soundness:** 2
**Presentation:** 1
**Contribution:** 2
**Rating:** 2
**Confidence:** 4

**Summary:**

The paper proposes a new model called MOE-NCDE for generating continuous time series from observations over an irregular time grid. MOD-NCDE consists of three main components: 1) a Mixture of Neural Controlled Differential Equations (NCDE), 2) a routing function to weight the experts, and 3) a diffusion model. The authors propose a decoupled training mechanism where the vector field of NCDE is trained, freezing the weights of the pretrained reconstruction model. MOE-NCDE is used to convert irregular time series into regular time series, and then a diffusion model is trained on the regular time series jointly with the MoE weights. Experimental results on various generation and prediction tasks demonstrate that the proposed model has superior performance.

**Strengths:**

1. The authors attempted to address an important task of generating continuous time series from Irregular data. It has wide applications in interpolation, where the distribution of the missing data can be computed.

2. Combining multiple components, such as MoE, NCDE, and Diffusion Models, is new.

3. Experimental results on multiple datasets for various tasks demonstrate that the proposed model is superior compared to the baseline models.

**Weaknesses:**

1. While the paper is understandable at a high level, many components are understated.

i) A clear problem statement is missing. It is not clear what exactly the task is aimed at: probabilistic interpolation, probabilistic forecasting, or purely generating new sequences.

ii) An explanation of the pretraining part for computing $z(0)$ and the reconstruction function is not provided.

iii) The training process of the MOE router $\mathcal{R}$ is not described.

2. Several related works are missing. While the main comparisons are with NCDE-based models, other works [1,2] can also perform the generation task. Both models can predict the distribution of observations at any time point given irregularly sampled data.

[1] Shukla et al., "Heteroscedastic Temporal Variational Autoencoder for Irregularly Sampled Time Series," ICLR 2022

[2] Yalavarthi et al., "Probabilistic Forecasting of Irregularly Sampled Time Series with Missing Values via Conditional Normalizing Flows," AAAI 2025

**Questions:**

1. What exactly are the channel-wise encoder and decoder? Given that the notation on p. 3 suggests univariate time series, how does “channel-wise” apply here?

2. In line 7 of Algorithm 1, is only one expert being used? It is unclear how the top-K experts are utilized, and Eq. 3 does not clarify this.

3. The model seems to work only for univariate irregular time series. Is there a way to extend it to multivariate series?

4. How does the diffusion model ensure that the generated ranks for the router are positive and sum to 1?

---

> ### Author Response · Authors · 2025-11-21
> **Response to Weakness 1**
>
> We sincerely thank the reviewer for these insightful comments. Below, we respond to each point in detail and have incorporated all corresponding revisions into the manuscript.
>
> **Weakness 1:** While the paper is understandable at a high level, many components are understated.
>
> **i)  A clear problem statement is missing. It is not clear what exactly the task is aimed at: probabilistic interpolation, probabilistic forecasting, or purely generating new sequences.**
>
> **Response to i):** We appreciate the reviewer’s observation. Our task focuses purely on generating new sequences. We have revised the Problem Definition section on p.3 to clarify this point and ensure the objective is clearly stated. **Please refer to line 158 to line 164.**
>
> **ii) An explanation of the pre-training part for computing $z(0)$ and the reconstruction function is not provided.**
>
> **Response to ii):** Thank you for pointing this out. While part of this procedure was mentioned in the **original manuscript** (lines 210–215), we agree that a more detailed explanation would improve readability. We have now added an explanation of $z(0)$ and the reconstruction function at **lines 174 to 177**. A dedicated Algorithm 4 is added in the **Appendix lines 728 to 741**. A concise description is provided below.
>
>  **Explanation of pre-training part**:  We pre-train an MLP-based encoder ($f_{CE}$)–decoder ($f_{CD}$) pair on the original irregular data by performing channel-wise encoding and reconstruction. All non-channel dimensions are flattened, and the model is trained via MSE loss to minimize the reconstruction error. Since the input and output dimensions of $f_{CE}$ and $f_{CD}$ match the number of time-series variables (channels), they can be directly used to map channels or hidden states within NCDE. The full channel-wise autoencoder procedure is now given as **Algorithm 4** in the revised manuscript.
>
>
> **Explanation of z(0)**: The latent initial state $z(0)$ represents the encoded hidden representation of the time series at $t_0$. The latent trajectory follows the controlled differential equation:
>
> \begin{equation}
>     z(t) = z(0) + \int_0^t f_\theta(z(s))\, dX(s),
> \end{equation}
> where $f_\theta$ is the learnable dynamics function and $X$ is the cubic-spline control path.
>
> To obtain $z_0 \in \mathbb{R}^{{BatchSize} \times {HiddenDim}}$, we take the observed values $x_0 \in \mathbb{R}^{{BatchSize} \times {Channels}}$ at $t=0$ from $X$ and map them into the latent space using the pre-trained encoder $f_{CE}$.
>
> **Explanation of reconstruction function**: After solving the NCDE, we obtain latent trajectories, $z_T \in \mathbb{R}^{\{BatchSize} \times \{SeqLen} \times \{HiddenDim}}$. To recover the projected variables (channels), we apply the decoder $f_{CD}$, yielding $f_{CD} (z_T)$, which serves as the reconstruction function. All explanations have been added to the revised paper of **line 712 to line 741 of Appendix section B CHANNEL-WISE AUTOENCODER AND ITS CONNECTION TO NCDE**.
>
> **iii) The training process of the MOE router is not described.**
>
> **Response to iii):**  We thank the reviewer for highlighting this omission. As mentioned in lines 187–188 of the original manuscript (the second paragraph of Section 3.2), MoE dynamics consists of the router $\mathcal{R}$ and dynamic functions $f_{\theta}$. The MOE router $\mathcal{R}$ serves as the front-end component for dynamic-function selection and is trained jointly with NCDE in an end-to-end manner. In Algorithm 1 “Training dynamic functions with a mixture of experts”, line 8 originally states “update MOE only”. This step indeed includes updating both the MOE experts and the router. To avoid ambiguity, we have revised the text to “update $f_{\rm MoE}$ (includes router $\mathcal{R}$) only” in **line 8 of Algorithm 1.**

---

> ### Author Response · Authors · 2025-11-21
> **Response to Weakness 2**
>
> **Weakness 2:** Several related works are missing. While the main comparisons are with NCDE-based models, other works [1,2] can also perform the generation task. Both models can predict the distribution of observations at any time point given irregularly sampled data.
>
> **Response to Weakness 2):** We sincerely thank the reviewer for bringing this important point to our attention. While our work primarily focuses on advancing NCDE-based methodologies and therefore emphasizes comparisons within this family, we appreciate the recommendation to consider additional related works addressing generation tasks under irregular sampling.
>
> We have now included HetVAE [1] as a baseline in our experiments. HetVAE is indeed capable of generating observations at arbitrary time points given irregularly sampled data based on attention mechanism, and we agree that it represents a meaningful point of comparison. The results have been added to **line 270 Table 1** (popular TSG datasets) and **line 324 Table 2** (medical datasets). As shown in the **Table 1 and Table 2 here**, our proposed MOE-NCDE achieves notably higher performance, suggesting that the use of multiple dynamic functions and control-path-based modeling allows our method to better capture diverse temporal patterns under missing-value conditions.
>
> Regarding ProFITi [2], we carefully examined its applicability. Although it is a normalizing-flow-based forecasting model, we attempt to adapt it to our generation scenario. However, during training, we found that only the forward (log-likelihood computation) training code is publicly available. The inverse-generation inference code—necessary for sampling new sequences—is not released. We have emailed the authors to request the code for the inference part, and we will add it as much as possible. Currently, we are unable to obtain generation results for this baseline, and thank you for your patience and understanding.
>
> We appreciate the reviewer’s constructive suggestions, which have helped us improve both the breadth and clarity of our experimental evaluation. We have integrated these methods into the revised related-work section (**line 112 to line 115**) and included HetVAE in the comparison baselines (**line 268**).
>
> **Table 1: Performance comparison with HetVAE on popular TSG datasets under irregular sampling, where 30%, 50%, and 70% of observations are removed. Results are averaged over sequence lengths of 12, 24, and 36. Detailed results for each length are provided in Appendix Tables 11, 12, and 13 of the revised manuscript. Lower values correspond to better fidelity and diversity in the generated data.**
> |    | Sines-30%|Stocks-30%| Energy-30%|MuJoCo-30%|Sines-50%|Stocks-50%| Energy-50%|MuJoCo-50%|Sines-70%|Stocks-70%| Energy-70%|MuJoCo-70%|
> |:-:|:-:|:-:|:-:|:----:|:-:|:-:|:-:|:----:|:-:|:-:|:-:|:----:|
> |Ours-DS|**0.105**| **0.142**| **0.422**| **0.293**| **0.128**| **0.137**| **0.487**| **0.375**| **0.182**| **0.106**| **0.497**| **0.393**|
> |HetVAE-DS|0.491| 0.328| 0.497| 0.499| 0.5| 0.443| 0.5| 0.499| 0.5| 0.487| 0.5| 0.5|
> |Ours-MDD|**0.953**| **0.25**| **0.270**| **0.347**| **1.093**| **0.281**| **0.252**| **0.318**| **1.308**| **0.299**| **0.279**| **0.297**|
> |HetVAE-MDD|3.158| 0.599| 0.625| 0.654| 3.449| 0.773| 1.095| 1.093| 3.967| 1.032| 1.323| 1.398|
> |Ours-KL|**0.013**| **0.074**| **0.020**| **0.021**| **0.023**| **0.094**| **0.022**| **0.009**| **0.033**| **0.091**| **0.017**| **0.014**|
> |HetVAE-KL|1.178| 1.085| 0.248| 0.258| 4.236| 1.4| 0.917| 0.996| 2.488| 1.612| 1.578| 2.202|
>
>
> **Table 2: Performance on medical datasets under irregular sampling, where 30%, 50%, and 70% of observations are removed. The sequence lengths of the four datasets are 96, 140, 136, and 82, respectively. Lower values correspond to better fidelity and diversity in the generated data.**
> |    | ECG200-30%|ECG5K-30%| ECGFD-30%|TLECG-30%|ECG200-50%|ECG5K-50%| ECGFD-50%|TLECG-50%|ECG200-70%|ECG5K-70%| ECGFD-70%|TLECG-70%|
> |:-:|:-:|:-:|:-:|:----:|:-:|:-:|:-:|:----:|:-:|:-:|:-:|:----:|
> |Ours-DS|**0.188**| **0.182**| **0.120**| **0.240**| **0.250**| **0.326**| **0.070**| **0.160**| **0.390**| **0.424**| **0.140**| **0.340**|
> |HetVAE-DS|0.5| 0.496| 0.42| 0.5| 0.5| 0.498| 0.43| 0.5| 0.5| 0.5| 0.43| 0.5|
> |Ours-MDD|**0.25**| **0.09**| **0.85**| **1.016**| **0.247**| **0.109**| **0.794**| **0.972**| **0.253**| **0.108**| **0.833**| **1.064**|
> |HetVAE-MDD|0.807| 0.492| 1.51| 1.608| 0.806| 0.461| 1.538| 1.576| 0.81| 0.569| 1.532| 1.594|
> |Ours-KL|**0.053**| **0.014**| **0.052**| **0.129**| **0.09**| **0.021**| **0.099**| **0.073**| **0.113**| **0.018**| **0.04**| **0.153**|
> |HetVAE-KL|17.216| 8.574| 8.218| 16.49| 17.199| 6.251| 13.445| 16.41| 17.193| 10.316| 13.445| 16.458|
>
> *[1] Shukla et al., "Heteroscedastic Temporal Variational Autoencoder for Irregularly Sampled Time Series," ICLR 2022*
>
> *[2] Yalavarthi et al., "Probabilistic Forecasting of Irregularly Sampled Time Series with Missing Values via Conditional Normalizing Flows," AAAI 2025*

---

> ### Author Response · Authors · 2025-11-21
> **Response to Questions 1 to 2**
>
> **Questions:**
>
> **Q1: What exactly are the channel-wise encoder and decoder? Given that the notation on p. 3 suggests univariate time series, how does “channel-wise” apply here?**
>
> **Response to Q1:** We thank the reviewer for this helpful question. Channel-wise encoder and decoder denotes the input and output dimensions of $f_{CE}$ and $f_{CD}$ match the number of time-series variables (channels), so they can be directly used to map channels or hidden states within NCDE. Specifically, they are used respectively for $z_0$ initialization and for mapping the latent state $z_T$ back to the variable dimensions. Both single-channel and multi-channel inputs are supported. A detailed explanation of the channel-wise encoder and decoder is provided in our Response to Weakness 1 (ii). In addition, although the example on p. 3 may give the impression of a univariate setting, our method is designed to handle both univariate and multivariate time series, and we have conducted experiments on both types (ECGs are univariate while the remaining are multivariate datasets).
> To avoid confusion, we have revised the description on p. 3 to explicitly state the multivariate setting (please refer to **line 149 to line 157**).
>
> **Q2: In line 7 of Algorithm 1, is only one expert being used? It is unclear how the top-K experts are utilized, and Eq. 3 does not clarify this.**
>
> **Response to Q2:**  We appreciate the reviewer’s careful reading. $s_i$ in line 7 denotes the group of experts associated with the $i$-th sample, rather than a single expert. As stated on line 267 of the **original manuscript**, the number of experts used in our experiments is 4. We have revised the text as "Get all expert weights $s$ for j-th sample from..." to avoid confusion (**please refer to lines 223 to line 224**).
>
>
> Regarding top-K: In our method, K is set to the total number of experts, meaning that all experts are used, i.e., we use dense MoE instead of sparse MoE. We have refined the wording to make the MoE mechanism clearer. We have added descriptions to make this explicit and to state clearly that the outputs of all experts are combined using their expert weights (**please refer to line 197 to 202**). Besides, to avoid confusion, we have removed "Top-K'" descriptions.  We truly appreciate the reviewer for pointing out this ambiguity.

---

> ### Author Response · Authors · 2025-11-21
> **Response to Questions 3 to 4**
>
> **Q3: the model seems to work only for univariate irregular time series. Is there a way to extend it to multivariate series?**
>
> **Response to Q3:** Thank you for raising this point. Our method supports multivariate time series. In our experiments, the number of variables for the sine, stock, energy, and mujoco datasets is 5, 6, 28, and 14, respectively. These results demonstrate that our method fully supports multivariate inputs and performs effectively in multivariate settings.
>
> **Q4: How does the diffusion model ensure that the generated ranks for the router are positive and sum to 1?**
>
> **Response to Q4:** We sincerely thank the reviewer for this insightful question.
>
> **Regarding positivity**: During training, all expert weights are positive. In practice, the diffusion model learns this distribution well, and we have not observed negative weights in the dataset. As shown in **Table 1**, the minimum expert weight across all samples and datasets remains strictly above zero.
>
> **Regarding the sum-to-one property:** We do not enforce a hard constraint to force the generated weights to sum to 1. Instead, we allow the diffusion model to learn this property implicitly—a form of adaptive soft constraint. As reported in **Table 2**, the sum of the generated weights is extremely close to 1 across all datasets, indicating that the model naturally preserves this property without explicit enforcement.
>
> **Quantitative evaluation:** **Table 3** compares forecasting performance using (i) raw generated weights and (ii) applying post-normalization to the raw generated weights so that they sum to 1. The results show almost no difference between the two settings. This suggests that the learned joint distribution already produces weight vectors that are appropriate for each sample.
>
> **Qualitative evaluation:** **Appendix Figures 8 and 9** visualize the effect of normalizing the weights. The qualitative results are fully consistent with the quantitative findings: whether or not the weights are normalized to sum to 1 has no noticeable impact on the generated trajectories.
>
> We appreciate the reviewer’s thoughtful comments, which helped us further clarify this aspect of our model. The above discussion has been consolidated in **Appendix line 823, F.1 section** **FURTHER DISCUSSIONS ON MOE WEIGHTS**.
>
>
> **Table 1:  Global minimum of the individual expert weights generated by the diffusion model across all samples and datasets.**
> |    | Sines|Stocks| Energy|MuJoCo|
> |:-:|:-:|:-:|:-:|:----:|
> |Global minimum|0.1566|0.1553|0.1345|0.1393|
>
> **Table 2:  Summary statistics of the sum of expert weights produced by the diffusion model across all samples from different datasets.**
> |    | Sines|Stocks| Energy|MuJoCo|
> |:-:|:-:|:-:|:-:|:----:|
> |Mean|1.0114|1.0046|1.0089|1.0085|
> |Maximum|1.0255|1.0305|1.0235|1.0163|
> |Minimum|0.9894|0.9766|0.9887|0.9726|
> |Standard deviation|0.0027|0.0044|0.0045|0.0068|
>
>
> **Table 3: Impact of enforcing the weights to sum to 1 on forecasting performance under continuous generation. Results are averaged over sequence lengths of 12, 24, and 36. “Normalize” indicates that each weight vector is divided by its sum so that it sums to 1, while “Ori” refers to directly using the raw weights generated by the diffusion model.**
> |    | Sines-30%|Stocks-30%| Energy-30%|MuJoCo-30%|Sines-50%|Stocks-50%| Energy-50%|MuJoCo-50%|Sines-70%|Stocks-70%| Energy-70%|MuJoCo-70%|
> |:-:|:-:|:-:|:-:|:----:|:-:|:-:|:-:|:----:|:-:|:-:|:-:|:----:|
> |Ori-MSE|0.026 |0.002 |0.013 |0.029 |0.026 |0.002 |0.013 |0.029 |0.028 |0.002 |0.013 |0.029|
> |Normalize-MSE|0.026 |0.002 |0.013 |0.029 |0.026 |0.002 |0.013 |0.029 |0.028 |0.002 |0.014 |0.029|
> |Ori-MAE|0.125 |0.025 |0.075 |0.129 |0.126 |0.026 |0.073 |0.131 |0.128 |0.025 |0.075 |0.131|
> |Normalize-MAE|0.125 |0.025 |0.075 |0.129 |0.126 |0.026 |0.073 |0.131 |0.128 |0.025 |0.075 |0.131|

---

> ### Author Response · Authors · 2025-11-28
> **Response to Weakness 2 (Adding new results on baseline ProFITi)**
>
> After receiving the authors’ response and updated code, we successfully applied their method to our own setting. We are pleased to share new experimental results.
>
> We note that ProFITi—benefiting from its distinctive triangular attention layers and normalizing flows—shows potential for irregular data modeling. However, it still performs significantly worse than our method across all metrics, particularly (1) as the proportion of missing values increases, and (2) when applied to medical datasets with complex temporal patterns.
>
>
> For example, its comparatively poorer DS metric may indicate that the generated samples lack key features and diversity, so it is easily identified by the discriminator. This is likely because ProFITi fails to capture complex temporal patterns with sufficient accuracy, thereby struggling to generate novel and important temporal dynamics effectively.
>
> The new results have been incorporated into **Tables 1 and 2 in the revised manuscript**.
>
> **Table 1: Performance comparison with **ProFITi** on popular TSG datasets under irregular sampling, where 30%, 50%, and 70% of observations are removed. Results are averaged over sequence lengths of 12, 24, and 36. Detailed results for each length are provided in Appendix Tables 11, 12, and 13 of the revised manuscript. Lower values correspond to better fidelity and diversity in the generated data.**
> |    | Sines-30%|Stocks-30%| Energy-30%|MuJoCo-30%|Sines-50%|Stocks-50%| Energy-50%|MuJoCo-50%|Sines-70%|Stocks-70%| Energy-70%|MuJoCo-70%|
> |:-:|:-:|:-:|:-:|:----:|:-:|:-:|:-:|:----:|:-:|:-:|:-:|:----:|
> |Ours-DS|**0.105**| **0.142**| **0.422**| **0.293**| **0.128**| **0.137**| **0.487**| **0.375**| **0.182**| **0.106**| **0.497**| **0.393**|
> |ProFITi-DS|0.307|0.454|0.495|0.492|0.344|0.483|0.499|0.497|0.392|0.489|0.5|0.497
> |Ours-MDD|**0.953**| **0.25**| 0.270| 0.347| **1.093**| **0.281**| **0.252**| **0.318**| **1.308**| **0.299**| **0.279**| **0.297**|
> |ProFITi-MDD|1.163|0.297|**0.255**|**0.295**|1.236|0.408|0.35|0.353|1.443|0.445|0.442|0.448
> |Ours-KL|**0.013**| **0.074**| **0.020**| 0.021| **0.023**| **0.094**| **0.022**| **0.009**| **0.033**| **0.091**| **0.017**| **0.014**|
> |ProFITi-KL|0.079|0.074|0.022|**0.015**|0.112|0.121|0.033|0.015|0.138|0.132|0.045|0.031
>
>
> **Table 2: Comparison with **ProFITi** on medical datasets under irregular sampling, where 30%, 50%, and 70% of observations are removed. The sequence lengths of the four datasets are 96, 140, 136, and 82, respectively. Lower values correspond to better fidelity and diversity in the generated data.**
> |    | ECG200-30%|ECG5K-30%| ECGFD-30%|TLECG-30%|ECG200-50%|ECG5K-50%| ECGFD-50%|TLECG-50%|ECG200-70%|ECG5K-70%| ECGFD-70%|TLECG-70%|
> |:-:|:-:|:-:|:-:|:----:|:-:|:-:|:-:|:----:|:-:|:-:|:-:|:----:|
> |Ours-DS|**0.188**| **0.182**| **0.120**| **0.240**| **0.250**| **0.326**| **0.070**| **0.160**| **0.390**| **0.424**| **0.140**| **0.340**|
> |ProFITi-DS|0.48|0.496|0.3|0.45|0.462|0.497|0.38|0.45|0.48|0.498|0.36|0.46
> |Ours-MDD|**0.25**| **0.09**| **0.85**| **1.016**| **0.247**| **0.109**| **0.794**| **0.972**| **0.253**| **0.108**| **0.833**| **1.064**|
> |ProFITi-MDD|0.31|0.185|0.989|1.074|0.298|0.196|1.006|1.115|0.325|0.247|1.003|1.138
> |Ours-KL|**0.053**| **0.014**| **0.052**| **0.129**| 0.09| **0.021**| **0.099**| **0.073**| **0.113**| **0.018**| **0.04**| **0.153**|
> |ProFITi-KL|0.085|0.053|0.204|0.245|**0.073**|0.044|0.262|0.346|0.149|0.066|0.395|1.188

---

### Author Response · Authors · 2025-11-21

We sincerely thank all reviewers for their time, constructive comments, and valuable feedback. Your insights have significantly improved the clarity, rigor, and overall quality of our work.

We have carefully addressed all comments and incorporated the corresponding revisions into the manuscript. **All updated or newly added content has been highlighted in blue in the revised version to ensure clarity and ease of review.**

We deeply appreciate your thoughtful review and efforts in helping us strengthen this paper, and we would be glad to further discuss any remaining concerns or clarify points where uncertainty may persist.

We look forward to your further feedback.

---

> ### Author Response · Authors · 2025-12-02
> **Revision Summary of the Newly Uploaded Manuscript**
>
> To address the reviewers’ concerns, the revisions in the newly uploaded manuscript are summarized as follows:
>
> 1. **Lines 112–115:**  Add descriptions of the ProFITi and HeTVAE baselines in the related work section.
> 2. **Lines 149–227:** Refine local descriptions (marked in blue text), including a clearer definition of the multivariate-generation task, clearer explanations of $z(0)$ and the reconstruction function in NCDE, and a clearer definition of the mixture-of-experts dynamic function.
> 3. **Lines 267–268:**  Add ProFITi and HeTVAE to the baseline of the experimental settings.
> 4. **Line 270 (Table 1) & Line 324 (Table 2):**  Add experimental results for ProFITi and HeTVAE.
> 5. **Lines 321–323:**  Added privacy-leakage-risk evaluation results for different methods.
> 6. **Line 443 (Table 5) & Lines 462–468:**  Compare MoE with a single expert by stacking the same number of parameters, showing that MoE is not equivalent to simply stacking parameters in a single expert.
> 7. **Lines 479–481 & Line 498 (Figure 6):**  Visualize the temporal patterns learned by different experts.
> 8. **Lines 712–742:**  Add Algorithm 4 and detailed descriptions of the channel-wise autoencoder.
> 9. **Lines 756–789:**  Clarify differences between our method and existing approaches and highlighted our unique contributions.
> 10. **Lines 822–856 & Tables 6–9:**  Add analyses of MoE weights (positivity/negativity, sum, mean), demonstrating the generated weights are reasonable and effective.
> 11. **Line 1079 & Lines 1112–1126 (Figures 14–16):**  Demonstrate that NCDE can learn non-smooth temporal patterns because, despite the control path being smooth, the nonlinear MOE dynamic functions can capture non-smooth behaviors.
> 12. **Lines 1127–1132 (Figures 11–13):**  Show that in continuous-generation tasks, MoE-NCDE can learn diverse temporal patterns, rather than only “saw-tooth” patterns.
> 13. **Line 1113 & Lines 1166–1179:**  Discuss the strategy for selecting the number of experts, and this hyperparameter shows good generalizability.

---

### Meta-Review · Area_Chair_88px · 2026-01-05

**Summary:**

This paper combines the mixture-of-expert and the neural controlled differential equation to better generate time series. The neural controlled differential equation is specialized to process irregular time seires for its spline algorithm for interpolating irregular observations. The mixture-of-exprt in this paper performs the dense routing to integerate multiple neural controlled differential models. The authors conducted experiments with strandard benchmark datasets and some baseline models. According to their analyses, the proposed MOE-NCDE shows the overall best performance.

**Reviewer Concerns:**

There are several concerns on this paper: i) the motiviation of adopting the mixture-of-expert concept for the neural controlled differential equation, ii) the lack of relialistic evaluation methods relfecting real-world irregular time series downstream tasks, iii) some unclear model architecture design motivations, iv) the motivation of utilizing the neural controlled differential equation in spite of its spline interpolation algorithm, v) the novelty of model design, and vi) missing privacy-related experiments.

Among them, iv) and v) are ciritical, raised by Reviewer YdVt. I also think that the overall model design is too naive. For instance, the dense mixture-of-experts is not frequently used in other domains but the authors do not explain anything on this. We typically choose top-K experts. The motiviation of adpoting the neural controlled differential equation for generating time series should also be developed in a more rigorous way.

**Reviewer Scores:**

I think that Reviewer YdVt and Reviewer gENa will not change his/her score even after the rebuttal.

However, the comments by Reviewer UQbt seems rather superficial so I cannot properfly expect how he/she will react.

---

### Decision · Program_Chairs · 2026-01-26

Reject